# An analytic theory of convolutional neural network inverse problems solvers

**Minh-Hai Nguyen** [1] **Quoc-Bao Do** [2] **Edouard Pauwels** [3] **Pierre Weiss** [1]

## Abstract

Supervised convolutional neural networks (CNNs) are widely used to solve imaging inverse problems, achieving state-of-the-art performance in numerous applications. However, despite their empirical success, these methods are poorly understood from a theoretical perspective and often treated as black boxes. To bridge this gap, we analyze trained neural networks through the lens of the Minimum Mean Square Error (MMSE) estimator, incorporating functional constraints that capture two fundamental inductive biases of CNNs: translation equivariance and locality via finite receptive fields. Under the empirical training distribution, we derive an analytic, interpretable, and tractable formula for this constrained variant, termed Local-Equivariant MMSE (LE-MMSE). Through extensive numerical experiments across various inverse problems (denoising, inpainting, deconvolution, accelerated MRI), datasets (FFHQ, CIFAR-10, FashionMNIST, FastMRI), and architectures (U-Net, ResNet, PatchMLP), we demonstrate that our theory matches the neural networks outputs (PSNR $\gtrsim 25$dB). Furthermore, we provide insights into the differences between *physics-aware* and *physics-agnostic* estimators, the impact of high-density regions in the training (patch) distribution, and the influence of other factors (dataset size, patch size, *etc.*).

## 1. Introduction

Let $A : \mathbb{R}^N \to \mathbb{R}^M$ be a linear mapping, $\boldsymbol{x}$ a random vector in $\mathbb{R}^N$. We consider the forward model with additive white Gaussian noise

$$\boldsymbol{y} = A\boldsymbol{x} + \boldsymbol{e} \qquad \text{where} \qquad \boldsymbol{e} \sim \mathcal{N}\left(0, \sigma^2 \mathrm{I}_M\right) \qquad (1)$$

[1]IRIT & CBI, CNRS & Université Toulouse, France [2]INSA Toulouse, Université Toulouse, France [3]Toulouse School of Economics, Université Toulouse Capitole, France. Correspondence to: Minh-Hai Nguyen <minh-hai.nguyen@irit.fr>.

*Proceedings of the $43^{rd}$ International Conference on Machine Learning*, Seoul, South Korea. PMLR 306, 2026. Copyright 2026 by the author(s).

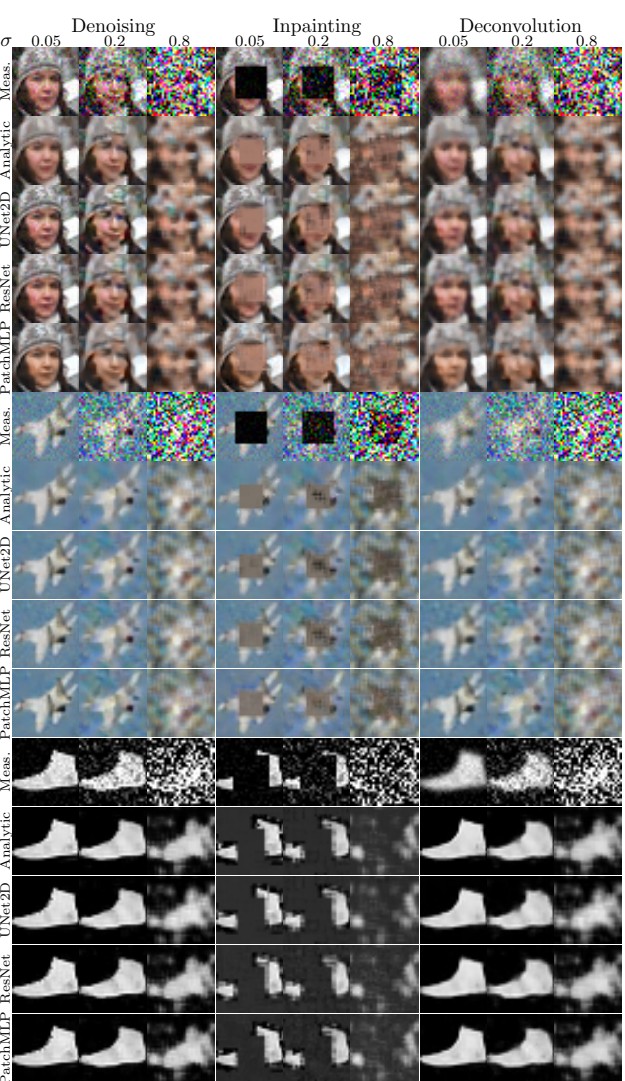

*Figure 1.* Our analytic theory accurately predicts neural network outputs across settings. We consider three inverse problems: denoising, inpainting, and deconvolution (left to right), on FFHQ, CIFAR10, and FashionMNIST datasets (top to bottom) with varying noise levels $\sigma$ (columns). For each setting, we show the measurements (top row), our analytic LE-MMSE estimator (second row), and outputs of trained UNet, ResNet, and PatchMLP models (last three rows). Theory closely matches network outputs.

and aim to construct an estimator $\hat{x}(\boldsymbol{y})$ of $\boldsymbol{x}$ given $\boldsymbol{y}$. A variety of neural network architectures have been proposed

for this task. Some are *physics-agnostic*, *i.e.* do not depend on $A$ explicitly and often rely on fully connected layers (Zhu et al., 2018). Others are *physics-aware*, and rely on an approximate inverse of $A$. In this work, we focus on linear operators $A$, but the analytical formulas extend to nonlinear operators as well. We consider estimators of the following form

$$\hat{x}(\boldsymbol{y}) = N_w(B\boldsymbol{y}), \tag{2}$$

where $N_w$ is a neural network with weights $w \in \Theta$ and $B \in \mathbb{R}^{N \times M}$ is a linear operator such as the identity when $M = N$ (physics-agnostic), or an approximate inverse of $A$ (physics-aware) such as the transpose $A^\top$, the pseudo-inverse $A^+$, or a Tikhonov-regularized inverse. Such designs were popularized in (Kim et al., 2016; Jin et al., 2017; Schlemper et al., 2017; Würfl et al., 2016) and are still widely used in practice (McCann et al., 2017; Zhou et al., 2022; Wang et al., 2020). They can be considered as a robust baseline, and be interpreted as a single-step variant of unrolled neural networks (Adler & Öktem, 2018b; Celledoni et al., 2021), which often achieve top performance in empirical benchmarks (Muckley et al., 2021).

**Empirical MMSE estimators**

**Definition 1.1** (Constrained empirical MMSE estimators)**.** Let $\mathcal{D}$ be a dataset of clean images. We define the empirical MMSE estimators as $\hat{x}_{\mathcal{M}} \stackrel{\text{def}}{=} \phi_{\mathcal{M}}^\star \circ B$, where

$$\phi_{\mathcal{M}}^\star \stackrel{\text{def}}{=} \underset{\phi \in \mathcal{M}}{\arg\min} \ \frac{1}{2} \mathbb{E}\left[ \|\phi(B\boldsymbol{y}) - \boldsymbol{x}\|^2 \right]. \tag{3}$$

Here, $\boldsymbol{x}$ follows the empirical distribution of the dataset $p_{\mathcal{D}} = \frac{1}{|\mathcal{D}|} \sum_{x \in \mathcal{D}} \delta_x$ and $\boldsymbol{y}$ defined by (1). The set $\mathcal{M}$ encodes the range of functions reachable by the neural network, $\mathcal{M} = \{N_w : w \in \Theta\}$, with $\Theta$ a set of admissible weights.

A precise characterization of the set $\mathcal{M}$ is not available for a given neural network architecture $N_w$. However, we can relate the trained neural network with statistical estimators constrained to function classes encoding architectural constraints of the neural networks as follows.

**Definition 1.2** (MMSE, E-MMSE, LE-MMSE)**.** Table 1 defines the constraints $\mathcal{M}$ and estimators studied in this work.

*Table 1.* Different constraint sets correspond to different variants of the MMSE estimator.

| CONSTRAINT $\mathcal{M}$ | ESTIMATOR | SEE |
|---|---|---|
| (1) ANY MEASURABLE FUNC. | MMSE $-\hat{x}_{\text{MMSE}}$ | (2.1) |
| (2) (1) + TRANSLATION EQUIV. | E-MMSE $-\hat{x}_{\mathcal{T}}$ | (3.5) |
| (2) + LOCALITY | LE-MMSE $-\hat{x}_{\mathcal{T},\text{LOC}}$ | (3.8) |

These function classes capture different architectural constraints of the neural network $N_w$. It is well known that sufficiently deep and wide multi-layer perceptrons (MLPs)

can approximate any measurable function (Hornik et al., 1989), hence the MMSE $\hat{x}_{\text{MMSE}}$ is a proxy for unconstrained MLPs. The E-MMSE $\hat{x}_{\mathcal{T}}$ approximates CNNs without constraints on the receptive field, while the LE-MMSE $\hat{x}_{\mathcal{T},\text{loc}}$ approximates the set of functions reachable by CNNs with finite receptive fields.

In what follows, for results involving neural networks, we let $\mathcal{M}$ denote the function class reachable by the considered architecture, and for results involving analytical formulas, we let $\mathcal{M}$ denote the abstract function class in Table 1.

**Related works**  In the context of generative diffusion models, (Kamb & Ganguli, 2025; Scarvelis et al., 2025) derive closed-form expressions for the MMSE under additive Gaussian noise (denoising) with architectural constraints such as equivariance and locality, and use them to study memorization and generalization. Locality has also been examined in (Kadkhodaie et al., 2023) for modeling patch distributions. For imaging inverse problems, equivariance has been widely explored in self-supervised learning (Chen et al., 2023; Terris et al., 2024). In the supervised setting for linear inverse problems, it is standard to assume that a trained neural network approximates the MMSE (Adler & Öktem, 2018a). To the best of our knowledge, for general inverse problems, our work is the first to introduce additional functional constraints to more accurately characterize the action of neural networks and to derive closed-form expressions for the resulting constrained MMSE estimators.

We provide an analytic LE-MMSE formula and show that it precisely predicts trained CNN outputs across tasks, datasets, and architectures, see Figure 1 for an overview, Figures 5, 12 and 13 for quantitative comparisons and Appendix C for more results.

**Contributions**  In the context of inverse problems outlined above, our contributions and outline are as follows:

1. We review the MMSE estimator (Section 2).

2. We define constrained MMSE variants that capture inductive biases of CNNs and derive their closed-form expressions (Section 3). Remarkably, we find tractable and interpretable formulas that generalize those of (Kamb & Ganguli, 2025) to arbitrary linear inverse problems.

3. We analyze the theoretical behavior of these estimators (Section 3), including the influence of the pre-inverse operator $B$. We demonstrate that while the MMSE and E-MMSE heavily rely on memorizing the training data, the LE-MMSE constructs a patchwork of training patches, leading to superior generalization.

4. We establish strong connections between the LE-MMSE and classical non-local means. This reveals that modern CNNs can be rigorously interpreted as structured, data-

adaptive generalizations of these methods, providing a theoretical foundation that addresses the challenge of explainability and guides the design of future, physics-aware architectures.

5. We validate our theory on extensive numerical experiments (Section 4) with various architectures, inverse problems and dataset. We find strong agreement between network outputs and the LE-MMSE estimator (PSNR $\gtrsim$ 25 dB for all tasks). See Figure 1.

6. We show empirically that different choices of architectures converge to the same solution, which is shown to be very close to our analytical formula (Section 4.2).

7. We characterize where the theory better matches the practice – high-density regions of the *patch distribution*, and explain how CNNs generalize. The influence of additional factors (training set size, noise level, receptive field size) is also investigated (Section 4.3).

In what follows, we let bold lowercase letters (*e.g.* $\boldsymbol{x}, \boldsymbol{y}$) denote random vectors, and non-bold lowercase letters (*e.g.* $x, y$) denote their realizations. We let $\bar{x}$ denote a fixed but unknown signal to recover, and $y = A\bar{x} + e$ be a realization of the measurement $\boldsymbol{y}$ defined in (1), with noise $e \sim \mathcal{N}\left(0, \sigma^2 \mathrm{I}_M\right)$.

## 2. Unconstrained MMSE estimator

It is well known that the MMSE estimator coincides with the *conditional mean* almost surely (Kay, 1993):

$$\hat{x}_{\mathrm{MMSE}}(y) = \mathbb{E}\left[\boldsymbol{x}|B\boldsymbol{y} = By\right] = \int x \cdot p(x|By)\,dx.$$

When the data distribution $p_{\boldsymbol{x}}$ is replaced by the empirical measure $p_{\mathcal{D}} = \frac{1}{|\mathcal{D}|}\sum_{x \in \mathcal{D}} \delta_x$, with a finite dataset $\mathcal{D}$, the empirical MMSE estimator becomes a kernel regressor (Nadaraya, 1964; Watson, 1964):

**Proposition 2.1** (Closed-form of the MMSE ). *The MMSE estimator in Definition 1.2 writes, for any $y \in \mathbb{R}^M$,*

$$\hat{x}_{MMSE}(y) = \sum_{x \in \mathcal{D}} x \cdot w(x|y), \qquad (4)$$

*where* $w(x|y) = \frac{\mathcal{N}\left(By; BAx, \sigma^2 BB^\top\right)}{\sum_{x' \in \mathcal{D}} \mathcal{N}(By; BAx', \sigma^2 BB^\top)}$.

A proof is provided in B.1. $\mathcal{N}\left(\cdot; \mu, \Sigma\right)$ refers to the density of a (possibly degenerate) Gaussian distribution with mean $\mu$ and covariance $\Sigma$, see Definition A.1. The weights $w(x|y)$ is the posterior probability that training sample $x$ explains $By$. The denominator term ensures that they sum to 1.

*Remark* 2.2. When $B = \mathrm{I}$, $\hat{x}_{\mathrm{MMSE}}$ is exactly the *posterior mean*: $\hat{x}_{\mathrm{MMSE}}(\boldsymbol{y}) = \mathbb{E}\left[\boldsymbol{x}|\boldsymbol{y}\right]$.

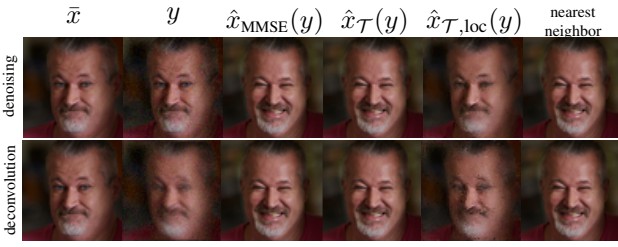

*Figure 2.* The MMSE and E-MMSE estimators memorize (yield the nearest neighbor), while the LE-MMSE estimator recombines training patches to give good reconstruction.

**Physics-aware estimator** To elucidate the role of $B$, we remark that the weights have the following explicit form:

$$w(x|y) \propto \exp\left(-\frac{1}{2\sigma^2}\left\|\Pi_{\mathrm{Im}(B^\top)}(A(\bar{x} - x) + e)\right\|^2\right)$$

Taking $B = A^+$ or $A^\top$, $\mathrm{Im}(B^T) = \mathrm{Im}(A)$ and the inner term is $A(\bar{x} - x) + \Pi_{\mathrm{Im}(A)}e$. The noise $e$ is projected onto the image of $A$, reducing its magnitude. Therefore, for the MMSE, physics-aware estimators is preferable to physics-agnostic ones, which is consistent with a popular belief and the usual practice. However, beyond the projection effect, the choice of $B$ only plays a little role, for example, the estimator is the same for any $B$ such that $\mathrm{Im}(B^T) = \mathrm{Im}(A)$. We discuss this further in Section 3.5.

**Memorization of the empirical MMSE** The weights in (4) satisfy $w(x|y) \geq 0$ and $\sum_{x \in \mathcal{D}} w(x|y) = 1$, hence $\hat{x}_{\mathrm{MMSE}}(y) \in \mathrm{conv}(\mathcal{D})$ — the convex envelop of the dataset $\mathcal{D}$, for every $y$. In particular, as $\sigma \to 0$, $\hat{x}_{\mathrm{MMSE}}(y)$ converges to $x \in \mathcal{D}$ such that $BAx$ is the closest to $By$ (1-nearest neighbor); ties being averaged. Therefore, the empirical MMSE "memorizes": it reconstructs by re-weighting training examples and never extrapolates outside $\mathrm{conv}(\mathcal{D})$, see Figure 2. This aligns with recent findings related to memorization in generative models (Kamb & Ganguli, 2025; Bertrand et al., 2025; Scarvelis et al., 2025).

## 3. Constrained MMSE estimators

We first provide details on the constraint classes and then describe our main theoretical results: analytical formulas and properties of constrained MMSE estimators.

### 3.1. Functional constraints: equivariance and locality

Motivated by the success of CNNs in inverse problems, we study function classes $\mathcal{M}$ that encode two core inductive biases – *equivariance* to geometric transformations (Cohen & Welling, 2016; Veeling et al., 2018) and *locality* via bounded receptive field (Kadkhodaie et al., 2023). We start with formal definition of these concepts (see Appendix A.1 for precise definitions). Let $x \in \mathbb{R}^N$ represent an image

defined on a discrete 2D grid of size $H \times W = N$.

**Definition 3.1** (Translation equivariant). Let $\mathcal{T} = \mathbb{Z}_H \times \mathbb{Z}_W$ be the group of 2D cyclic translations, where $\mathbb{Z}_H = \{0, 1, \dots, H - 1\}$ and $\mathbb{Z}_W = \{0, 1, \dots, W - 1\}$. For each $g = (g_h, g_w) \in \mathcal{T}$, its action is represented by a permutation matrix $T_g \in \mathbb{R}^{N \times N}$, where $T_g x$ shifts the image $x \in \mathbb{R}^N$ by $g_h$ pixels vertically and $g_w$ pixels horizontally with periodic boundary conditions. A measurable map $\phi : \mathbb{R}^N \to \mathbb{R}^N$ is said to be *translation equivariant* if it satisfies $\phi(T_g x) = T_g \phi(x)$ for all $x \in \mathbb{R}^N$ and $g \in \mathcal{T}$. We denote by $\mathcal{M}_\mathcal{T}$ the set of all such maps.

We now add locality constraint. For each pixel $n$, consider the patch extractor $\Pi_n : x \in \mathbb{R}^N \mapsto \Pi_n x = x[\omega_n] \in \mathbb{R}^P$ that extracts a square patch $x[\omega_n]$ of size $\sqrt{P} \times \sqrt{P}$ centered at pixel $n$, and $\omega_n$ is the set of pixel indices in the patch with circular boundary conditions.

**Definition 3.2** (Local and translation equivariant). A measurable map $\phi : \mathbb{R}^N \to \mathbb{R}^N$ is *local and translation equivariant* if there exists a measurable map $f : \mathbb{R}^P \to \mathbb{R}$ such that, for all $x \in \mathbb{R}^N$ and any pixel $n$, the output of $\phi$ at pixel $n$ denoted by $\phi(x)[n]$, depends only on the local patch $x[\omega_n]$, that is $\phi(x)[n] = f(\Pi_n x)$. We denote by $\mathcal{M}_{\mathcal{T},\mathrm{loc}}$ the set of all such functions.

The class $\mathcal{M}_{\mathcal{T},\mathrm{loc}}$ captures standard CNNs with small kernels and weight sharing or patch-based models, *e.g.* MLPs acting on patches (Khorashadizadeh et al., 2025).

### 3.2. Preliminary remarks

**Constraints and projection**   The constrained MMSE has a simple geometric interpretation, see proof in A.6.

**Proposition 3.3.** *For a closed set $\mathcal{M}$, the $\mathcal{M}$-constrained MMSE estimator in Definition 1.1 is the **orthogonal projection** in $L^2$ of the posterior mean $\mathbb{E}[\boldsymbol{x}|\boldsymbol{y}]$ onto the subspace of random vectors of the form $\mathcal{X} = \{\phi(B\boldsymbol{y}) : \phi \in \mathcal{M}\}$, that is $\hat{x}_\mathcal{M}(\boldsymbol{y}) = \Pi_\mathcal{X}(\mathbb{E}[\boldsymbol{x}|\boldsymbol{y}])$.*

**Architecture equivariance and data augmentation**   A standard approach to obtain equivariant estimators is by using data augmentation: the set $\mathcal{D}$ is replaced by $\mathcal{T}(\mathcal{D}) = \{T_g x : x \in \mathcal{D}, g \in \mathcal{T}\}$ at training time. This promotes *reconstruction equivariance* (Chen et al., 2021) of $\hat{x}_\mathcal{M}$:

$$\hat{x}_\mathcal{M}(A T_g \bar{x} + e) = T_g \hat{x}_\mathcal{M}(A \bar{x} + T_g^{-1} e). \tag{5}$$

The *data-augmented* MMSE estimators, are defined below.

**Definition 3.4** (Data-augmented MMSE estimator). The data-augmented MMSE estimator $\hat{x}_\mathcal{M}^{\mathrm{aug}}$ is defined as the MMSE estimator $\hat{x}_\mathcal{M}$ with respect to the empirical measure on $\mathcal{T}(\mathcal{D})$.

Reconstruction equivariance is often a desired property, but it differs from the *structural equivariance* of $\phi$ (Chen et al., 2020; Nordenfors & Flinth, 2025), as illustrated in Figure 3.

### 3.3. Translation Equivariant MMSE estimator

We start by the closed-form of the E-MMSE estimator $\hat{x}_\mathcal{T}$ and its properties. See Appendix B.2 for proofs.

**Theorem 3.5** (Closed-form of E-MMSE). *The E-MMSE estimator writes, for any $y \in \mathbb{R}^M$*

$$\hat{x}_\mathcal{T}(y) = \sum_{x \in \mathcal{D}, g \in \mathcal{T}} T_g x \cdot w_g(x|y), \tag{6}$$

*where $w_g(x|y) \propto \mathcal{N}\left(T_g^{-1} B y; B A x, \sigma^2 B B^\top\right)$.*

*Remark* 3.6. This result holds true for any group $\mathcal{G}$ of orthogonal transformations, not just translations.

The normalization constant of the weights $w_g(x|y)$ ensures that they sum to 1 over all $x \in \mathcal{D}$ and $g \in \mathcal{T}$, we ignore it here for brevity. The E-MMSE estimator $\hat{x}_\mathcal{T}$ is a weighted average of the augmented dataset $\mathcal{T}(\mathcal{D})$, which is similar to $\hat{x}_\mathrm{MMSE}^{\mathrm{aug}}$ in Definition 3.4. In particular, for denoising ($A = B = \mathrm{I}$), they coincide. However, depending on the forward operator $A$ and the pre-inverse $B$, it can differ significantly from $\hat{x}_\mathrm{MMSE}^{\mathrm{aug}}$. Some properties of the E-MMSE $\hat{x}_\mathcal{T}$ are presented in Corollary 3.7 and illustrated in Figure 3.

**Corollary 3.7.** *The E-MMSE estimator $\hat{x}_\mathcal{T}$ in Theorem 3.5 satisfies the following properties:*

- *The E-MMSE estimator $\hat{x}_\mathcal{T}$ memorizes the augmented dataset: reconstructed images live in $\mathrm{conv}(\mathcal{T}(\mathcal{D}))$.*

- *Data-augmentation and architecture equivariance are identical ($\hat{x}_\mathcal{T} = \hat{x}_\mathrm{MMSE}^{\mathrm{aug}}$): if $A$ and $B$ are circular convolution, with $B$ invertible, then the E-MMSE is reconstruction equivariant (satisfies (5)). Moreover, physics-agnostic and physics-aware solvers are identical.*

- *This is near equivalent: for invertible $A, B$, if $w_g(x|y) = w(T_g x|y)$ then $A$ and $B$ are circular convolutions.*

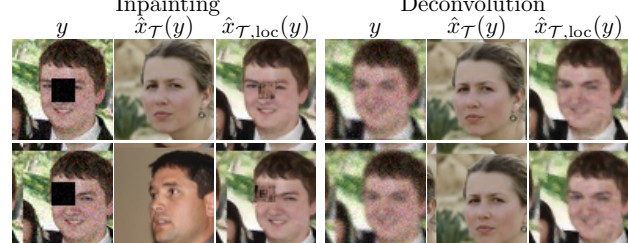

*Figure 3.* Architectural equivariance does not always guarantee reconstruction equivariance (5). Input $y$ in 2nd row is shifted. The E-MMSE estimator $\hat{x}_\mathcal{T}$ is reconstruction equivariant for deconvolution but not inpainting. The LE-MMSE estimator $\hat{x}_{\mathcal{T},\mathrm{loc}}$ is reconstruction equivariant for deconvolution and shows reduced sensitivity to shifts for inpainting.

## 3.4. Locality and Translation Equivariance

We now analyze the effect of local receptive fields. See Appendix B.3 for proofs.

**Theorem 3.8** (Closed-form of LE-MMSE). *Suppose that the $N$ matrices $Q_n \overset{\text{def}}{=} \Pi_n B \in \mathbb{R}^{P \times M}$ have the same rank $r > 0$ for any $n$[1]. The LE-MMSE estimator $\hat{x}_{\mathcal{T},loc}$ admits, for any $y \in \mathbb{R}^M$ the following expression, defined pixel-wise for each pixel $n'$:*

$$\hat{x}_{\mathcal{T},loc}(y)[n'] = \sum_{x \in \mathcal{D}} \sum_{n=1}^{N} x[n] \cdot w_{n',n}(x|y), \qquad (7)$$

*where $w_{n',n}(x|y) \propto \mathcal{N}\left(Q_{n'}y; Q_n A x, \sigma^2 Q_n Q_n^\top\right)$.*

The value at pixel $n'$ of the LE-MMSE estimator is a weighted average of *all* the pixels in the training images. We show that the resulting estimator can be interpreted as a *patchwork* of training patches in Figures 25 and 26. Interestingly, the analytic formula of the LE-MMSE estimator can be seen as a *data-driven* counterpart of the classical non-local means (NLM) estimator (Buades et al., 2005), where the patches are from a training set instead of the input image.

The recombination in equation 7 can produce images outside $\text{conv}(\mathcal{D})$ or $\text{conv}(\mathcal{T}(\mathcal{D}))$ (contrary to the MMSE or E-MMSE); see Figure 2 and Section 4. The weights $w_{n',n}(x|y)$ can be seen as the posterior probability of the pixel $x[n]$ given the patch $(By)[\omega_{n'}] = Q_{n'}y$.

Recall that $y = A\bar{x} + e$, with $\bar{x}$ the signal to recover and let $\Delta_{n',n}(\bar{x}, x) \overset{\text{def}}{=} (BA\bar{x})[\omega_{n'}] - (BAx)[\omega_n]$, the weights can be rewritten as $w_{n',n}(x|y) \propto \exp\left(-\frac{\eta^2}{2\sigma^2}\right)$ with:

$$\eta \overset{\text{def}}{=} \left\| Q_n^+ \Delta_{n',n}(\bar{x}, x) + Q_n^+ Q_{n'} e \right\|. \qquad (8)$$

Hence, the weights measure the similarity between *local patches* of measured-then-reconstructed images with the metric $Q_n^+$. It is perturbed by the noise term $Q_n^+ Q_{n'} e$, which is responsible for the estimator's variance. Some properties of the LE-MMSE $\hat{x}_{\mathcal{T},\text{loc}}$ are given in Corollary 3.9. See Appendix B.3 for proofs and Appendix D.4 for further discussions and illustrations.

**Corollary 3.9.** *The LE-MMSE estimator $\hat{x}_{\mathcal{T},loc}$ in Theorem 3.8 satisfies the following properties:*

- *The LE-MMSE estimator $\hat{x}_{\mathcal{T},loc}$ is a* patchwork *(see Appendix D.5) of training patches. As $\sigma \to 0$, each pixel is set to the central pixel of the best matching patch from $\mathcal{D}$.*

- *It is not a posterior mean: if $A$ is the identity (denoising), for a generic database $\mathcal{D}$, there exists no prior distribution of $\boldsymbol{x}$ such that $\hat{x}_{\mathcal{T},loc}(y) = \mathbb{E}[\boldsymbol{x}|\boldsymbol{y} = y]$.*

## 3.5. Physics-agnostic and physics-aware estimators

The expectation of $\eta^2$ in (8) is given by

$$\mathbb{E}_{\boldsymbol{e}}\left[\eta^2\right] = \left\| Q_n^+ \Delta_{n',n}(\bar{x}, x) \right\|^2 + \sigma^2 \cdot \text{Tr}\left(\text{Cov}_{n',n}\right)$$

with $\text{Cov}_{n',n} = Q_n^+ Q_{n'} Q_{n'}^\top Q_n^{+\top}$. This expression reveals that the pre-inverse $B$ plays two distinct roles:

- *Signal discrimination*: the term $Q_n^+ \Delta_{n',n}(\bar{x}, x)$ measures the amplification/reduction of difference between dissimilar/similar patches for the pre-inverse $B$. A good choice of $B$ should improve this discrimination.

- *Noise robustness*: the second term $\text{Tr}\left(\text{Cov}_{n',n}\right)$ measures noise amplification/reduction for $B$. A good choice should reduce this amplification.

Ideally, one would like to choose $B$ to optimize both criteria, but unfortunately they can be conflicting: preserving the signal $BAx \approx x$ may amplify the noise, in relation to the classical bias-variance decomposition. Most importantly this effect is operator dependent, as we now discuss.

For inpainting, choosing a physics-aware estimator with $B = A^+$ is an effective way to reduce the noise sensitivity while keeping signal's information. Indeed, with this choice, the noise within the masked region is eliminated, while the signal is preserved in the unmasked region. For deconvolution, however, the situation is different. Choosing $B = A^+$ can result in significant noise amplification (*i.e.* $\text{Tr}\left(\text{Cov}_{n',n}\right)$ is large), which is not counter-balanced by a better signal's discrimination. In this setting, a regularized inverse, balancing both aspects should likely be preferred. This effect is illustrated in Figure 4. When designing reconstruction algorithms, this trade-off between data-consistency and noise amplification must be carefully considered, in relation to the inverse problem at hand.

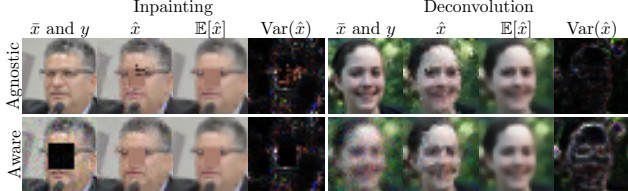

*Figure 4.* Physics-aware ($B = A^+$, bottom) has lower variance for inpainting (left), while physics-agnostic ($B = I$, top) has lower variance for deconvolution (right). Mean and pixel-wise variance are computed w.r.t 50 noise realizations.

# 4. Numerical Experiments

While the MMSE and E-MMSE estimators are interesting from a theoretical perspective, the LE-MMSE estimator is more relevant to practical neural network architectures used in imaging inverse problems. We validate our theoretical findings about the LE-MMSE on three representative inverse problems: denoising, inpainting, and deconvolution.

## 4.1. Experimental setup

We implement three different local and translation equivariant neural network architectures: UNet2D (Ronneberger et al., 2015), ResNet (He et al., 2016) and PatchMLP (an MLP acting on patches). All architectures are modified to be *strictly* local and translation equivariant.

Implementing the analytical formula in Theorem 3.8 requires computing a huge amount of pairwise distances, which is computationally demanding. We therefore restrict our experiments to $32 \times 32$ color images and datasets of $10^4$ images. To accumulate Gaussian weights robustly and stably across batches, we use a careful online log-sum-exp implementation of the weighted averages.

For inpainting we use a square mask of size $15 \times 15$ at the center. For deconvolution, we use an isotropic Gaussian kernel of standard deviation of $1.0$. The noise level $\sigma$ is varying uniformly between $0$ and $1.0$ during training. Full details are described in Appendix C and the implementation is available at https://github.com/mh-nguyen712/analytical_mmse.

## 4.2. Analytical formula and neural networks

**Case-by-case neural outputs prediction** We verify that trained networks approximate the LE-MMSE estimator, by comparing their outputs to the formula in Theorem 3.8. The PSNR between the trained UNet2D, the formula and the ground truth are reported in Figure 5. The reconstruction quality (orange and blue curves) degrades as noise level increases, which is expected as noise makes the problem harder. However, the PSNR between neural networks and the analytical formula (green curves) remains high ($\gtrsim 25$ dB) across all noise levels, indicating a strong alignment between the two. Similar behavior is observed in Table 2 for other architectures and datasets. For completeness, we also report the SSIM and LPIPS metrics in Figure 14. We provide extensive numerical results on various settings (architectures, datasets, tasks, physics-agnostic/aware, patch sizes) in Appendix D.1. These results confirm that *trained neural networks closely approximate the analytical LE-MMSE estimator*. It is remarkable that different architectures consistently yield a similar output, as shown in Figures 1 and 15 to 18.

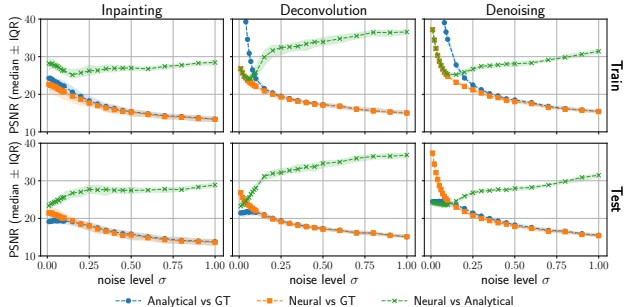

*Figure 5.* The green curves reveals a strong agreement (PSNR $\gtrsim 25$ dB) between the trained UNet2D and the analytical formula for all inverse problems. Median and interquartile range (IQR) using 50 images per $\sigma$, $P = 5 \times 5$ and $B = I$. See Figures 12 and 13 for other architectures.

**Low-density regions** A noticeable train-test gap remains in Figure 5 and Table 2, most pronounced at low noise level. We attribute this to the *measurement distribution* $p(y)$ induced by the empirical distribution. It is a Gaussian mixture, centered at the training measurements $Ax$: $p(y) = \frac{1}{|\mathcal{D}|} \sum_{x \in \mathcal{D}} \mathcal{N}\left(y; Ax, \sigma^2 I_M\right)$. Its density is high near the centers $Ax, x \in \mathcal{D}$, or for large noise levels $\sigma$, creating overlap between Gaussians. It becomes negligible far from the centers in low noise regimes.

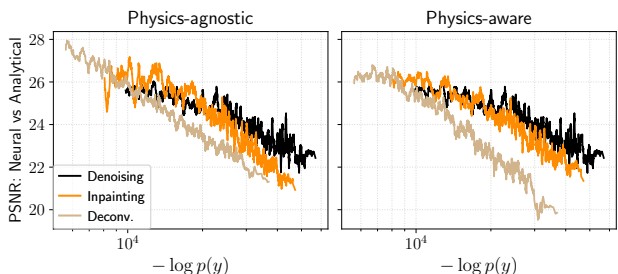

*Figure 6.* Higher density yield better alignment. We select test images from FFHQ-32, compute $-\log p(y)$ and plot it against the PSNR between the outputs of the LE-MMSE formula and a trained UNet2D, $\sigma = 0.05$ and $P = 5 \times 5$.

Both LE-MMSE and the networks are optimized with respect to the measurement distribution $p(y)$. The networks and the theoretical formula best align in high-density regions of $p(y)$. This phenomenon is illustrated in Figure 6, where we show that the alignment degrades as $-\log p(y)$ increases (or equivalently as $p(y)$ decreases). Notice that for large $\sigma$, the density is higher as the Gaussians overlap more, resulting in a better agreement.

**Out-of-distribution: how do networks generalize?** Neural networks often generalize well to out-of-distribution (OOD) data, but understanding this behavior remains an open question. Interestingly, our analytical formulas provide insights to explain this phenomenon. We consider here

*Table 2.* Theorem 3.8 is verified across architectures, tasks, and datasets. We show the median of the PSNR between neural networks and the analytical formula with $P = 5 \times 5$ and $B = I$. The PSNR on FashionMNIST is consistently higher ($2 \sim 3$ dB) than on FFHQ and CIFAR10, which we attribute to the lower complexity of FashionMNIST images. A lower PSNR on the test set in low-noise regimes is explained by the low-density regions.

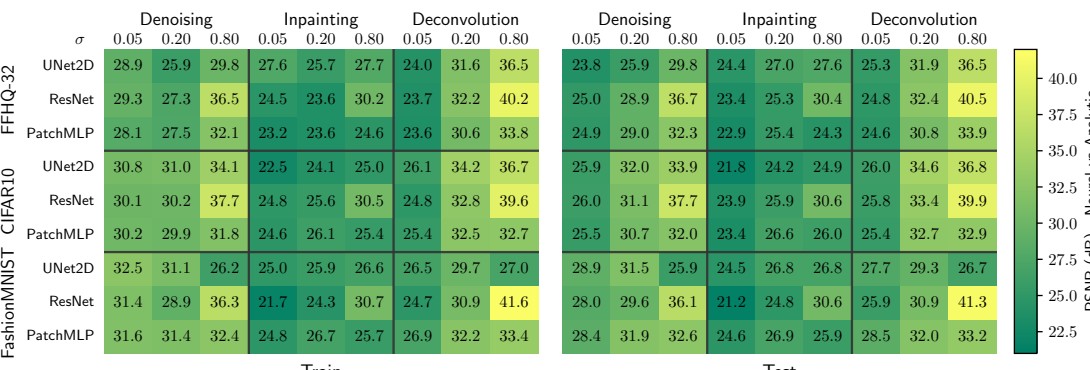

| | | Denoising | | | Inpainting | | | Deconvolution | | | | Denoising | | | Inpainting | | | Deconvolution | | |
|---|---|---|---|---|---|---|---|---|---|---|---|---|---|---|---|---|---|---|---|---|
| | $\sigma$ | 0.05 | 0.20 | 0.80 | 0.05 | 0.20 | 0.80 | 0.05 | 0.20 | 0.80 | | 0.05 | 0.20 | 0.80 | 0.05 | 0.20 | 0.80 | 0.05 | 0.20 | 0.80 |
| FFHQ-32 | UNet2D | 28.9 | 25.9 | 29.8 | 27.6 | 25.7 | 27.7 | 24.0 | 31.6 | 36.5 | | 23.8 | 25.9 | 29.8 | 24.4 | 27.0 | 27.6 | 25.3 | 31.9 | 36.5 |
| | ResNet | 29.3 | 27.3 | 36.5 | 24.5 | 23.6 | 30.2 | 23.7 | 32.2 | 40.2 | | 25.0 | 28.9 | 36.7 | 23.4 | 25.3 | 30.4 | 24.8 | 32.4 | 40.5 |
| | PatchMLP | 28.1 | 27.5 | 32.1 | 23.2 | 23.6 | 24.6 | 23.6 | 30.6 | 33.8 | | 24.9 | 29.0 | 32.3 | 22.9 | 25.4 | 24.3 | 24.6 | 30.8 | 33.9 |
| CIFAR10 | UNet2D | 30.8 | 31.0 | 34.1 | 22.5 | 24.1 | 25.0 | 26.1 | 34.2 | 36.7 | | 25.9 | 32.0 | 33.9 | 21.8 | 24.2 | 24.9 | 26.0 | 34.6 | 36.8 |
| | ResNet | 30.1 | 30.2 | 37.7 | 24.8 | 25.6 | 30.5 | 24.8 | 32.8 | 39.6 | | 26.0 | 31.1 | 37.7 | 23.9 | 25.9 | 30.6 | 25.8 | 33.4 | 39.9 |
| | PatchMLP | 30.2 | 29.9 | 31.8 | 24.6 | 26.1 | 25.4 | 25.4 | 32.5 | 32.7 | | 25.5 | 30.7 | 32.0 | 23.4 | 26.6 | 26.0 | 25.4 | 32.7 | 32.9 |
| FashionMNIST | UNet2D | 32.5 | 31.1 | 26.2 | 25.0 | 25.9 | 26.6 | 26.5 | 29.7 | 27.0 | | 28.9 | 31.5 | 25.9 | 24.5 | 26.8 | 26.8 | 27.7 | 29.3 | 26.7 |
| | ResNet | 31.4 | 28.9 | 36.3 | 21.7 | 24.3 | 30.7 | 24.7 | 30.9 | 41.6 | | 28.0 | 29.6 | 36.1 | 21.2 | 24.8 | 30.6 | 25.9 | 30.9 | 41.3 |
| | PatchMLP | 31.6 | 31.4 | 32.4 | 24.8 | 26.7 | 25.7 | 26.9 | 32.2 | 33.4 | | 28.4 | 31.9 | 32.6 | 24.6 | 26.9 | 25.9 | 28.5 | 32.0 | 33.2 |
| | | | | Train | | | | | | | | | | | Test | | | | | |

images $\bar{x}$ that lie in another dataset $\mathcal{D}'$ disjoint from the training set $\mathcal{D}$. In Figure 7, we compare the output of UNet2D

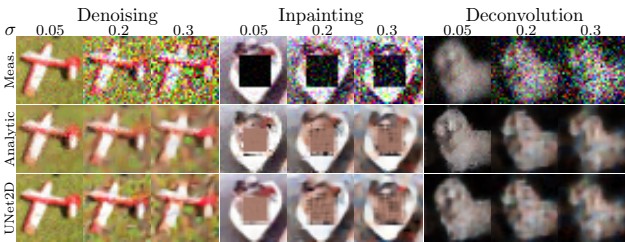

*Figure 7.* Our theory can predict the neural network outputs for out-of-distribution data: both UNet2D and the LE-MMSE formula are trained (or computed) on $\mathcal{D} = $ FFHQ-32. They are tested on $\mathcal{D}' = $ CIFAR10.

and the LE-MMSE formula. For large $\sigma$, we observe very close outputs between the neural network and the analytical formula. For small $\sigma$, as the OOD images lie in low-density regions of $p(y)$, the alignment degrades as expected from the previous discussion. However, even in this regime, the PSNR between the two outputs remains reasonably high (see Appendix D.6 for quantitative results). This suggests that the network approximates the LE-MMSE estimator even on OOD data: generalization of the neural networks can be understood through the lens of the LE-MMSE estimator – it leverages local patches from the training set to reconstruct unseen images.

### 4.3. Hyperparameter influence

**Patch size** The patch size $P$ in the LE-MMSE estimator plays a crucial role and should be chosen carefully depending on the inverse problem and noise level $\sigma$. For large noise levels, the inverse problem requires a higher degree of regularization, which can be achieved by using larger patches. On the contrary, small patches are preferred for low noise levels to capture fine details. This effect is illus-

trated in Figure 21, where $5 \times 5$ patches should be preferred for the test set at low noise levels, while $11 \times 11$ patches perform better at high noise levels.

On the other hand, the match between the analytical formula and the neural networks output is higher for small patches, as they depend on the density of the *measurement patches* distribution $p(y[\omega])$, see Figure 20. There are $N \cdot |\mathcal{D}|$ patches, but the space dimension is $P$. Consequently, for a fixed dataset size $|\mathcal{D}|$, increasing $P$ lowers the density of $p(y[\omega])$ significantly. A similar conclusion was already reached in denoising diffusion models, where (Kamb & Ganguli, 2025) proposed to vary the patch sizes from large to small across noise levels to guarantee a better match.

**Dataset size** We further investigate the influence of the dataset size $|\mathcal{D}|$ on the alignment between neural networks and the LE-MMSE formula. Specifically, we vary $|\mathcal{D}|$ from $10^3$ to $5 \times 10^4$ images on FFHQ-32 and trained UNet2D with $P = 5 \times 5$ and $B = I$. As shown in Figure 28, our theory holds for all dataset sizes, with small variations in PSNR between the neural network and the formula.

### 4.4. Smoothed LE-MMSE and spectral bias

Deep neural networks also exhibit a *spectral bias* (Xu et al., 2019; Rahaman et al., 2019), *i.e.* a preference for low-frequency functions, which yields smoother reconstructions in practice. Describing this spectral bias with functional constraints is missing from our derivation. We model this effect by considering a *randomized smoothing* (Duchi et al., 2012) variant: for a smoothing parameter $\epsilon > 0$ and $z \sim \mathcal{N}(0, I_M)$, define $\hat{x}_{\mathcal{T},\text{loc}}^{\text{smooth}}(y) \overset{\text{def}}{=} \mathbb{E}\left[\hat{x}_{\mathcal{T},\text{loc}}(y + \epsilon \cdot z)\right]$. This averages the estimates over random perturbations, acting as a nonlinear low-pass filter that attenuates high-frequency variability in the output. It also mirrors standard neural network training, where different noise real-

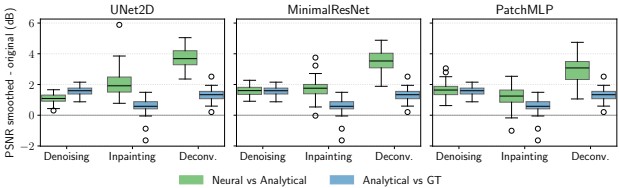

*Figure 8.* Smoothed LE-MMSE ($\epsilon = 0.05$) better matches neural network and improves reconstruction quality.

*Table 3.* The alignment between trained ResNet and the analytical formula of the LE-MMSE estimator under different noise levels $\sigma$.

| **Train** $(\sigma)$ | 0.01 | 0.02 | 0.05 | 0.1 | 0.2 | 0.3 |
|---|---|---|---|---|---|---|
| PSNR $\uparrow$ | 32.88 | 32.85 | 32.74 | 32.52 | 32.62 | 32.43 |
| SSIM $\uparrow$ | 0.891 | 0.890 | 0.889 | 0.892 | 0.908 | 0.894 |
| LPIPS $\downarrow$ | 0.081 | 0.081 | 0.077 | 0.067 | 0.062 | 0.128 |
| **Test** $(\sigma)$ | 0.01 | 0.02 | 0.05 | 0.1 | 0.2 | 0.3 |
| PSNR $\uparrow$ | 28.66 | 28.67 | 28.83 | 29.39 | 30.24 | 30.40 |
| SSIM $\uparrow$ | 0.827 | 0.828 | 0.839 | 0.863 | 0.895 | 0.880 |
| LPIPS $\downarrow$ | 0.237 | 0.232 | 0.212 | 0.152 | 0.105 | 0.156 |

izations are seen across epochs. In Figure 8, we compare the smoothed estimator $\hat{x}_{\mathcal{T},\text{loc}}^{\text{smooth}}$ with the original $\hat{x}_{\mathcal{T},\text{loc}}$, both computed with $P = 5 \times 5, \sigma = 0.05$ and $B = \text{I}$ on FFHQ-32 images. The expectation for the smoothed estimator is approximated by 50 Monte-Carlo samples. We observe that the smoothed LE-MMSE better matches the neural network output ($1 \sim 3$ dB) and improves reconstruction quality ($\sim 1$dB). This suggests that we can better capture neural network behavior by incorporating carefully designed smoothness constraints.

### 4.5. Validation on higher resolution images

To further validate our theoretical findings, we conduct experiments on FFHQ images of size $64 \times 64$ with $10^4$ samples. We train a UNet2D architecture with patch size $P = 11 \times 11$ for denoising, inpainting, and deconvolution tasks. We observe that the trained neural network closely approximates the analytical LE-MMSE estimator, further strengthening our conclusions (see Figures 9 and 29). Details about architecture, training and additional results are provided in Appendix D.8. Note that computing the analytical formula at this resolution is computationally intensive, limiting the extent of experiments compared to the $32 \times 32$ case.

Additional experiments on $128 \times 128$ images are provided in Appendix D.9, where we observe the same conclusions.

### 4.6. Beyond natural images: accelerated MRI

To demonstrate the applicability of our theoretical framework beyond natural images, we conduct an additional experiment on $4x$ accelerated MRI reconstruction using the fastMRI dataset (Zbontar et al., 2018). The forward operator $A = SF$ is a subsampled Fourier transform, where $F$ is the 2D Fourier transform and $S$ is a binary mask that selects $25\%$ of the Fourier coefficients. We trained a ResNet ($P = 11 \times 11$) on the knee singlecoil train subset (4865 slices of size $128 \times 128$). We observe in Table 3 and Figure 10 a good alignment between the neural and the LE-MMSE estimator both on the train ($\geq 32$dB) and the test set ($\geq 28.6$dB), showcasing the relevance of our framework for scientific inverse problems.

## 5. Discussions and conclusion

### 5.1. Neural networks: compress and rearticulate data

Although our analysis focuses on the local and translation equivariant MMSE estimator, the underlying conclusions extend to far more general learning settings. Neural networks compress and rearticulate the training data in an efficient manner. This viewpoint is consistent with recent findings in deep networks (Arpit et al., 2017), including large language models (Biderman et al., 2023) and image generation models (Somepalli et al., 2023), where new samples are synthesized by recombining and transforming elements implicitly stored from the training distribution.

In our experiments, neural networks with roughly $4 \times 10^6$ parameters are trained on datasets containing $10^4$ images of resolution 3×32×32, with approximately 3.1×$10^7$ pixels. The networks compress this information into a comparatively small number of parameters (roughly $13\%$ of the number of pixels) while still retaining the ability to reconstruct images by rearticulating training set patches. Furthermore, neural network inference is substantially faster than evaluating the analytical estimator: $\sim 4$ms versus $\sim 0.7$s for denoising, resulting in $\sim 175\times$ speedup. This gain is even more pronounced for physics-aware estimator or other operators: the formula inverses the operators $Q_n$ for each patch while the neural networks learn to approximate it from training data. This demonstrates the practical advantage of neural networks in approximating complex estimators such as the LE-MMSE: they not only compress the data but also enable efficient, near-instantaneous evaluation. Originally considered as black boxes, our analysis provides a theoretical foundation for understanding how neural networks achieve this remarkable feature.

### 5.2. Perspective and future works

**Reconstruction guarantee and stability** Neural networks often lack theoretical guarantees regarding reconstruction accuracy and stability. By modeling CNNs as LE-MMSE approximations, our closed-form expressions enable the theoretical analysis of these properties. This

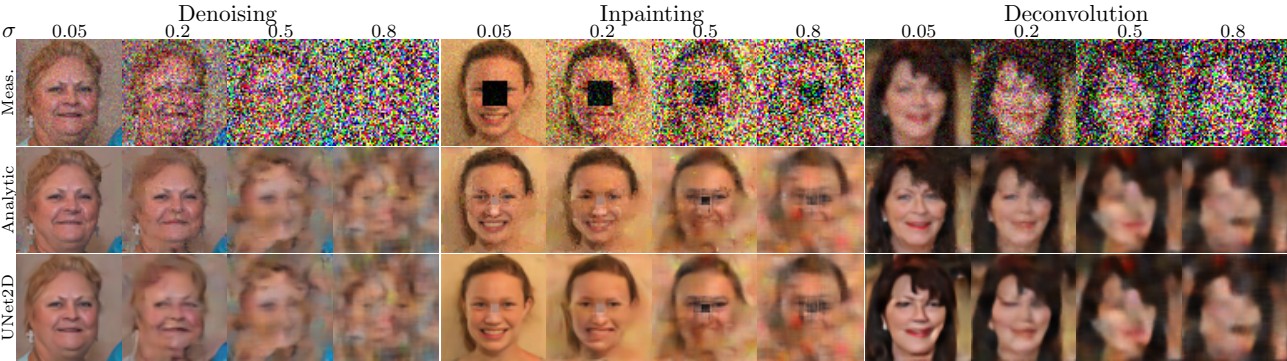

*Figure 9.* Our theory is validated at higher resolution: comparison of UNet2D output and the analytical formula on FFHQ-64.

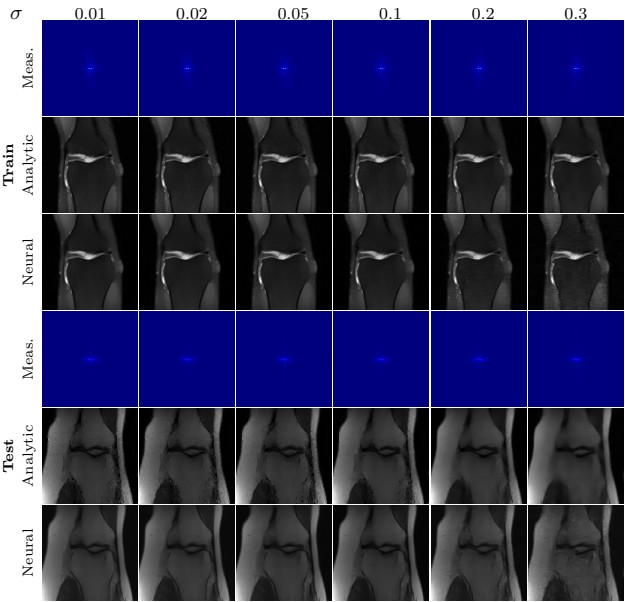

*Figure 10.* Qualitative comparison between the trained ResNet and the analytical formula of the LE-MMSE estimator on fastMRI.

## 5.3. Conclusion

We presented an analytic framework for CNN-based inverse problem solvers by considering MMSE with functional constraints. We derived closed-form expressions that make explicit the impact the forward operator $A$ and a pre-inverse $B$ on the reconstruction. The theory separates memorization and generalization behaviors: while the MMSE and E-MMSE memorize, the LE-MMSE forms patchwork reconstructions and closely matches trained CNN outputs across settings. This perspective turns black-box inverse problem CNNs into predictable estimators, enabling future theoretical analysis of generalization, stability, and principled choices of pre-inverses.

framework moves beyond purely empirical validation, offering a rigorous basis for understanding the strengths and limitations of neural inverse problem solvers.

**Extensions beyond current assumptions**   While we assumed in this work additive Gaussian noise and local, translation equivariant functional classes, the framework can be broadened. For example, vision transformers (Dosovitskiy, 2020) with self-attention capture long-range dependencies, but it is permutation equivariant (Xu et al., 2024) without positional embeddings. The Gaussian assumption can be relaxed (e.g., exponential-family) by using the appropriate likelihood $p_{\boldsymbol{y}|\boldsymbol{x}}(y|x)$. Finally, beyond MSE, practical training often uses $\ell_1$ (yielding posterior median), perceptual, or mixed losses. Analyzing the MMSE analogs under these alternatives is an interesting future venue.

## Acknowledgement

The authors acknowledge a support from the ANR Micro-Blind (ANR-620 21-CE48-0008) and from the ANR CLEAR-Microscopy (ANR-25-CE45-3780). This work was performed using HPC resources from GENCI-IDRIS (Grant AD011012210). EP thanks TSE-P and acknowledges the support of the AI Interdisciplinary Institute ANITI funding, through the ANR under the France 2030 program (grant ANR-23-IACL-0002), Chair TRIAL, the Madlearning project under the France 2030 program (ANR-25-PEIA-0002) and the ANR PRC MAD (ANR-24-CE23-1529).

## Impact Statement

This paper presents work whose goal is to advance the field of Machine Learning. There are many potential societal consequences of our work, none which we feel must be specifically highlighted here.

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

# A. Mathematical formalism

## A.1. Preliminaries

**Notation conventions**   In what follows, we use the following notation:

- Sets are indicated by calligraphic letters, *e.g.* $\mathcal{D}, \mathcal{M}, \mathcal{T}, \mathcal{G}$.

- Random vectors are denoted by bold lowercase letters, *e.g.* $\boldsymbol{x}, \boldsymbol{y}, \boldsymbol{z}$.

- $A \in \mathbb{R}^{M \times N}$ denotes a linear operator from $\mathbb{R}^N$ to $\mathbb{R}^M$ (*e.g.* convolution, inpainting).

- $\mathcal{N}(x; \mu, \Sigma)$ denotes the probability density function (PDF) of a Gaussian distribution with mean $\mu$ and covariance $\Sigma$ evaluated at $x$. The covariance matrix $\Sigma$ can be non-singular or singular (see Definition A.1).

- The $n$-th coordinate of a vector $x \in \mathbb{R}^N$ is denoted either $x_n$ or $x[n]$. Similarly, the coordinates of a multivalued function $\phi : \mathbb{R}^N \to \mathbb{R}^N$ can be denoted either $\phi_n(x)$ or $\phi(x)[n]$.

- The empirical data distribution $p_{\mathcal{D}}$ is defined by

$$p_{\mathcal{D}} = \frac{1}{|\mathcal{D}|} \sum_{x \in \mathcal{D}} \delta_x, \tag{9}$$

where $\mathcal{D}$ is the training dataset of finite size $|\mathcal{D}|$.

**Degenerate Gaussian distribution**   We can define the multivariate Gaussian distribution for positive semi-definite covariance matrices (Rao et al., 1973). Below, we recall a simple definition of the positive semi-definite covariance multivariate Gaussian, sometimes called the degenerate or singular multivariate Gaussian.

**Definition A.1** (Degenerate Gaussian distribution). A random vector $\boldsymbol{z} \in \mathbb{R}^N$ has a Gaussian distribution with mean $\mu \in \mathbb{R}^N$ and covariance matrix $\Sigma \in \mathbb{R}^{N \times N}, \Sigma \succeq 0$ if its probability density function (PDF) is given by:

$$\mathcal{N}(z; \mu, \Sigma) = \frac{1}{(2\pi)^{r/2}\sqrt{|\Sigma|_+}} \exp\left(-\frac{1}{2}(z - \mu)^\top \Sigma^+ (z - \mu)\right) \mathbb{1}_{\text{supp}(\boldsymbol{z})}(z), \tag{10}$$

on the support $z \in \text{supp}(\boldsymbol{z}) = \mu + \text{Im}(\Sigma)$ and zero elsewhere. In this equation, $r = \text{rank}(\Sigma)$, $\Sigma^+$ denotes the Moore-Penrose pseudo-inverse of $\Sigma$ and $|\Sigma|_+$ is the pseudo-determinant of $\Sigma$ (product of non-zero eigenvalues).

The support $\text{supp}(\boldsymbol{z})$ of the distribution is the $r-$ dimensional affine subspace of $\mathbb{R}^N$, $r$ is the rank of $\Sigma$. The density is with respect to the Lebesgue measure restricted to $\text{supp}(\boldsymbol{z})$, constructed via the pushforward measure of the standard Lebesgue measure on $\mathbb{R}^r$ by the affine map $v \mapsto \mu + \Sigma_r v$, where $\Sigma = \Sigma_r \Sigma_r^\top$. This measure also coincides (up to a normalization constant, equal to the pseudo-determinant of $\Sigma$) with the $r-$dimensional Hausdorff measure. For completeness, we provide a derivation of the density of $\boldsymbol{z}$ with respect to the $r$-dimensional Hausdorff measure $\mathcal{H}^r$ on the affine support $\text{supp}(\boldsymbol{z})$.

Suppose $\Sigma_r \in \mathbb{R}^{n \times r}$ satisfies $\Sigma = \Sigma_r \Sigma_r^\top$ and $\text{rank}(\Sigma) = r$. Define the random vector $\boldsymbol{z} = \mu + \Sigma_r \boldsymbol{w}$, where $\boldsymbol{w} \sim \mathcal{N}(0, I_r)$. Consider the affine map $\Phi : \mathbb{R}^r \to \mathbb{R}^N$ defined by $\Phi(w) = \mu + \Sigma_r w$. This map is a smooth bijection from $\mathbb{R}^r$ onto the support $\text{supp}(\boldsymbol{z}) = \mu + \text{Im}(\Sigma)$.

The probability measure of $\boldsymbol{z}$, denoted $\mathbb{P}_{\boldsymbol{z}}$, is the pushforward of the standard Gaussian measure $\gamma^r$ via $\Phi$. The standard Gaussian density on $\mathbb{R}^r$ is $g(w) = (2\pi)^{-r/2} e^{-\frac{1}{2}\|w\|^2}$. For any measurable set $A \subset \text{supp}(\boldsymbol{z})$, we have:

$$\mathbb{P}_{\boldsymbol{z}}(A) = \gamma^r(\Phi^{-1}(A)) = \int_{\Phi^{-1}(A)} g(w) \, d\lambda^r(w). \tag{11}$$

To find the density with respect to the Hausdorff measure $\mathcal{H}^r$ (the standard volume measure on the support), we apply the Area Formula (Federer, 2014) (change of variables). The Jacobian determinant of $\Phi$ is:

$$J\Phi = \sqrt{\det(\Sigma_r^\top \Sigma_r)} = \sqrt{|\Sigma|_+}.$$

Using the change of variables $z = \Phi(w)$, the volume elements relate via $d\mathcal{H}^r(z) = J\Phi\, d\lambda^r(w)$. Therefore:

$$d\lambda^r(w) = \frac{1}{\sqrt{|\Sigma|_+}} d\mathcal{H}^r(z).$$

Substituting this into the integral and using $w = \Phi^{-1}(z) = \Sigma_r^+(z - \mu)$:

$$\mathbb{P}_{\boldsymbol{z}}(A) = \int_A g(\Phi^{-1}(z)) \frac{1}{\sqrt{|\Sigma|_+}} d\mathcal{H}^r(z)$$

$$= \int_A \frac{1}{(2\pi)^{r/2}\sqrt{|\Sigma|_+}} \exp\left(-\frac{1}{2}\|\Sigma_r^+(z - \mu)\|^2\right) d\mathcal{H}^r(z).$$

Noting that $\|\Sigma_r^+(z - \mu)\|^2 = (z - \mu)^\top (\Sigma_r^+)^\top \Sigma_r^+ (z - \mu) = (z - \mu)^\top \Sigma^+ (z - \mu)$, we recover the density given in Definition A.1.

When the covariance matrix is non-singular, we recover the standard PDF of a Gaussian distribution.

**Degenerate Multivariate Gaussian and Mahalanobis Distance**  The Mahalanobis distance quantifies the distance from $z$ to $\mu$ relative to the covariance structure and is defined as:

$$D_\Sigma(z, \mu) = \sqrt{(z - \mu)^\top \Sigma^+ (z - \mu)}. \tag{12}$$

When $z \notin \mathrm{supp}\,(\boldsymbol{z})$, the density is zero, corresponding to an infinite effective distance outside the support.

- Eigenvalue Decomposition. Consider the spectral decomposition of the covariance matrix:

$$\Sigma = U\Lambda U^\top,$$

  where:

  - $U$ is an orthogonal matrix whose columns are the eigenvectors of $\Sigma$.
  - $\Lambda = \mathrm{diag}(\lambda_1, \lambda_2, \ldots, \lambda_d)$ is a diagonal matrix with eigenvalues $\lambda_1 \geq \lambda_2 \geq \cdots \geq \lambda_r > 0 = \lambda_{r+1} = \cdots = \lambda_d$, sorted in descending order, with zeros corresponding to the degenerate directions.

  The pseudo-inverse is:

$$\Sigma^+ = U\Lambda^+ U^\top, \tag{13}$$

  where $\Lambda^+ = \mathrm{diag}(1/\lambda_1, \ldots, 1/\lambda_r, 0, \ldots, 0)$.

  Transform the coordinates to the eigen-basis by defining $y = U^\top(x - \mu)$. The Mahalanobis distance simplifies to:

$$D_\Sigma(x, \mu) = \sqrt{y^\top \Lambda^+ y} = \sqrt{\sum_{i=1}^r \frac{y_i^2}{\lambda_i}}, \tag{14}$$

  since $y_i = 0$ for $i > r$ on the support $\mathrm{supp}\,(\boldsymbol{z})$. The exponent in the density becomes:

$$-\frac{1}{2}\sum_{i=1}^r \frac{y_i^2}{\lambda_i}. \tag{15}$$

- Distance in Different Directions

  The effect of deviations from the mean $\mu$ along the directions of the eigenvectors (principal axes) depends on the corresponding eigenvalues:

  - **Directions with large eigenvalues** ($\lambda_i \gg 0$): The term $\frac{y_i^2}{\lambda_i}$ grows slowly as $|y_i|$ increases. Deviations along these high-variance directions contribute less to reducing the density, effectively scaling the distance by $\sqrt{\lambda_i}$. This allows larger Euclidean deviations in these directions.

- **Directions with small positive eigenvalues** ($0 < \lambda_i \ll 1$): The term $\frac{y_i^2}{\lambda_i}$ grows rapidly even for small $|y_i|$. Deviations are heavily penalized, scaling the distance by $1/\sqrt{\lambda_i}$, making small deviations appear "far" in the Mahalanobis sense.

- **Directions with zero eigenvalues** ($\lambda_i = 0$): These correspond to the null space of $\Sigma$. Here, $y_i$ must be exactly zero for the density to be non-zero, enforced by the Dirac delta measure. Any non-zero deviation results in zero density, equivalent to an infinite Mahalanobis distance, indicating no variability in these directions.

**Integration on linear subspaces**    The following lemma is useful for change of variable with linear mapping (Federer, 2014).

**Lemma A.2** (Integration on linear subspaces)*. Let $\mathcal{N}(y; \mu, \Sigma)$ be the PDF of a Gaussian distribution in $\mathbb{R}^N$, with mean $\mu \in \mathbb{R}^N$ and covariance matrix $\Sigma \in \mathbb{R}^{N \times N}$ of rank $r_0 \leq N$. For any linear application $B : \mathbb{R}^N \to \mathbb{R}^P$, and any measurable $f$, we have:*

$$\int_{\mathbb{R}^N} f(By)\mathcal{N}(y; \mu, \Sigma)\, d\mathcal{H}^{r_0}(y) = \int_{\mathbb{R}^P} f(v)\mathcal{N}\left(v; B\mu, B\Sigma B^\top\right) d\mathcal{H}^r(v), \tag{16}$$

*where $\mathcal{N}\left(v; B\mu, B\Sigma B^\top\right)$ is the PDF of a (possibly degenerate) Gaussian distribution with respect to the $r-$dimensional Hausdorff measure $\mathcal{H}^r$, with $r$ being the rank of $B\Sigma B^\top$. This PDF is supported on the $r-$dimensional subspace $B\mu + \mathrm{Im}(B\Sigma B^\top) \subset \mathrm{Im}(B) \subset \mathbb{R}^P$.*

*Proof.* The proof of this result is relatively straightforward by using the density of a degenerate Gaussian distribution Definition A.1. Let $\boldsymbol{y} \sim \mathcal{N}(\mu, \Sigma)$ and $\boldsymbol{z} = B\boldsymbol{y}$, then $\boldsymbol{z} \sim \mathcal{N}\left(B\mu, B\Sigma B^\top\right)$. We have

$$\int_{\mathbb{R}^N} f(By)\mathcal{N}(y; \mu, \Sigma)\, d\mathcal{H}^{r_0}(y) = \mathbb{E}\left[f(B\boldsymbol{y})\right] \tag{17}$$

$$= \mathbb{E}\left[f(\boldsymbol{z})\right] \tag{18}$$

$$= \int_{\mathbb{R}^P} f(v)\mathcal{N}\left(v; B\mu, B\Sigma B^\top\right) d\mathcal{H}^r(v). \tag{19}$$

$\square$

*Remark* A.3.  When $\Sigma$ is full rank, the Hausdorff measure $\mathcal{H}^{r_0}$ coincides with the Lebesgue measure on $\mathbb{R}^N$.

## A.2. Minimum Mean Square Error (MMSE) estimator

The MMSE estimator is the optimal estimator in the following sense

**Definition A.4** (MMSE estimator)**.  Given two random vectors $\boldsymbol{x} \in \mathbb{R}^N, \boldsymbol{y} \in \mathbb{R}^M$ and a linear operator $B : \mathbb{R}^M \to \mathbb{R}^N$. The MMSE estimator of $\boldsymbol{x}$ given $B\boldsymbol{y}$ is the best approximation *random variable* $\phi^\star(B\boldsymbol{y})$ to $\boldsymbol{x}$, in the least-square sense:

$$\hat{x}_{\mathrm{MMSE}} = \phi^\star \circ B \quad \text{where} \quad \phi^\star = \underset{\phi:\mathbb{R}^N \to \mathbb{R}^N}{\arg\min}\, \mathbb{E}\left[\|\phi(B\boldsymbol{y}) - \boldsymbol{x}\|^2\right] \tag{20}$$

It is well-known that the MMSE estimator coincides with the *conditional expectation*. That is, for any $y$ such that $p(y) > 0$, we have:

$$\hat{x}_{\mathrm{MMSE}}(y) = \mathbb{E}\left[\boldsymbol{x}|B\boldsymbol{y} = By\right] = \int x \cdot p(x|By)dx \tag{21}$$

Composing $\hat{x}_{\mathrm{MMSE}}$ with the random variable $\boldsymbol{y}$, we get

$$\hat{x}_{\mathrm{MMSE}}(\boldsymbol{y}) = \mathbb{E}\left[\boldsymbol{x}|B\boldsymbol{y}\right].$$

In particular, if $B = \mathrm{I}$ (in this case $M = N$), the MMSE estimator reduces to the classical *posterior mean*: $\hat{x}_{\mathrm{MMSE}}(y) = \mathbb{E}\left[\boldsymbol{x}|\boldsymbol{y} = y\right]$. Note that both $\mathbb{E}\left[\boldsymbol{x}|\boldsymbol{y} = y\right]$ and $\mathbb{E}\left[\boldsymbol{x}|\boldsymbol{y}\right]$ are often called condition expectation, but these are different objects. In particular, $\mathbb{E}\left[\boldsymbol{x}|\boldsymbol{y} = \cdot\right] = \hat{x}_{\mathrm{MMSE}}(\cdot)$ is a function $\mathbb{R}^M \to \mathbb{R}^N$ while $\mathbb{E}\left[\boldsymbol{x}|\boldsymbol{y}\right]$ is a random variable assuming values in $\mathbb{R}^N$. However, finding the MMSE estimator amounts to finding the optimal function $\phi^\star$.

### A.3. Constrained Minimum Mean Square Error (MMSE) estimator

The classical MMSE estimator is defined as the best approximation function over the space of measurable function from $\mathbb{R}^M$ to $\mathbb{R}^N$. When adding functional constraints to the estimator, we would like to find the best approximation function over a subspace $\mathcal{M}$.

$$\min_{\phi \in \mathcal{M}} \mathbb{E}\left[\|\phi(B\boldsymbol{y}) - \boldsymbol{x}\|^2\right] \tag{22}$$

For example, the subspace $\mathcal{M}$ could be the set of measurable functions from $\mathbb{R}^N \to \mathbb{R}^N$ and translation equivariant.

### A.4. Optimality condition

We first state a simple first-order sufficient and necessary optimality condition for solving the Constrained MMSE.

**Proposition A.5** (Optimality condition). *Let $\boldsymbol{x} \in \mathbb{R}^N, \boldsymbol{y} \in \mathbb{R}^M$ be two random variables and $\mathcal{M}$ be a vector space of measurable functions from $\mathbb{R}^N$ to $\mathbb{R}^N$, $B$ be a linear operator from $\mathbb{R}^M$ to $\mathbb{R}^N$. Then $\phi^\star$ is a minimizer of*

$$\min_{\phi \in \mathcal{M}} \mathbb{E}\left[\|\phi(B\boldsymbol{y}) - \boldsymbol{x}\|^2\right]$$

*if and only if*

$$\mathbb{E}\left[\langle \varphi(B\boldsymbol{y}), \phi^\star(B\boldsymbol{y}) - \boldsymbol{x}\rangle\right] = 0 \qquad \text{for all } \varphi \in \mathcal{M}. \tag{23}$$

*Proof.* Let $J(\phi) = \mathbb{E}\left[\|\phi(B\boldsymbol{y}) - \boldsymbol{x}\|^2\right]$. For all $t \in \mathbb{R}$ and for all $\varphi \in \mathcal{M}$, we have

$$
\begin{aligned}
J(\phi^\star + t\varphi) &= \mathbb{E}\left[\|(\phi^\star + t\varphi)(B\boldsymbol{y}) - \boldsymbol{x}\|^2\right] \\
&= J(\phi^\star) + 2t\underbrace{\mathbb{E}\left[\langle \varphi(B\boldsymbol{y}), \phi^\star(B\boldsymbol{y}) - \boldsymbol{x}\rangle\right]}_{a} + t^2\underbrace{\mathbb{E}\left[\|\varphi(B\boldsymbol{y})\|^2\right]}_{b} \\
&= J(\phi^\star) + 2at + bt^2
\end{aligned}
$$

Therefore, $\phi^\star$ is a minimizer of $J$ on $\mathcal{M}$ if and only if $J(\phi^\star + t\varphi) \geq J(\phi^\star)$ for all $t \in \mathbb{R}$ and all $\varphi \in \mathcal{M}$. This is equivalent to the condition that $2at + bt^2 \geq 0$, for all $t \in \mathbb{R}$ and all $\varphi \in \mathcal{M}$. Since this difference term is a quadratic function in $t$ and $b \geq 0$, it is non-negative for all $t \in \mathbb{R}$ if and only if $a = 0$. Therefore, we have the sufficient and necessary condition that $\mathbb{E}\left[\langle \varphi(B\boldsymbol{y}), \phi^\star(B\boldsymbol{y}) - \boldsymbol{x}\rangle\right] = 0$ for all $\varphi \in \mathcal{M}$. $\qquad\square$

The optimality condition Proposition A.5 states that the residual $\phi^\star(B\boldsymbol{y}) - \boldsymbol{x}$ is orthogonal, in the $L^2$ sense, to every perturbation in the feasible set $\mathcal{M}$. Equivalently, $\phi^\star(B\boldsymbol{y})$ is the orthogonal projection of $\boldsymbol{x}$ onto $\mathcal{M}$ in the $L^2$ sense. When $\mathcal{M}$ is the space of all square-integrable functions of $\boldsymbol{y}$ (*i.e.* no constraints) and $B = \mathrm{I}$, this projection yields the classical MMSE estimator (posterior mean), $\mathbb{E}[\boldsymbol{x}|\boldsymbol{y}]$. In the constrained case, $\phi^\star(B\boldsymbol{y})$ can also be viewed as the orthogonal projection of the conditional expectation $\mathbb{E}[\boldsymbol{x}|\boldsymbol{y}]$ onto the subspace $\{\phi(B\boldsymbol{y}) : \phi \in \mathcal{M}\}$.

**Proposition A.6.** *[Proposition 3.3 in the main paper] Given a closed set $\mathcal{M}$. The $\mathcal{M}$-constrained MMSE estimator in Definition 1.1 is the orthogonal projection (in $L^2$ sense) of the posterior mean $\mathbb{E}[\boldsymbol{x}|\boldsymbol{y}]$ onto the subspace of $\boldsymbol{y}$-measurable random vectors of form $\mathcal{X} = \{\phi(B\boldsymbol{y}) : \phi \in \mathcal{M}\}$. That is, $\hat{x}_\mathcal{M}(\boldsymbol{y}) = \Pi_\mathcal{X} \mathbb{E}[\boldsymbol{x}|\boldsymbol{y}]$.*

*Proof.* The classes $\mathcal{M}_\mathcal{T}$ and $\mathcal{M}_{\mathcal{T},\mathrm{loc}}$ are linear subspaces of the space of measurable functions from $\mathbb{R}^N$ to $\mathbb{R}^N$: they are closed under addition and scalar multiplication, hence the projection is well-defined. The MMSE estimator in Proposition 2.1 is the posterior mean $\mathbb{E}[\boldsymbol{x}|\boldsymbol{y}]$, which is the orthogonal projection of $\boldsymbol{x}$ onto the space of $\boldsymbol{y}$-measurable random vectors. Similarly, the $\mathcal{M}-$constrained MMSE estimator is the projection of $\boldsymbol{x}$ on to the subspace $\{\phi(B\boldsymbol{y}) : \phi \in \mathcal{M}\}$. Using the Pythagorean decomposition, for any $\phi \in \mathcal{M}$, we have

$$\mathbb{E}\left[\|\phi(B\boldsymbol{y}) - \boldsymbol{x}\|^2\right] = \mathbb{E}\left[\|\phi(B\boldsymbol{y}) - \mathbb{E}[\boldsymbol{x}|\boldsymbol{y}]\|^2\right] + \mathbb{E}\left[\|\mathbb{E}[\boldsymbol{x}|\boldsymbol{y}] - \boldsymbol{x}\|^2\right] \tag{24}$$

Therefore

$$\arg\min_{\phi \in \mathcal{M}} \mathbb{E}\left[\|\phi(B\boldsymbol{y}) - \boldsymbol{x}\|^2\right] = \arg\min_{\phi \in \mathcal{M}} \mathbb{E}\left[\|\phi(B\boldsymbol{y}) - \mathbb{E}[\boldsymbol{x}|\boldsymbol{y}]\|^2\right] \tag{25}$$

and the $\mathcal{M}-$constrained MMSE in Definition 1.1 is the projection of the posterior mean $\mathbb{E}\left[\boldsymbol{x}|\boldsymbol{y}\right]$ onto the subspace $\{\phi(B\boldsymbol{y}) : \phi \in \mathcal{M}\}$.

Moreover, the decomposition in Equation (24) has interesting interpretation: it isolates the sources of error in the estimation. The first term $\mathbb{E}\left[\|\phi(B\boldsymbol{y}) - \mathbb{E}\left[\boldsymbol{x}|\boldsymbol{y}\right]\|^2\right]$ is the approximation error due to the restriction to the subspace $\mathcal{M}$, while the second term $\mathbb{E}\left[\|\mathbb{E}\left[\boldsymbol{x}|\boldsymbol{y}\right] - \boldsymbol{x}\|^2\right]$ is the irreducible Bayes error.                               $\square$

### A.5. Structural constraint: equivariant functions

The below definitions are taken from (Celledoni et al., 2021).

**Definition A.7** (Group). A *group*, to be denoted $\mathcal{G}$, is a set equipped with an associative operator $\cdot : \mathcal{G} \times \mathcal{G} \to \mathcal{G}$, which satisfies the following conditions:

1. If $g_1, g_2 \in \mathcal{G}$ then $g_2 \cdot g_1 \in \mathcal{G}$

2. If $g_1, g_2, g_3 \in \mathcal{G}$ then $(g_1 \cdot g_2) \cdot g_3 = g_1 \cdot (g_2 \cdot g_3)$

3. There exists $\iota \in \mathcal{G}$ such that $e \cdot g = g \cdot \iota = g$ for all $g \in \mathcal{G}$.

4. If $g \in \mathcal{G}$ there exists $g^{-1} \in \mathcal{G}$ such that $g^{-1} \cdot g = g \cdot g^{-1} = \iota$.

**Definition A.8** (Group action). Given a group $\mathcal{G}$ and a set $\mathbb{R}^N \subset \mathbb{R}^N$, we say that $\mathcal{G}$ acts on $\mathbb{R}^N$ if there exists a function $T : \mathcal{G} \times \mathbb{R}^N \to \mathbb{R}^N$ (we denote by $T_g(x)$ for $g \in \mathcal{G}$ and $x \in \mathbb{R}^N$) that satisfies:

$$T_{g_1} \circ T_{g_2} = T_{g_1 \cdot g_2} \qquad \text{and} \qquad T_\iota = \text{id}$$

Given a general group $\mathcal{G}$. A function $\phi : \mathbb{R}^N \to \mathbb{R}^N$ and group action $T$ of $\mathcal{G}$ on $\mathbb{R}^N$. A function $\phi$ is called $\mathcal{G}$-*equivariant* if it satisfies

$$\phi(T_g(y)) = T_g \phi(y) \qquad \text{for all } x \in \mathbb{R}^N \text{ and for all } g \in \mathcal{G}.$$

**Proposition A.9.** *If the group $\mathcal{G}$ is finite, the following properties hold true:*

1. ***Invariance**: for any function $\phi : \mathbb{R}^N \to \mathbb{R}$, the function $\bar{\phi} = \sum_g \phi \circ T_g$ is invariant. (And similarly for function defined in $\mathbb{R}^M$).*

2. ***Equivariance**: for any function $\phi : \mathbb{R}^N \to \mathbb{R}^N$, the function $\bar{\phi} = \sum_g T_g^{-1} \circ \phi \circ T_g$ is equivariant. This is called Reynolds averaging.*

For image data $x \in \mathbb{R}^N$, let $H, W \in \mathbb{N}$ be the dimensions of a discrete grid $\Omega = \mathbb{Z}_H \times \mathbb{Z}_W$ with $H \times W = N$.

**Definition A.10** (Translation equivariant functions). Let $\mathcal{T} = \mathbb{Z}_H \times \mathbb{Z}_W$ be the group of 2D cyclic translations. For every group element $g = (g_h, g_v) \in \mathcal{T}$, we define the *translation operator* $T_g : \mathbb{R}^N \to \mathbb{R}^N$ as the permutation matrix that acts on an image $x \in \mathbb{R}^N$ by shifting its indices:

$$(T_g x)[i, j] = x[(i - g_h) \bmod H, (j - g_v) \bmod W]$$

for all $(i, j) \in \Omega$. A measurable map $\phi : \mathbb{R}^N \to \mathbb{R}^N$ is said to be *translation equivariant* if it commutes with the translation operator for all $g \in \mathcal{T}$:

$$\phi(T_g x) = T_g \phi(x) \qquad \text{for all } x \in \mathbb{R}^N.$$

### A.6. Structural constraints: local and translation equivariant functions

We first define precisely the patch extractor. Let $n \in \Omega$ be a pixel coordinate on the grid. We define the *patch extractor* $\Pi_n : x \in \mathbb{R}^N \to \Pi_n x = x[\omega_n] \in \mathbb{R}^P$ which extracts a square patch $x[\omega_n]$ of size $\sqrt{P} \times \sqrt{P}$ centered at $n$. The extraction uses circular boundary conditions, such that the patch is given by the grid values at indices:

$$\omega_n = \{n + \delta \quad (\bmod (H, W)) \mid \delta \in \Delta\}$$

where $\Delta$ is the set of offsets defining the square neighborhood centered at zero.

**Definition A.11** (Local and translation-equivariant functions). A measurable map $\phi : \mathbb{R}^N \to \mathbb{R}^N$ is said to be *local* if it can be represented as a sliding window operation. Specifically, if there exist a measurable function $f : \mathbb{R}^P \to \mathbb{R}$ such that for all $x \in \mathbb{R}^N$ and all $n \in \Omega$:

$$\phi(x)[n] = f(\Pi_n x).$$

By construction, any such function $\phi$ is also translation equivariant due to the circular boundary conditions of the patch extractor $\Pi_n$. We denote by $\mathcal{M}_{\mathcal{T},\text{loc}}$ the set of all such maps.

This class captures standard CNNs with finite kernels and weight sharing or patch-based methods, *e.g.* MLPs acting on patches (Khorashadizadeh et al., 2025): the output of $\phi$ at pixel $n$, denoted $\phi(x)[n]$ or $\phi_n(x)$, depends only on the local patch (receptive field) $x[\omega_n]$. Note that by construction, functions in Definition A.11 are translation equivariant.

## B. Analytical solution for inverse problems

In the following, we will derive the analytical solution for the MMSE estimator under these constraints.

Consider a random variable $\boldsymbol{x} \sim p_{\boldsymbol{x}}, \boldsymbol{x} \in \mathbb{R}^N$ and the forward (measurement) model

$$\boldsymbol{y} = A\boldsymbol{x} + \boldsymbol{e} \qquad \text{where} \qquad \boldsymbol{e} \sim \mathcal{N}\left(0, \sigma^2 \mathrm{I}\right). \tag{26}$$

We would like to find the MMSE estimator of $\boldsymbol{x}$ given $\boldsymbol{y}$, with or without constraints. In this section, we will derive the analytical solution for the MMSE estimator under various constraints, when the true underlying distribution of $\boldsymbol{x}$ is replaced by the empirical distribution $p_{\mathcal{D}}$. Under Gaussian noise, the likelihood of the measurement $\boldsymbol{y}$ given $\boldsymbol{x}$ is given by

$$p(y|x) = \mathcal{N}\left(y; Ax, \sigma^2 \mathrm{I}\right) \propto \exp\left(-\frac{\|y - Ax\|^2}{2\sigma^2}\right). \tag{27}$$

Therefore, for any function $\phi^\star$ and $\varphi$, we have

$$\begin{aligned}
\mathbb{E}\left[\langle \varphi(B\boldsymbol{y}), \phi^\star(B\boldsymbol{y}) - \boldsymbol{x}\rangle\right] &= \mathbb{E}_{\boldsymbol{x} \sim p_{\mathcal{D}}} \mathbb{E}_{\boldsymbol{y}|\boldsymbol{x}} \left[\langle \varphi(B\boldsymbol{y}), \phi^\star(B\boldsymbol{y}) - \boldsymbol{x}\rangle\right] \\
&= \frac{1}{|\mathcal{D}|} \sum_{x \in \mathcal{D}} \int_{\mathbb{R}^M} \langle \varphi(By), \phi^\star(By) - x\rangle \, p(y|x) dy \\
&= \frac{1}{|\mathcal{D}|} \sum_{x \in \mathcal{D}} \int_{\mathbb{R}^M} \langle \varphi(By), \phi^\star(By) - x\rangle \, \mathcal{N}\left(y; Ax, \sigma^2 \mathrm{I}\right) dy
\end{aligned} \tag{28}$$

This simplified expression (28) is useful for deriving the analytical solution of the MMSE estimator under various constraints.

### B.1. Unconstrained MMSE estimator

We start with the unconstrained MMSE estimator, which is the optimal estimator in the least-square sense. Then, we will derive the equivariant MMSE estimator, which is the optimal estimator under the constraint of equivariance to translation. Then, we will derive the local MMSE estimator, which is the optimal estimator under the constraint of locality. Finally, we will also show that combining both constraints leads to a local and equivariant MMSE estimator.

**Proposition B.1** (Unconstrained MMSE estimator). *When the data distribution is replaced by the empirical distribution $p_{\mathcal{D}}$, the unconstrained MMSE estimator of $\boldsymbol{x}$ given $\boldsymbol{y}$ is given by*

$$\hat{x}_{MMSE}(y) = \phi^\star(By) \quad \text{where} \quad \phi^\star(v) = \frac{\sum_{x \in \mathcal{D}} x \cdot \mathcal{N}\left(v; BAx, \sigma^2 BB^\top\right)}{\sum_{x \in \mathcal{D}} \mathcal{N}\left(v; BAx, \sigma^2 BB^\top\right)} \quad \text{for any } v \in \text{Im}(B). \tag{29}$$

*The estimator is well-defined for all $y \in \mathbb{R}^M$.*

*Proof.* We will verify that $\phi^*$ satisfies the optimality condition from Proposition A.5. Using Equation (28), for any $\varphi$, we

have:

$$
\begin{aligned}
\mathbb{E}\left[\langle\varphi(B\boldsymbol{y}),\phi^*(B\boldsymbol{y})-\boldsymbol{x}\rangle\right] &= \frac{1}{|\mathcal{D}|}\sum_{x\in\mathcal{D}}\int_{\mathbb{R}^M}\langle\varphi(By),\phi^*(By)-x\rangle\mathcal{N}\left(y,Ax,\sigma^2\mathrm{I}_M\right)dy \\
&= \frac{1}{|\mathcal{D}|}\sum_{x\in\mathcal{D}}\int_{\mathbb{R}^N}\langle\varphi(v),\phi^*(v)-x\rangle\mathcal{N}\left(v,BAx,\sigma^2BB^\top\right)d\mathcal{H}^r(v) &\text{(30)}\\
&= \frac{1}{|\mathcal{D}|}\int_{\mathbb{R}^N}\left\langle\varphi(v),\phi^*(v)\sum_{x\in\mathcal{D}}\mathcal{N}\left(v,BAx,\sigma^2BB^\top\right)-x\sum_{x\in\mathcal{D}}\mathcal{N}\left(v,BAx,\sigma^2BB^\top\right)\right\rangle d\mathcal{H}^r(v)\\
&= 0
\end{aligned}
$$

where Equation (30) comes from the change of variable $v = By$ and Lemma A.2. The last equality holds by definition of $\phi^*$. $\qquad\square$

## B.2. Equivariant MMSE estimator

**Proposition B.2** (Translation equivariant MMSE estimator)**.** *The translation equivariant MMSE estimator is as follows, for any* $y \in \mathbb{R}^M$*:*

$$
\hat{x}_{\mathcal{T}}(y) = \phi^\star(By) \qquad where \qquad \phi^\star(v) = \frac{\sum_{x\in\mathcal{D},g\in\mathcal{T}}T_g x\cdot\mathcal{N}\left(T_g^{-1}v;BAx,\sigma^2BB^\top\right)}{\sum_{x\in\mathcal{D},g\in\mathcal{T}}\mathcal{N}\left(T_g^{-1}v;BAx,\sigma^2BB^\top\right)} \tag{31}
$$

*for any* $v \in \bigcup_g T_g\mathrm{Im}(B)\subset\mathbb{R}^N$.

*Proof.* We will verify that $\phi^\star$ is admissible and satisfies the optimality condition Proposition A.5.

**Admissibility.** Let $q(v,x) = \mathcal{N}\left(v;BAx,\sigma^2BB^\top\right)$ denote the PDF of a Gaussian distribution with mean $BAx$ and covariance $\sigma^2BB^\top$. The optimal estimator can be written as:

$$
\phi^\star(v) = \frac{\sum_{x\in\mathcal{D},g\in\mathcal{T}}T_g x q(T_g^{-1}v,x)}{\sum_{x\in\mathcal{D},g\in\mathcal{T}}q(T_g^{-1}v,x)} \tag{32}
$$

The denominator $\sum_{x\in\mathcal{D},g\in\mathcal{T}}q(T_g^{-1}v,x) = \sum_{g\in\mathcal{T}}\bar{q}_1(T_g^{-1}v)$ is $\mathcal{T}-$invariant by Proposition A.9, where $\bar{q}_1(v) = \sum_{x\in\mathcal{D}}q(v,x)$.

The nominator $\sum_{x\in\mathcal{D},g\in\mathcal{T}}T_g x\cdot q(T_g^{-1}v,x) = \sum_{g\in\mathcal{T}}T_g\bar{q}_2(T_g^{-1}v)$ is $\mathcal{T}-$equivariant, where $\bar{q}_2(v) = \sum_{x\in\mathcal{D}}x\cdot q(v,x)$. Therefore, $\phi^\star$ is admissible, that is $\phi^\star\in\mathcal{M}_\mathcal{T}$.

**Optimality condition.** By using Equation (28), for any $\varphi\in\mathcal{M}_\mathcal{T}$, we have:

$$
\begin{aligned}
&\mathbb{E}\left[\langle\varphi(B\boldsymbol{y}),\phi^\star(B\boldsymbol{y})-\boldsymbol{x}\rangle\right]\\
&= \frac{1}{|\mathcal{D}|}\sum_{x\in\mathcal{D}}\int_{\mathbb{R}^M}\langle\varphi(By),\phi^\star(By)-x\rangle\mathcal{N}\left(y;Ax,\sigma^2\mathrm{I}_N\right)dy\\
&= \frac{1}{|\mathcal{D}|}\sum_{x\in\mathcal{D}}\int_{\mathbb{R}^N}\langle\varphi(v),\phi^\star(v)-x\rangle\mathcal{N}\left(v;BAx,\sigma^2BB^\top\right)d\mathcal{H}^r(v) &\text{(33)}\\
&= \frac{1}{|\mathcal{D}||\mathcal{T}|}\sum_{g\in\mathcal{T},x\in\mathcal{D}}\int_{\mathbb{R}^N}\langle T_g^{-1}\varphi(T_g v),T_g^{-1}\phi^\star(T_g v)-x\rangle\mathcal{N}\left(v;BAx,\sigma^2BB^\top\right)d\mathcal{H}^r(v) &\text{(34)}\\
&= \frac{1}{|\mathcal{D}||\mathcal{T}|}\sum_{g\in\mathcal{T},x\in\mathcal{D}}\int_{\mathbb{R}^N}\langle\varphi(T_g v),\phi^\star(T_g v)-T_g x\rangle\mathcal{N}\left(v;BAx,\sigma^2BB^\top\right)d\mathcal{H}^r(v)\\
&= \frac{1}{|\mathcal{D}||\mathcal{T}|}\sum_{g\in\mathcal{T},x\in\mathcal{D}}\int_{\mathbb{R}^N}\langle\varphi(v),\phi^\star(v)-T_g x\rangle\mathcal{N}\left(T_g^{-1}v;BAx,\sigma^2BB^\top\right)d\mathcal{H}^r(v) &\text{(35)}\\
&= \frac{1}{|\mathcal{D}||\mathcal{T}|}\int_{\mathbb{R}^N}\left\langle\varphi(v),\phi^\star(v)\sum_{g\in\mathcal{T},x\in\mathcal{D}}q(T_g^{-1}v,n,x)-\sum_{g\in\mathcal{T},x\in\mathcal{D}}T_g x\cdot q(T_g^{-1}v,n,x)\right\rangle d\mathcal{H}^r(v)
\end{aligned}
$$

$$= 0$$

where $q(T_g^{-1}v, n, x) \overset{\text{def}}{=} \mathcal{N}\left(T_g^{-1}v; BAx, \sigma^2 BB^\top\right)$ and

- In Equation (33), we apply Lemma A.2 for $B$ in the change of variables.

- In Equation (34), we use the fact that $\varphi = \varphi \circ T_g \circ T_g^{-1} = T_g \circ \varphi \circ T_g^{-1}$ for all $g \in \mathcal{T}$ and $\varphi \in \mathcal{M}_\mathcal{T}$. Moreover, $T_g^{-1} = T_g^\top$ for all $g \in \mathcal{T}$.

- In Equation (35), we use the change of variable $T_g v \to v$ and the fact that $T_g$ is an isometry.

$\square$

We next state and prove that the augmented MMSE estimator is reconstruction equivariant.

**Lemma B.3** (Reconstruction equivariance of $\hat{x}_{\text{MMSE}}^{aug}$). *Let $A : \mathbb{R}^N \to \mathbb{R}^M$ be a circular convolution operator, i.e., $AT_g = T_g A$ for all $g \in \mathcal{T}$. Then, the data-augmented MMSE estimator $\hat{x}_{\text{MMSE}}^{aug}$ defined in Definition 3.4 satisfies the reconstruction equivariance property in Equation (5):*

$$\hat{x}_{\text{MMSE}}^{aug}(AT_g\bar{x} + e) = T_g \hat{x}_{\text{MMSE}}^{aug}(A\bar{x} + T_g^{-1}e)$$

*for all $\bar{x} \in \mathbb{R}^N$, all $e \in \mathbb{R}^M$, and all $g \in \mathcal{T}$.*

*Proof.* Let $y = AT_g\bar{x} + e$ be the input observation. Recall the definition of the data-augmented MMSE estimator in Definition 3.4:

$$\hat{x}_{\text{MMSE}}^{aug}(y) = \frac{\sum_{x \in \mathcal{T}(\mathcal{D})} x \cdot \exp\left(-\frac{1}{2\sigma^2}\|y - Ax\|^2\right)}{\sum_{x \in \mathcal{T}(\mathcal{D})} \exp\left(-\frac{1}{2\sigma^2}\|y - Ax\|^2\right)}.$$

Substituting the input $y = AT_g\bar{x} + e$ into the estimator:

$$\hat{x}_{\text{MMSE}}^{aug}(y) = \frac{\sum_{x \in \mathcal{T}(\mathcal{D})} x \cdot \exp\left(-\frac{1}{2\sigma^2}\|AT_g\bar{x} + e - Ax\|^2\right)}{\sum_{x \in \mathcal{T}(\mathcal{D})} \exp\left(-\frac{1}{2\sigma^2}\|AT_g\bar{x} + e - Ax\|^2\right)}.$$

Since $\mathcal{T}(\mathcal{D})$ is invariant under the group action, we can perform a change of variable $z = T_g^{-1}x$. As $x$ traverses $\mathcal{T}(\mathcal{D})$, $z$ also traverses $\mathcal{T}(\mathcal{D})$. Note that $x = T_g z$. Substituting $x$ with $T_g z$ in the numerator and denominator:

$$\hat{x}_{\text{MMSE}}^{aug}(y) = \frac{\sum_{z \in \mathcal{T}(\mathcal{D})} T_g z \cdot \exp\left(-\frac{1}{2\sigma^2}\|AT_g\bar{x} + e - AT_g z\|^2\right)}{\sum_{z \in \mathcal{T}(\mathcal{D})} \exp\left(-\frac{1}{2\sigma^2}\|AT_g\bar{x} + e - AT_g z\|^2\right)}.$$

Using the assumption that $A$ is a circular convolution ($AT_g = T_g A$) and $T_g$ is linear, the term inside the norm becomes:

$$AT_g\bar{x} + e - AT_g z = T_g A\bar{x} + T_g(T_g^{-1}e) - T_g Az$$
$$= T_g\left(A\bar{x} + T_g^{-1}e - Az\right).$$

Since $T_g$ is unitary, it preserves the norm, i.e., $\|T_g u\| = \|u\|$. Therefore:

$$\|AT_g\bar{x} + e - AT_g z\|^2 = \|A\bar{x} + T_g^{-1}e - Az\|^2.$$

Substituting this back into the estimator expression and factoring the linear operator $T_g$ out of the sum in the numerator:

$$\hat{x}_{\text{MMSE}}^{aug}(y) = \frac{T_g \sum_{z \in \mathcal{T}(\mathcal{D})} z \cdot \exp\left(-\frac{1}{2\sigma^2}\left\|(A\bar{x} + T_g^{-1}e) - Az\right\|^2\right)}{\sum_{z \in \mathcal{T}(\mathcal{D})} \exp\left(-\frac{1}{2\sigma^2}\left\|(A\bar{x} + T_g^{-1}e) - Az\right\|^2\right)}$$
$$= T_g\left(\frac{\sum_{z \in \mathcal{T}(\mathcal{D})} z \cdot \mathcal{N}\left(A\bar{x} + T_g^{-1}e; Az, \sigma^2 I\right)}{\sum_{z' \in \mathcal{T}(\mathcal{D})} \mathcal{N}\left(A\bar{x} + T_g^{-1}e; Az', \sigma^2 I\right)}\right).$$

The term in the parentheses is exactly the estimator evaluated at the input $A\bar{x} + T_g^{-1}e$. Thus:

$$\hat{x}_{\text{MMSE}}^{\text{aug}}(AT_g\bar{x} + e) = T_g\hat{x}_{\text{MMSE}}^{\text{aug}}(A\bar{x} + T_g^{-1}e).$$

$\square$

Now we state and prove the properties of the E-MMSE estimator presented in Corollary 3.7.

*Proof of Corollary 3.7.* The first point is a direct consequence of Theorem 3.5. We prove the second and third point as follows. We first express the weights of both estimators. The two estimators admit the same form:

$$\hat{x}_{\mathcal{T}}(y) = \sum_{x \in \mathcal{D}, g \in \mathcal{T}} T_g x \cdot w_g(x|y) \qquad \text{and} \qquad \hat{x}_{\text{MMSE}}^{\text{aug}}(y) = \sum_{x \in \mathcal{D}, g \in \mathcal{T}} T_g x \cdot w_g^{\text{aug}}(x|y).$$

The weights of the E-MMSE estimator $\hat{x}_{\mathcal{T}}$ corresponding to the component $(x, g)$ are given by:

$$
\begin{aligned}
w_g(x|y) &= \frac{\mathcal{N}\left(T_g^{-1}By; BAx, \sigma^2 BB^\top\right)}{\sum_{x' \in \mathcal{D}, g' \in \mathcal{T}} \mathcal{N}\left(T_{g'}^{-1}By; BAx', \sigma^2 BB^\top\right)} \\
&= \frac{\exp\left(-\left\|B^{-1}(T_g^{-1}By - BAx)\right\|^2/(2\sigma^2)\right)}{\sum_{x' \in \mathcal{D}, g' \in \mathcal{T}} \exp\left(-\left\|B^{-1}(T_{g'}^{-1}By - BAx')\right\|^2/(2\sigma^2)\right)} \\
&= \frac{\exp\left(-\left\|B^{-1}T_g^{-1}By - Ax\right\|^2/(2\sigma^2)\right)}{\sum_{x' \in \mathcal{D}, g' \in \mathcal{T}} \exp\left(-\left\|B^{-1}T_{g'}^{-1}By - Ax'\right\|^2/(2\sigma^2)\right)}
\end{aligned}
$$

where we used the invertibility of $B$ to write $B^+ = B^{-1}$ and simplified the Mahalanobis distance. The weights of the data-augmented estimator $\hat{x}_{\text{MMSE}}^{\text{aug}}$ are:

$$
\begin{aligned}
w_g^{\text{aug}}(x|y) &= \frac{\mathcal{N}\left(By; BAT_g x, \sigma^2 BB^\top\right)}{\sum_{x' \in \mathcal{D}, g' \in \mathcal{T}} \mathcal{N}\left(By; BAT_{g'}x', \sigma^2 BB^\top\right)} \\
&= \frac{\exp\left(-\left\|B^{-1}(By - BAT_g x)\right\|^2/(2\sigma^2)\right)}{\sum_{x' \in \mathcal{D}, g' \in \mathcal{T}} \exp\left(-\left\|B^{-1}(By - BAT_{g'}x')\right\|^2/(2\sigma^2)\right)} \\
&= \frac{\exp\left(-\left\|y - AT_g x\right\|^2/(2\sigma^2)\right)}{\sum_{x' \in \mathcal{D}, g' \in \mathcal{T}} \exp\left(-\left\|y - AT_{g'}x'\right\|^2/(2\sigma^2)\right)}.
\end{aligned}
$$

**Sufficiency ($\Longleftarrow$)** Assume $A$ and $B$ are circular convolutions (they commute with $T_g$) and $B$ is invertible.

Commutativity implies $B^{-1}T_g^{-1} = T_g^{-1}B^{-1}$. We can rearrange the term in the exponent of the above weights as follows:

$$\left\|B^{-1}T_g^{-1}By - Ax\right\|^2 = \left\|B^{-1}BT_g^{-1}y - B^{-1}BAx\right\|^2 = \left\|T_g^{-1}y - Ax\right\|^2.$$

Since $T_g$ is unitary and $A$ commutes with $T_g$, we have:

$$\left\|T_g^{-1}y - Ax\right\|^2 = \left\|T_g(T_g^{-1}y - Ax)\right\|^2 = \left\|y - T_g Ax\right\|^2 = \left\|y - AT_g x\right\|^2.$$

Thus $\hat{x}_{\mathcal{T}} = \hat{x}_{\text{MMSE}}^{\text{aug}}$. Furthermore, since the weights are independent of $B$, the physics-agnostic and physics-aware solvers are identical. The reconstruction equivariance follows immediately from Lemma B.3.

**Necessity ($\Rightarrow$)** Assume that $w_g^{\text{aug}}(x|y) = w_g(x|y)$ for all $y \in \mathbb{R}^M$ and all $x \in \mathbb{R}^N$ and that $A$ and $B$ are invertible. This implies that:

$$\left\| B^{-1} T_g^{-1} By - Ax \right\|^2 = \|y - AT_g x\|^2 + c(y) \tag{36}$$

for some function $c(y)$ independent of $x$ and $g$. We expand the squared norms and the terms depending solely on $y$ can be absorbed into $c(y)$.

$$\text{LHS} = \underbrace{\left\| B^{-1} T_g^{-1} By \right\|^2}_{\text{depends only on } y} - 2\langle B^{-1} T_g^{-1} By, Ax \rangle + \|Ax\|^2,$$

$$\text{RHS} = \underbrace{\|y\|^2}_{\text{depends only on } y} - 2\langle y, AT_g x \rangle + \|AT_g x\|^2 + c(y).$$

For the equality in Equation (36) to hold for all $x \in \mathbb{R}^N$ and $y \in \mathbb{R}^M$, the cross-terms (the bilinear forms coupling $y$ and $x$) must be identical. Moreover, since the estimator $\hat{x}_{\text{MMSE}}^{\text{aug}}$ is *independent* of $B$, the Equation (36) must hold for all invertible $B$. In particular, taking $B = \text{I}$, we deduce that

$$\left\langle T_g^{-1} y, Ax \right\rangle = \langle y, AT_g x \rangle \quad \text{for all } x \in \mathbb{R}^N, g \in \mathcal{T}, y \in \mathbb{R}^M.$$

This equality yields $T_g A = AT_g$, or that $A$ commutes with $T_g$. Plugging this result into the cross-term equality for a general invertible $B$, we get:

$$\left\langle B^{-1} T_g^{-1} By, Ax \right\rangle = \langle y, AT_g x \rangle = \left\langle T_g^{-1} y, Ax \right\rangle \quad \text{for all } x \in \mathbb{R}^N, g \in \mathcal{T}, y \in \mathbb{R}^M.$$

This implies that $A^\top B^{-1} T_g^{-1} B = A^\top T_g^{-1}$ for all $g$. Since $A$ is invertible, this implies that $B^{-1} T_g^{-1} B = T_g^{-1}$ for all $g \in \mathcal{T}$. Multiplying by $B$ on each side of the equality implies that $B$ commutes with $T_g$, concluding the proof. $\qquad\square$

### B.3. Local and Translation Equivariant MMSE estimator

**Proposition B.4** (Local and translation equivariant MMSE estimator)**.** *Suppose that the $N$ matrices $Q_n = \Pi_n B \in \mathbb{R}^{P \times M}$ have **constant rank** $r > 0$. The local and equivariant MMSE is defined for any $y \in \mathbb{R}^M$ by:*

$$\hat{x}_{\mathcal{T},loc}(y) = \phi^\star(By), \tag{37}$$

*where*

$$\phi_n^\star = f_{\phi^\star} \circ \Pi_n \quad \text{with} \quad f_{\phi^\star}(v) = \frac{\sum_{x \in \mathcal{D}} \sum_{n=1}^N x_n \mathcal{N}\left(v; Q_n Ax, \sigma^2 Q_n Q_n^\top\right)}{\sum_{x \in \mathcal{D}} \sum_{n=1}^N \mathcal{N}\left(v; Q_n Ax, \sigma^2 Q_n Q_n^\top\right)}. \tag{38}$$

*Proof.* We will verify that $\phi^\star$ is admissible and satisfies the optimality condition Proposition A.5.

**Admissibility.** Firstly, we show that $\phi^\star \in \mathcal{M}_{\mathcal{T},\text{loc}}$. By construction, we have $\phi^\star = (\phi_1^\star, \cdots \phi_N^\star) = (f_{\phi^\star} \circ \Pi_1, \cdots f_{\phi^\star} \circ \Pi_N)$, therefore $\phi^\star$ is local. For any $g \in \{1, \cdots N\}$, the group transformation (translation) $T_g$ is defined as $T_g : (x_1, \cdots, x_N) \in \mathbb{R}^N \mapsto (x_{1-g}, \cdots, x_{N-g}) \in \mathbb{R}^N$, where $x_{n-g} = x_{n-g \equiv N}$ (circular boundary). For any $g$, we have

$$\begin{aligned}
\phi^\star \circ T_g &= (\phi_1^\star \circ T_g, \cdots \phi_N^\star \circ T_g) \\
&= (f_{\phi^\star} \circ \Pi_1 \circ T_g, \cdots f_{\phi^\star} \circ \Pi_N \circ T_g) \\
&= (f_{\phi^\star} \circ \Pi_{1-g}, \cdots f_{\phi^\star} \circ \Pi_{N-g}) \quad \text{where } \Pi_{n-g} = \Pi_{n-g \equiv N} \text{ (circular boundary)} \\
&= T_g \circ \phi^\star
\end{aligned}$$

Therefore, $\phi^\star$ is translation equivariant. We deduce that $\phi^\star \in \mathcal{M}_{\mathcal{T},\text{loc}}$.

**Optimality condition.** We will verify that this estimator satisfies the optimality condition Proposition A.5. By using Equation (28), for any $\varphi \in \mathcal{M}_{\mathcal{T},\text{loc}}$, we have:

$$\begin{aligned}
&\mathbb{E}\left[\langle \varphi(B\boldsymbol{y}), \phi^\star(B\boldsymbol{y}) - \boldsymbol{x} \rangle \right] \\
&= \frac{1}{|\mathcal{D}|} \sum_{x \in \mathcal{D}} \int_{\mathbb{R}^M} \langle \varphi(By), \phi^\star(By) - x \rangle \mathcal{N}\left(y; Ax, \sigma^2 \text{I}_N\right) dy
\end{aligned}$$

$$= \frac{1}{|\mathcal{D}|} \sum_{x \in \mathcal{D}} \sum_{n=1}^{N} \int_{\mathbb{R}^M} f_\varphi(\Pi_n B y) \left(f_{\phi^\star}(\Pi_n B y) - x_n\right) \mathcal{N}\left(y; Ax, \sigma^2 \mathrm{I}_N\right) dy$$

$$= \frac{1}{|\mathcal{D}|} \sum_{x \in \mathcal{D}} \sum_{n=1}^{N} \int_{\mathbb{R}^P} f_\varphi(v) \left(f_{\phi^\star}(v) - x_n\right) \underbrace{\mathcal{N}\left(v; Q_n Ax, \sigma^2 Q_n Q_n^\top\right)}_{q(v,n,x)} d\mathcal{H}^r(v)$$

$$= \frac{1}{|\mathcal{D}|} \int_{\mathbb{R}^P} f_\varphi(v) \left(f_{\phi^\star}(v) \sum_{x \in \mathcal{D}} \sum_{n=1}^{N} q(v,n,x) - \sum_{x \in \mathcal{D}} \sum_{n=1}^{N} x_n q(v,n,x)\right) d\mathcal{H}^r(v)$$

$$= 0$$

$\square$

The constant rank assumption in Proposition B.4 can be relaxed by stratifying the image space according to the rank of the matrices $Q_n$ as follows.

**Proposition B.5** (Local and translation equivariant MMSE estimator – rank stratification). *Let $Q_n = \Pi_n B \in \mathbb{R}^{P \times M}$. For each rank $r \in \{0, 1, \ldots, P\}$, define the index set and the union of subspaces: $I_r = \{n \in [1:N] : \mathrm{rank}\,(Q_n) = r\}$, $E_r = \bigcup_{n \in I_r} \mathrm{Im}(Q_n) \subset \mathbb{R}^P$. We define the disjoint strata recursively: $\bar{E}_r = E_r \setminus \bigcup_{r' < r} E_{r'}$. Then $\bigcup_{r=0}^{P} \bar{E}_r = \bigcup_{n=1}^{N} \mathrm{Im}(Q_n)$ is a disjoint union and, for any $r' < r$, we have $\mathcal{H}^r(E_{r'}) = 0$.*

*Define the weighted density sums for $v \in \mathbb{R}^P$:*

$$a_r(v) = \sum_{x \in \mathcal{D}} \sum_{n \in I_r} x_n \mathcal{N}\left(v; Q_n Ax, \sigma^2 Q_n Q_n^\top\right),$$

$$b_r(v) = \sum_{x \in \mathcal{D}} \sum_{n \in I_r} \mathcal{N}\left(v; Q_n Ax, \sigma^2 Q_n Q_n^\top\right).$$

*The local and translation equivariant MMSE estimator $\phi^\star$ is given component-wise by $\phi_n^\star = f_{\phi^\star} \circ \Pi_n$, where:*

$$f_{\phi^\star}(v) = \begin{cases} \sum_{r=0}^{P} \frac{a_r(v)}{b_r(v)} \mathbb{1}_{\bar{E}_r}(v) & \text{if } v \in \bigcup \mathrm{Im}(Q)_n \text{ and } b_r(v) > 0, \\ 0 & \text{otherwise.} \end{cases} \tag{39}$$

*Proof.* **Admissibility (locality and translation equivariance).**

- Locality. By definition, the estimator is defined component-wise by $\phi_n^\star = f_{\phi^\star} \circ \Pi_n$, so it is local.

- Translation equivariance on $\mathrm{Im}(B)$. Let $T_g$ denote the circular translation on $\mathbb{R}^N$, the selection operators commute with $T_g$ by a simple index check

$$\Pi_n \circ T_g = \Pi_{n-g}, \qquad \forall\, n, g \in [\![1, N]\!].$$

Then, for any $v \in \mathrm{Im}(B)$ and any $n$,

$$\left[\phi^\star(T_g v)\right]_n = f_{\phi^\star}\left(\Pi_n T_g v\right) = f_{\phi^\star}\left(\Pi_{n-g} v\right) = \left[\phi^\star(v)\right]_{n-g} = \left[T_g \phi^\star(v)\right]_n. \tag{40}$$

Hence $\phi^\star \circ T_g = T_g \circ \phi^\star$ on $\mathrm{Im}(B)$, i.e., it is translation equivariant. Combining with locality gives $\phi^\star \in \mathcal{M}_{\mathcal{T}, \mathrm{loc}}$.

- Well-definedness. On each $\bar{E}_r$, $f_{\phi^\star}$ is defined by the ratio $a_r/b_r$. If $b_r(v) = 0$ on a negligible set (w.r.t. $\mathcal{H}^r$), assign any fixed value (*e.g.* 0); this does not affect admissibility nor optimality.

**Optimality.** For any $\varphi \in \mathcal{M}_{\mathcal{T}, \mathrm{loc}}$, we will verify that the optimality condition Proposition A.5 holds. By using Equation (28),

we have:

$$\mathbb{E}\left[\langle \varphi(B\boldsymbol{y}), \phi^{\star}(B\boldsymbol{y}) - \boldsymbol{x}\rangle\right]$$

$$= \frac{1}{|\mathcal{D}|}\sum_{x\in\mathcal{D}}\int_{\mathbb{R}^M}\langle \varphi(By), \phi^{\star}(By) - x\rangle\,\mathcal{N}\left(y; Ax, \sigma^2\mathrm{I}_N\right)dy$$

$$= \frac{1}{|\mathcal{D}|}\sum_{x\in\mathcal{D}}\sum_{n=1}^{N}\int_{\mathbb{R}^M} f_\varphi(\Pi_n By)\big(f_{\phi^\star}(\Pi_n By) - x_n\big)\mathcal{N}\left(y; Ax, \sigma^2 I\right)\,dy$$

$$\overset{(\star)}{=} \frac{1}{|\mathcal{D}|}\sum_{x\in\mathcal{D}}\sum_{n=1}^{N}\int_{\mathbb{R}^P} f_\varphi(v)\big(f_{\phi^\star}(v) - x_n\big)\mathcal{N}\left(v; Q_n Ax, \sigma^2 Q_n Q_n^\top\right)\,d\mathcal{H}^{r_n}(v),$$

where $(\star)$ uses Lemma A.2 and $r_n = \mathrm{rank}\,(Q_n)$. We decompose the summation over $n$ by rank. Note that for $n \in I_r$, the domain is $\mathrm{Im}(Q)_n$. We observe that $\mathrm{Im}(Q)_n \setminus \bar{E}_r \subseteq \bigcup_{r'<r} E_{r'}$. Since the union of lower-dimensional subspaces has $\mathcal{H}^r$-measure zero, we can restrict the integration domain from $\mathrm{Im}(Q)_n$ to $\mathrm{Im}(Q)_n \cap \bar{E}_r$ without changing the value of the integral. Thus, the previous expression becomes:

$$\sum_{r=0}^{P}\int_{\bar{E}_r} f_\varphi(v)\Big(f_{\phi^\star}(v)\underbrace{\sum_{x\in\mathcal{D}}\sum_{n\in I_r}\mathcal{N}\left(v; Q_n Ax, \sigma^2 Q_n Q_n^\top\right)}_{b_r(v)} \tag{41}$$

$$-\underbrace{\sum_{x\in\mathcal{D}}\sum_{n\in I_r} x_n\mathcal{N}\left(v; Q_n Ax, \sigma^2 Q_n Q_n^\top\right)}_{a_r(v)}\Big)d\mathcal{H}^r(v). \tag{42}$$

Choosing $f_{\phi^\star}(v) = a_r(v)/b_r(v)$ on $\bar{E}_r$ cancels each integrand, hence the optimality condition Proposition A.5 holds. $\square$

*Remark* B.6. If all $Q_n$ have the same rank $r$, then $\bar{E}_r = \bigcup_{n=1}^{N}\mathrm{Im}(Q_n)$ and the formula reduces to the constant-rank case in Proposition B.4, with a single ratio over $n = 1, \ldots, N$.

*Remark* B.7 (Singular limit and rank stratification). Consider a regularization of $B$ as $B^{(\epsilon)} = U\Sigma_\epsilon V^\top$ and $Q_n^{(\epsilon)} = \Pi_n B^{(\epsilon)}$, where $B = U\Sigma V^\top$ is the SVD of $B$ and the diagonal matrix $\Sigma_\epsilon$ is constructed by adding a small positive number $\epsilon$ to the singular values in $\Sigma$. For $\epsilon > 0$ (and $M \geq P$), each $Q_n^{(\epsilon)}$ has full row rank $P$. For each $(n, x)$, define the probability measure $\mu_{n,x}^{(\epsilon)}$ on $\mathbb{R}^P$ with density (Radon–Nikodým derivative) with respect to Lebesgue measure $\lambda^P$ given by $\frac{d\mu_{n,x}^{(\epsilon)}}{d\lambda^P}(v) = \mathcal{N}\left(v; Q_n^{(\epsilon)}Ax, \sigma^2 Q_n^{(\epsilon)}Q_n^{(\epsilon)T}\right)$. As $\epsilon \to 0$, $Q_n^{(\epsilon)} \to Q_n$ and some ranks $r_n = \mathrm{rank}\,(Q)_n$ may drop. Then $\mu_{n,x}^{(\epsilon)}$ converges weakly to a probability measure $\mu_{n,x}$ supported on the linear subspace $\mathrm{Im}(Q)_n$, which is absolutely continuous with respect to the Hausdorff measure $\mathcal{H}^{r_n}$ on $\mathrm{Im}(Q)_n$, with density $\frac{d\mu_{n,x}}{d\mathcal{H}^{r_n}}(v) = \mathcal{N}\left(v; Q_n Ax, \sigma^2 Q_n Q_n^\top\right)$, $v \in \mathrm{Im}(Q_n)$, where, $\mathcal{N}\left(\cdot; \mu, \Sigma\right)$ denotes the degenerate Gaussian density Definition A.1 with respect to the appropriate Hausdorff measure on its support (and vanishes off that support).

For $\epsilon > 0$, the estimator reads (constant-rank case, similar to Proposition B.4) $f_{\phi^\star}^{(\epsilon)}(v) = \frac{\sum_{x\in\mathcal{D}}\sum_{n=1}^{N} x_n\mathcal{N}(v; Q_n Ax, \sigma^2 Q_n Q_n^\top)}{\sum_{x\in\mathcal{D}}\sum_{n=1}^{N}\mathcal{N}(v; Q_n Ax, \sigma^2 Q_n Q_n^\top)}$. In the singular limit, for each $r$ and $\mathcal{H}^r$-a.e. $v \in \bar{E}_r$ with $b_r(v) > 0$, $\lim_{\epsilon\to 0} f_{\phi^\star}^{(\epsilon)}(v) = \frac{a_r(v)}{b_r(v)}$, *i.e.* the $\epsilon$-regularized estimator converges (stratum-wise, $\mathcal{H}^r$-a.e.) to the rank-stratified formula $f_{\phi^\star}(v) = \sum_{r=0}^{P}\frac{a_r(v)}{b_r(v)}\mathbb{1}_{\bar{E}_r}(v)$, interpreted up to $\mathcal{H}^r$-null sets on each $\bar{E}_r$. If all $Q_n$ have the same rank $r$, only the stratum $\bar{E}_r$ is nonempty, and the above reduces to the constant-rank expression in Proposition B.4.

*Proof of i) of Corollary 3.9 — Physics-agnostic LE-MMSE estimator.* When $B = \mathrm{I}$, the weights of the LE-MMSE estimator in Theorem 3.8 simplifies to

$$w_{n',n}(x|y) \propto \mathcal{N}\left(\Pi_{n'}y; \Pi_n Ax, \sigma^2\Pi_n\Pi_n^\top\right) \propto \exp\left(-\frac{1}{2\sigma^2}\|y[\omega_{n'}] - (Ax)[\omega_n]\|^2\right)$$

Therefore, the LE-MMSE estimator at pixel $n'$ reads:

$$\hat{x}_{\mathcal{T},\text{loc}}(y)[n'] = \frac{\sum_{x\in\mathcal{D}}\sum_{n=1}^{N} x_n \exp\left(-\frac{1}{2\sigma^2}\|y[\omega_{n'}] - (Ax)[\omega_n]\|^2\right)}{\sum_{x\in\mathcal{D}}\sum_{n=1}^{N} \exp\left(-\frac{1}{2\sigma^2}\|y[\omega_{n'}] - (Ax)[\omega_n]\|^2\right)}.$$

For small noise level $\sigma \to 0$, it returns the central pixel value of the patches in the dataset $\mathcal{D}$ whose degraded version $(Ax)[\omega_n]$ is closest to the observed patch $y[\omega_{n'}]$. Hence, the LE-MMSE estimator is a patch-work of training patches. □

*Proof of ii) of Corollary 3.9 — LE-MMSE is not a posterior mean.* We focus on the simplest case of denoising, that is $y = x + e$. Recall that the posterior mean satisfies the so-called Tweedie formula:

$$\mathbb{E}[x|y] = y + \sigma^2 \nabla \log p_y(y).$$

Taking the gradient of both sides w.r.t $y$ yields:

$$\nabla_y \mathbb{E}[x|y] = I + \sigma^2 \nabla^2 \log p_y(y).$$

Therefore, the Jacobian of any pure MMSE estimator (posterior mean) must be symmetric since the Hessian of $\log p_y$ is symmetric. To simplify the notation, we use $f$ instead of $\hat{x}_{\mathcal{T},\text{loc}}$ in what follows. Using the fact that the scaling by $\sigma^{-2}$ does not affect the symmetry (note that $p_y$ is the convolution of $p_x$ and a Gaussian so it's $\mathcal{C}^\infty$), if $\hat{x}_{\mathcal{T},\text{loc}}$ were a posterior mean, we must have:

$$\frac{\partial}{\partial y_m} f_n(y) = \frac{\partial}{\partial y_n} f_m(y), \quad \forall n, m \text{ and } \forall y. \tag{43}$$

For notation simplicity, we let $e_{n,l}(x,y) = \exp\left(-\frac{\|y[\omega_n] - x[\omega_l]\|^2}{2\sigma^2}\right)$. We have the weights of the LE-MMSE estimator in Theorem 3.8 becomes:

$$w_{n,l}(x|y) = \exp\left(-\frac{\|y[\omega_n] - x[\omega_l]\|}{2\sigma^2}\right)/Z_n(y) = \frac{e_{n,l}(x,y)}{Z_n(y)}.$$

where $Z_n(y) = \sum_{x'\in\mathcal{D}}\sum_{l'=1}^{N} e_{n,l'}(x',y)$ is the normalization constant. Therefore, the LE-MMSE estimator at pixel $n$ reads:

$$f_n(y) = \frac{1}{Z_n(y)} \sum_{x\in\mathcal{D}} \sum_{l=1}^{N} x_l \cdot e_{n,l}(x,y).$$

That is:

$$f_n(y) = \frac{S_n(y)}{Z_n(y)} \qquad \text{where} \qquad S_n(y) = \sum_{x\in\mathcal{D}} \sum_{l=1}^{N} x_l \cdot e_{n,l}(x,y).$$

A key observation is that *this estimator is real-analytic for $\sigma > 0$*. To prove it, we remind that the product of polynomials and exponentials are real-analytic functions, and that the sum and quotient (with non-vanishing denominator) of real-analytic functions are also real-analytic. The denominator $Z_n(y)$ does not vanish for $\sigma > 0$, which concludes the proof of real-analyticity.

We will use the following classical result (see e.g. (Federer, 2014, Section 3.1.24) and (Mityagin, 2020) for an elementary proof):

**Lemma B.8** (Zeros of real-analytic functions). *Let $f : \mathbb{R}^p \to \mathbb{R}$ be a real-analytic function for some $p \in \mathbb{N}^*$. If $f$ is not identically zero, then the set of zeros of $f$ has Lebesgue measure zero.*

Applying this lemma, it therefore suffices to find a dataset $\mathcal{D}$ and a point $y$ such that $\frac{\partial}{\partial y_m} f_n(y) - \frac{\partial}{\partial y_n} f_m(y) \neq 0$. To this end, we consider the case of overlapping patches $\omega_n \cap \omega_m \neq \emptyset$ with $n \neq m$, which is always possible for patch sizes $P \geq 2$. Now consider the case where $n \in \omega_m$ and $m \in \omega_n$ (they are equivalent).

The chain rule gives

$$\frac{\partial e_{n,l}}{\partial y_m}(x,y) = -\sigma^{-2}(y_m - x_{l+m-n}) \cdot e_{n,l}(x,y)$$

and the quotient rule gives:

$$\frac{\partial w_{n,l}}{\partial y_m}(x,y) = \frac{\partial e_{n,l}}{\partial y_m}(x,y) \cdot \frac{1}{Z_n(y)} - e_{n,l}(x,y) \cdot \frac{\partial Z_n}{\partial y_m}(y) \cdot \frac{1}{Z_n(y)^2}$$

$$= -\sigma^{-2}(y_m - x_{l+m-n}) \cdot \frac{e_{n,l}(x,y)}{Z_n(y)} - \frac{e_{n,l}(x,y)}{Z_n(y)^2} \cdot \frac{\partial Z_n(y)}{\partial y_m}$$

$$= -\sigma^{-2}(y_m - x_{l+m-n}) \cdot w_{n,l}(x,y) - w_{n,l}(x,y) \cdot \frac{1}{Z_n(y)} \cdot \left( \sum_{x' \in \mathcal{D}} \sum_{l'=1}^{N} \frac{\partial e_{n,l'}}{\partial y_m}(x',y) \right)$$

$$= -\sigma^{-2}\left( (y_m - x_{l+m-n}) \cdot w_{n,l}(x,y) + w_{n,l}(x,y) \cdot \frac{1}{Z_n(y)} \cdot \left( \sum_{x' \in \mathcal{D}} \sum_{l'=1}^{N} (y_m - x_{l'+m-n}) \cdot e_{n,l'}(x',y) \right) \right)$$

$$= -\sigma^{-2}\left( (y_m - x_{l+m-n}) \cdot w_{n,l}(x,y) + w_{n,l}(x,y) \cdot \frac{1}{Z_n(y)} \cdot \left( y_m \cdot Z_n(y) - \sum_{x' \in \mathcal{D}} \sum_{l'=1}^{N} x_{l'+m-n} \cdot e_{n,l'}(x',y) \right) \right)$$

$$= \sigma^{-2} \cdot \left( x_{l+m-n} \cdot w_{n,l}(x,y) - w_{n,l}(x,y) \cdot \frac{1}{Z_n(y)} \cdot \sum_{x' \in \mathcal{D}} \sum_{l'=1}^{N} x_{l'+m-n} \cdot e_{n,l'}(x',y) \right)$$

$$= \sigma^{-2} \cdot \left( x_{l+m-n} \cdot w_{n,l}(x,y) - w_{n,l}(x,y) \cdot \sum_{x' \in \mathcal{D}} \sum_{l'=1}^{N} x_{l'+m-n} \cdot w_{n,l'}(x',y) \right)$$

$$= \sigma^{-2} w_{n,l}(x,y) \left( x_{l+m-n} - \bar{x}_n^{(m-n)}(y) \right)$$

where we define $\bar{x}_n^{(k)}(y) = \sum_{x' \in \mathcal{D}} \sum_{l'=1}^{N} x_{l'+k} \cdot w_{n,l'}(x',y)$ the local posterior mean of pixel $n$ with offset $k$. Finally, differentiating $f_n(y)$ gives:

$$\frac{\partial f_n}{\partial y_m}(y) = \sum_{x \in \mathcal{D}} \sum_{l=1}^{N} x_l \cdot \frac{\partial w_{n,l}}{\partial y_m}(x,y)$$

$$= \sigma^{-2} \sum_{x \in \mathcal{D}} \sum_{l=1}^{N} x_l \cdot w_{n,l}(x,y) \left( x_{l+m-n} - \bar{x}_n^{(m-n)}(y) \right)$$

Set $n = 2$ and $m = 3$. Set $\mathcal{D} = \{x, 0, \dots 0\}$ to be dataset containing only one non-zero image $x$ and the rest are zeros. We choose $x$ such that $x_2 > 0$ and $x_i = 0$ for $i \neq 2$. The derivatives simplify to:

$$\frac{\partial f_2}{\partial y_3}(y) = \sigma^{-2} x_2 w_{2,2}(x,y)(x_3 - \bar{x}_2^{(1)}(y)) = -\sigma^{-2} x_2^2 w_{2,2}(x,y) w_{2,1}(x,y)$$

$$\frac{\partial f_3}{\partial y_2}(y) = \sigma^{-2} x_2 w_{3,2}(x,y)(x_1 - \bar{x}_3^{(-1)}(y)) = -\sigma^{-2} x_2^2 w_{3,2}(x,y) w_{3,3}(x,y)$$

Fix an index $1 \leq i \leq N$ such that $y_i$ is a variable appearing in the vector $y[\omega_3]$ but not in the vector $y[\omega_2]$ (note that $i \neq 3$, otherwise $y_i = y_3$ is the center of the patch $y[\omega_3]$ and it's also in the patch $y[\omega_2]$ since the patch size $P > 1$). We have

$$\frac{\partial w_{2,2}(x,y)}{\partial y_i} = \frac{\partial w_{2,1}(x,y)}{\partial y_i} = 0 \quad \text{and} \quad \frac{\partial^2 f_2}{\partial y_3 y_i}(x,y) = 0.$$

On the other hand, since $i \neq 3$, we have

$$\frac{\partial w_{3,2}(x,y)}{\partial y_i} = \sigma^{-2} w_{3,2}(x,y) \left( x_{2+i-3} - \bar{x}_3^{(i-3)}(y) \right) = -\sigma^{-2} x_2 w_{3,2}(x,y) w_{3,2-(i-3)}(x,y)$$

$$\frac{\partial w_{3,3}(x,y)}{\partial y_i} = \sigma^{-2} w_{3,3}(x,y) \left( x_i - \bar{x}_3^{(i-3)}(y) \right) = -\sigma^{-2} x_2 w_{3,3}(x,y) w_{3,2-(i-3)}(x,y).$$

We obtain

$$\frac{\partial^2 f_3(x,y)}{\partial y_2 y_i} = \sigma^{-4} x_2^3 w_{3,2}(x,y) w_{3,2-(i-3)} w_{3,3}(x,y) + \sigma^{-4} x_2^3 w_{3,3}(x,y) w_{3,2-(i-3)}(x,y) w_{3,2}(x,y) > 0.$$

Therefore, for this value of $x$, $\frac{\partial f_3}{\partial y_2}$ and $\frac{\partial f_2}{\partial y_3}$ are independent, when seen as functions of $y$. We deduce that the equation $\frac{\partial f_2}{\partial y_3} = \frac{\partial f_3}{\partial y_2}$ is a nondegenerate analytic equation in variables $\mathcal{D}, y$. Using Lemma B.8 and by Fubini's theorem, the set $\left\{ \mathcal{D}, \forall y, \frac{\partial f_2}{\partial y_3}(y) = \frac{\partial f_3}{\partial y_2}(y) \right\}$ also has zero measure.

In other words, for almost every dataset $\mathcal{D}$, there exists $y$ such that the symmetry condition Equation (43) does not hold, and the LE-MMSE estimator is not a posterior mean. $\qquad\square$

*Table 4.* Details of neural network architectures with various receptive fields used in our experiments for images at $32 \times 32$ resolution.

| | RECEPTIVE FIELD (PATCH SIZE) | ARCHI. HYPER-PARAMETERS | | NUM. PARAMETERS |
|---|---|---|---|---|
| UNET2D | | `block_out_channels` | `kernel_size` | |
| | 5 | $(96, 224, 480)$ | $(3, 1, 1, 1)$–1 | 4.6M |
| | 7 | $(64, 192, 448)$ | $(3, 1, 1, 1)$–3 | 5.1M |
| | 9 | $(96, 192, 288)$ | $(3, 1, 1, 1)$–5 | 4.3M |
| | 11 | $(64, 96, 160, 288)$ | $(3, 1, 1, 1, 3)$–1 | 4.3M |
| RESNET | | `num_res_blocks` | `num_channels` | |
| | 5 | 1 | 640 | 4.5M |
| | 7 | 2 | 448 | 4.2M |
| | 9 | 3 | 384 | 4.5M |
| | 11 | 4 | 328 | 4.4M |
| PATCHMLP | | `hidden_dim` | `num_blocks` | |
| | 5 | 7168 | 5 | 3.5M |
| | 7 | 6144 | 5 | 3.7M |
| | 9 | 5120 | 5 | 4.3M |
| | 11 | 3072 | 5 | 4.0M |

## C. Numerical experiments

### C.1. Datasets

We use the following datasets in our experiments: FFHQ (Karras et al., 2019) downscaled to $32 \times 32$ or $64 \times 64$, CIFAR-10 (Krizhevsky et al., 2009) and Fashion-MNIST (Xiao et al., 2017). For each dataset, we randomly select $10,000$ images for training, which is denoted by $\mathcal{D}$ in the main paper.

### C.2. Network architectures

For the local and translation equivariant estimator, we examine 3 different architectures with noise level $\sigma$ conditioning:

- UNet2D: we use a state-of-the-art UNet2D (Ronneberger et al., 2015) from the `diffusers`[2] library. The model has several downsampling and upsampling layers and skip connections, with varying channel dimensions (defined by `block_out_channels`) and kernel size (defined by `kernel_size`, for down-blocks and mid-blocks). The architecture is modified slightly (circular boundary conditions and kernel sizes of convolutional layers) to have a desired receptive field and to ensure translation equivariance. The noise level $\sigma$ is conditioned using a time embedding module with linear layers.

- ResNet: we use a minimal ResNet with residual block (He et al., 2016), with a $1 \times 1$ convolutional layer at the beginning and at the end, similar to (Kamb & Ganguli, 2025). Each residual block contains a convolutional layer with $3 \times 3$ kernels at a channel dimension defined by `num_channels`, followed by a ReLU nonlinearity. The noise level $\sigma$ is conditioned using sine-cosine positional embedding.

- PatchMLP: a fully local MLP acting on patches. The MLP has 5 residual blocks, each containing two linear layers with hidden dimension of `hidden_dim`, followed by a layer normalization and GELU activation. The noise level $\sigma$ is conditioned using sine-cosine positional embedding.

All convolutional layers use circular padding to ensure translation equivariance and they have approximately 4 million parameters. Details of the architectures with various receptive fields are provided in Table 4.

### C.3. Training procedure

All models are trained on a single NVIDIA A100 GPU with the following settings:

---

[2]https://huggingface.co/docs/diffusers/en/api/models/unet2d

- Optimizer: Adam optimizer (Kingma & Ba, 2014)

- Learning rate: starting at $10^{-4}$ with cosine decay schedule and minimum learning rate at $10^{-6}$.

- Number of epochs: 600 for images at $32 \times 32$ resolution and 900 for images at $64 \times 64$ resolution.

- Batch size: 256

- Exponential moving average (EMA) with decay rate 0.99 from the 1000-th training step and updates every 5 steps. The EMA weights are used for evaluation.

### C.4. Forward operators

We consider the 3 representative inverse problems as forward operators $A$:

- Denoising: the forward operator is simply the identity $A = I$ and only the physics-agnostic estimator is applicable in this case.

- Inpainting: we consider the inpainting operator with a center square mask of size $15 \times 15$. The forward operator $A$ is therefore a diagonal matrix with 0 on the masked pixels and 1 elsewhere. In this case, the pseudo-inverse is $BA^+ = A$: it simply removes noise inside the masked region and keeps the observed pixels unchanged. In our experiments, we consider the physics-aware estimator with $B = A^+ + \epsilon I$, where $\epsilon = 10^{-5}$ is a small regularization parameter. This ensures that $B$ is full-rank and we can apply the constant-rank formula of the LE-MMSE estimator in Theorem 3.8. As $\epsilon$ is very small, this estimator closely approximates the ideal physics-aware estimator with $B = A^+$, as discussed in Remark B.7.

- Deconvolution: we consider an isotropic Gaussian blur kernel with standard deviation 1.0 with circular boundary conditions. The same kernel is used for all color channels. In this case, we build the full matrix $A$ as a block-circulant matrix representing the convolution operation and $B$ is its pseudo-inverse, which is computed once, in double precision.

### C.5. Analytical formula implementation

Implementing the analytical formulas of the E-MMSE Theorem 3.5 and the LE-MMSE Theorem 3.8 estimators requires computing many distance terms between images or patches in the dataset $\mathcal{D}$ and the observed measurement $y$. We process by batch to avoid memory overflow. For numerical stability and avoidance of overflow/underflow, the exponential terms are accumulated using an online log-sum-exp trick. All theoretical estimators are computed in an *exact* manner without any approximation, modular finite precision arithmetic. We use PyTorch (Paszke et al., 2019) for implementation and all computations are performed in single precision (FP32) on a single A-100 GPU, unless otherwise specified. All estimators are implemented, even the rank-deficient cases. Full implementation details are available at https://github.com/mh-nguyen712/analytical_mmse.

# D. Additional numerical results

## D.1. Comparison between trained neural networks and the analytical LE-MMSE estimator

We show in Figures 11 to 13 additional PSNR results between trained neural networks and the analytical LE-MMSE estimator for different architectures (UNet2D, ResNet, PatchMLP) on both training and test sets of various datasets. We recover a consistent conclusion across different settings: the trained neural networks closely approximate the analytical LE-MMSE estimator, with PSNR values exceeding 20 in most cases and often reaching above 30 dB.

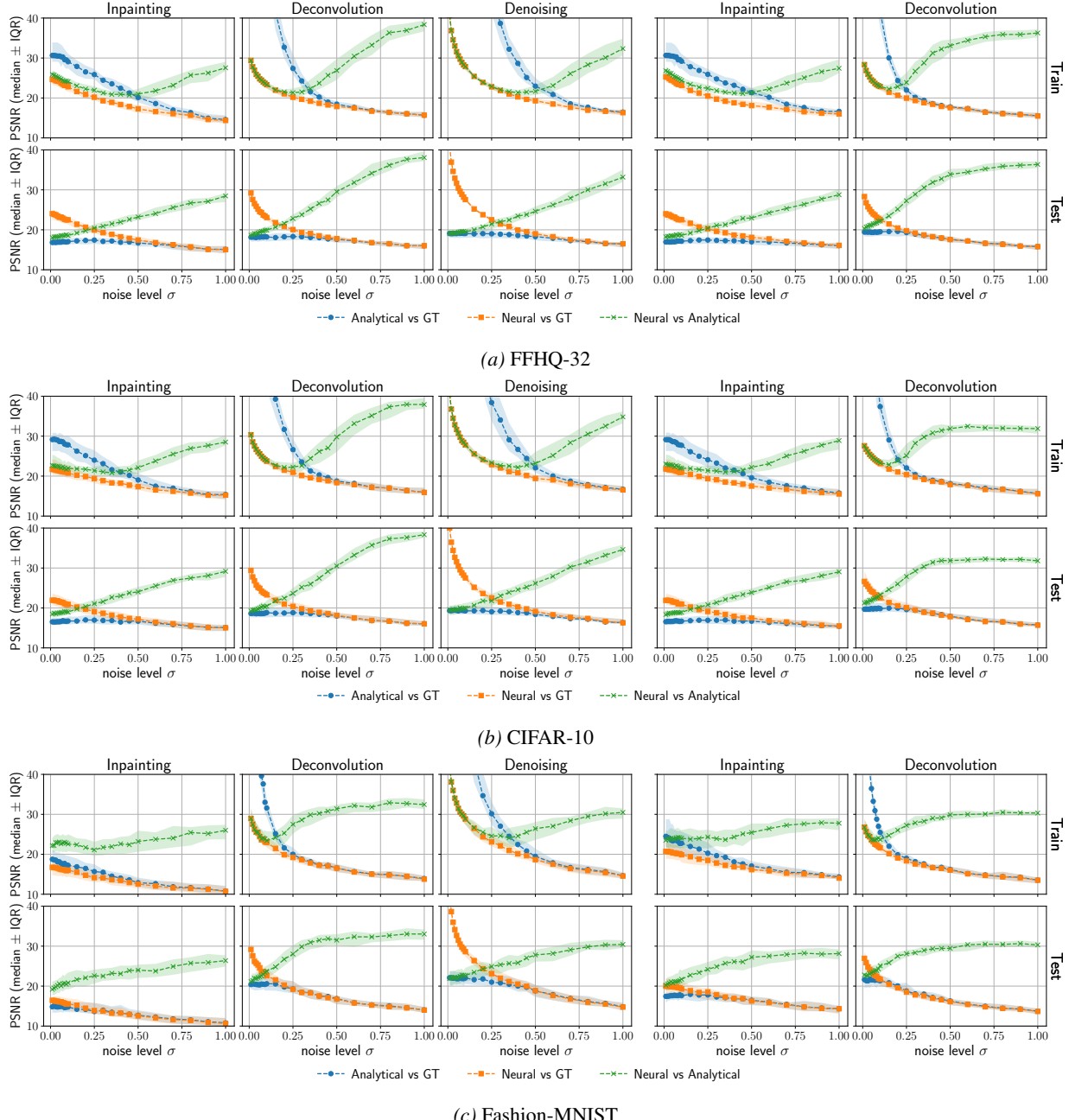

*(a)* FFHQ-32

*(b)* CIFAR-10

*(c)* Fashion-MNIST

*Figure 11.* Additional PSNR between trained UNet2D and the analytical formula of the LE-MMSE for different inverse problems on both training and test sets of various datasets. The patch size is $P = 11 \times 11$. Left: physics-agnostic estimator with $B = I$. Right: physics-aware estimator with $B = A^+$. We recover the same conclusions as in Figure 5.

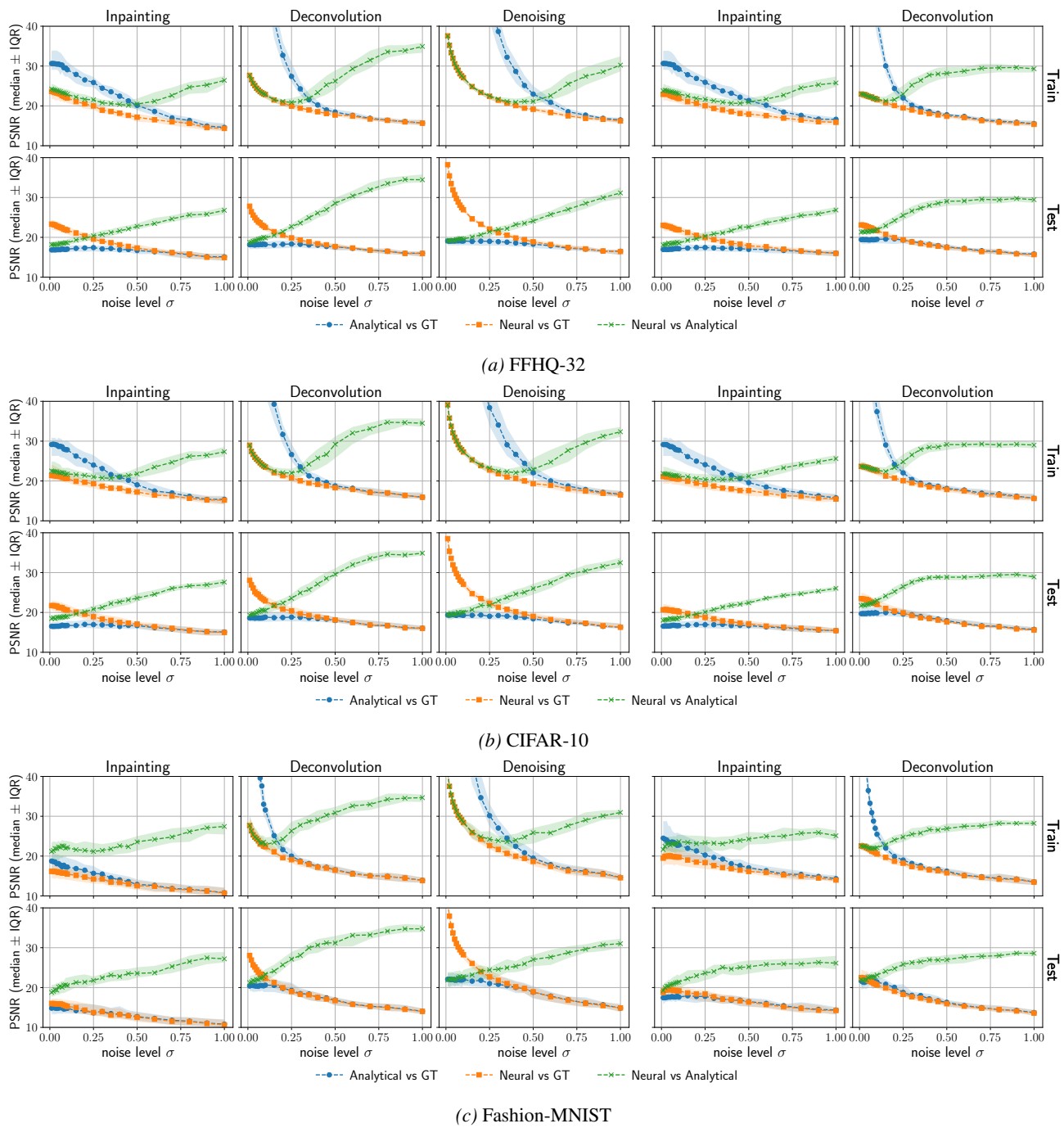

*Figure 12.* PSNR between trained ResNet and the analytical formula of the LE-MMSE for different inverse problems on both training and test sets of various datasets. The patch size is $P = 11 \times 11$. Left: physics-agnostic estimator with $B = I$. Right: physics-aware estimator with $B = A^+$. We recover the same conclusions accross architectures and settings.

**Structural and Inception metrics: SSIM and LPIPS**    To complement the PSNR metric, we also compute the structural similarity index (SSIM) and the Learned Perceptual Image Patch Similarity (LPIPS) metrics.

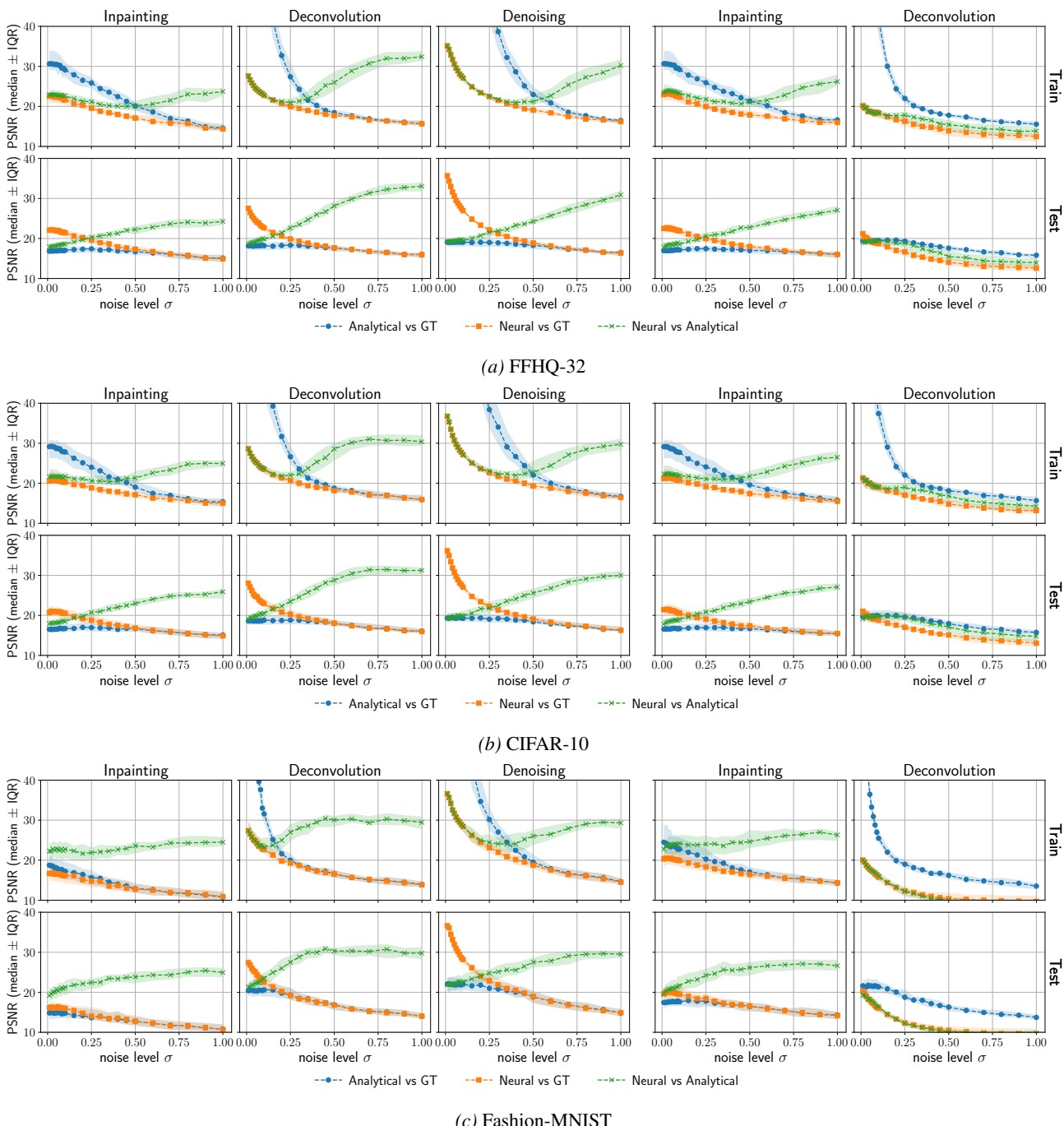

*(a)* FFHQ-32

*(b)* CIFAR-10

*(c)* Fashion-MNIST

*Figure 13.* PSNR between trained PatchMLP neural network and the analytical formula of the LE-MMSE for different inverse problems on both training and test sets of various datasets. The patch size is $P = 11 \times 11$. We recover the same conclusions as in Figures 5, 11 and 12. An exception is observed for the deconvolution task with physics-aware models (rightmost column). Here, the noise is amplified by the inversion of the blurring operator and the local PatchMLP architecture struggles to accurately reconstruct fine details, leading to a lower PSNR comparing to CNNs. We hypothesize that CNNs have other inductive biases that are more suited to handle such challenging tasks.

### D.2. Addition qualitative comparison

We provide additional qualitative comparisons between the analytical LE-MMSE estimator and trained neural networks (UNet2D, ResNet, PatchMLP) for different patch sizes in Figures 15 to 18 on the FFHQ, CIFAR10 and FashionMNIST

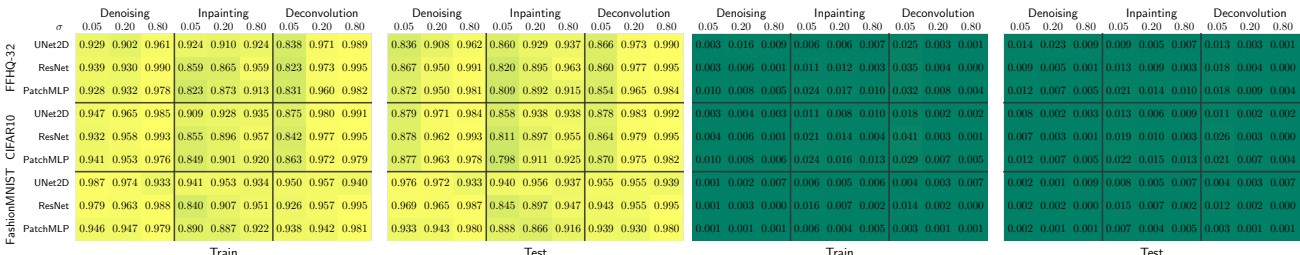

| | | Denoising | | | Inpainting | | | Deconvolution | | | Denoising | | | Inpainting | | | Deconvolution | | | Denoising | | | Inpainting | | | Deconvolution | | | Denoising | | | Inpainting | | | Deconvolution | | |
|---|---|---|---|---|---|---|---|---|---|---|---|---|---|---|---|---|---|---|---|---|---|---|---|---|---|---|---|---|---|---|---|---|---|---|---|---|---|
| $\sigma$ | | 0.05 | 0.20 | 0.80 | 0.05 | 0.20 | 0.80 | 0.05 | 0.20 | 0.80 | 0.05 | 0.20 | 0.80 | 0.05 | 0.20 | 0.80 | 0.05 | 0.20 | 0.80 | 0.05 | 0.20 | 0.80 | 0.05 | 0.20 | 0.80 | 0.05 | 0.20 | 0.80 | 0.05 | 0.20 | 0.80 | 0.05 | 0.20 | 0.80 | 0.05 | 0.20 | 0.80 |
| FFHQ-32 UNet2D | | 0.929 | 0.902 | 0.961 | 0.924 | 0.910 | 0.924 | 0.838 | 0.971 | 0.989 | 0.836 | 0.908 | 0.962 | 0.860 | 0.929 | 0.937 | 0.866 | 0.973 | 0.990 | 0.003 | 0.016 | 0.009 | 0.006 | 0.006 | 0.007 | 0.025 | 0.003 | 0.001 | 0.014 | 0.023 | 0.009 | 0.009 | 0.005 | 0.007 | 0.013 | 0.003 | 0.001 |
| FFHQ-32 ResNet | | 0.939 | 0.930 | 0.990 | 0.859 | 0.865 | 0.959 | 0.823 | 0.973 | 0.995 | 0.867 | 0.950 | 0.991 | 0.820 | 0.895 | 0.963 | 0.860 | 0.977 | 0.995 | 0.003 | 0.006 | 0.001 | 0.011 | 0.012 | 0.003 | 0.035 | 0.004 | 0.000 | 0.009 | 0.005 | 0.001 | 0.013 | 0.009 | 0.003 | 0.018 | 0.004 | 0.000 |
| FFHQ-32 PatchMLP | | 0.928 | 0.932 | 0.978 | 0.823 | 0.873 | 0.913 | 0.831 | 0.960 | 0.982 | 0.872 | 0.950 | 0.981 | 0.809 | 0.892 | 0.915 | 0.854 | 0.965 | 0.984 | 0.010 | 0.008 | 0.005 | 0.024 | 0.017 | 0.010 | 0.032 | 0.008 | 0.004 | 0.012 | 0.007 | 0.005 | 0.021 | 0.014 | 0.010 | 0.018 | 0.009 | 0.004 |
| CIFAR10 UNet2D | | 0.947 | 0.965 | 0.985 | 0.909 | 0.928 | 0.935 | 0.875 | 0.980 | 0.991 | 0.879 | 0.971 | 0.984 | 0.858 | 0.938 | 0.938 | 0.878 | 0.983 | 0.992 | 0.003 | 0.004 | 0.003 | 0.011 | 0.008 | 0.010 | 0.018 | 0.002 | 0.002 | 0.008 | 0.002 | 0.003 | 0.013 | 0.006 | 0.009 | 0.011 | 0.002 | 0.002 |
| CIFAR10 ResNet | | 0.932 | 0.958 | 0.993 | 0.855 | 0.896 | 0.957 | 0.842 | 0.977 | 0.995 | 0.878 | 0.962 | 0.993 | 0.811 | 0.897 | 0.955 | 0.864 | 0.979 | 0.995 | 0.004 | 0.004 | 0.001 | 0.021 | 0.014 | 0.004 | 0.041 | 0.003 | 0.001 | 0.007 | 0.003 | 0.001 | 0.019 | 0.010 | 0.003 | 0.026 | 0.003 | 0.000 |
| CIFAR10 PatchMLP | | 0.941 | 0.953 | 0.976 | 0.849 | 0.901 | 0.920 | 0.863 | 0.972 | 0.979 | 0.877 | 0.963 | 0.978 | 0.798 | 0.911 | 0.925 | 0.870 | 0.975 | 0.982 | 0.010 | 0.008 | 0.006 | 0.024 | 0.016 | 0.013 | 0.029 | 0.007 | 0.005 | 0.012 | 0.007 | 0.005 | 0.022 | 0.015 | 0.013 | 0.021 | 0.007 | 0.004 |
| FashionMNIST UNet2D | | 0.987 | 0.974 | 0.933 | 0.941 | 0.953 | 0.934 | 0.950 | 0.957 | 0.940 | 0.976 | 0.972 | 0.933 | 0.940 | 0.956 | 0.937 | 0.955 | 0.955 | 0.939 | 0.001 | 0.002 | 0.007 | 0.006 | 0.005 | 0.006 | 0.004 | 0.003 | 0.007 | 0.002 | 0.001 | 0.009 | 0.008 | 0.005 | 0.007 | 0.004 | 0.003 | 0.007 |
| FashionMNIST ResNet | | 0.979 | 0.963 | 0.988 | 0.840 | 0.907 | 0.951 | 0.926 | 0.957 | 0.995 | 0.969 | 0.965 | 0.987 | 0.845 | 0.897 | 0.947 | 0.943 | 0.955 | 0.995 | 0.001 | 0.003 | 0.000 | 0.016 | 0.007 | 0.002 | 0.014 | 0.002 | 0.000 | 0.002 | 0.002 | 0.000 | 0.015 | 0.007 | 0.002 | 0.012 | 0.002 | 0.000 |
| FashionMNIST PatchMLP | | 0.946 | 0.947 | 0.979 | 0.890 | 0.887 | 0.922 | 0.938 | 0.942 | 0.981 | 0.933 | 0.943 | 0.980 | 0.888 | 0.866 | 0.916 | 0.939 | 0.930 | 0.980 | 0.001 | 0.001 | 0.001 | 0.006 | 0.004 | 0.005 | 0.003 | 0.001 | 0.001 | 0.002 | 0.001 | 0.001 | 0.007 | 0.004 | 0.005 | 0.003 | 0.001 | 0.001 |
| | | Train | | | | | | | | | Test | | | | | | | | | Train | | | | | | | | | Test | | | | | | | | |

*Figure 14.* Addition metrics for the alignment between the trained UNet2D and the analytical LE-MMSE, with receptive field of $P = 5 \times 5$. This complements the PSNR results in Figure 5 and confirms the same conclusions. Left: SSIM. Right: LPIPS.

datasets. Overall, the neural networks closely match the analytical solution on both training and test sets across all datasets and inverse problems, confirming our theoretical findings.

*Figure 15.* Qualitative comparison. The patch size is $P = 5 \times 5$ and $B = \mathrm{I}$. The noise level is $\sigma = 0.05, 0.2, 0.8$ from left to right for each column. The neural networks (UNet2D, ResNet, and PatchMLP) closely match the analytical LE-MMSE on both training and test sets across all datasets (FFHQ,CIFAR10, and FashionMNIST) and inverse problems, confirming our theoretical findings.

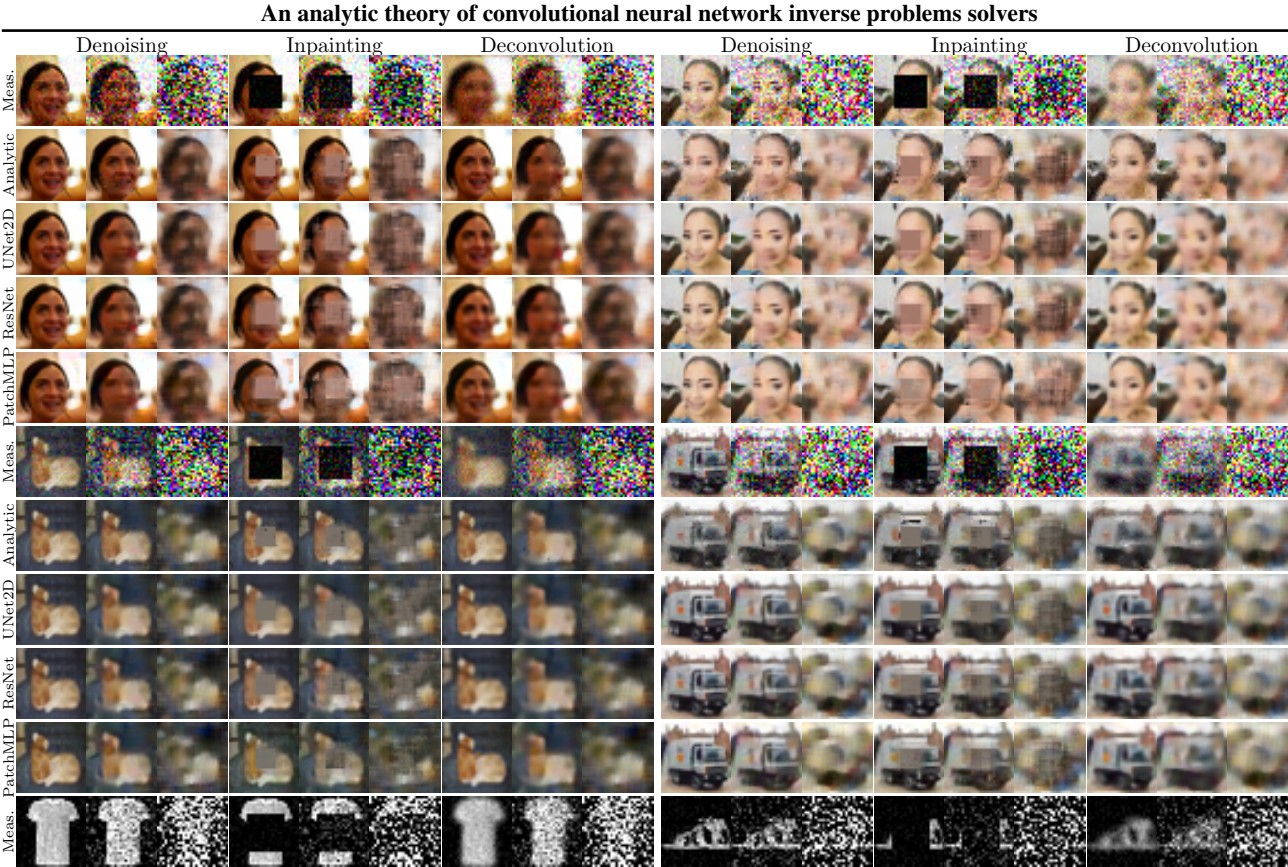

*Figure 16.* Qualitative comparison between the analytical formula LE-MMSE, UNet2D, ResNet, and PatchMLP, with $B = I$ on FFHQ, CIFAR10 and FashionMNIST. The patch size is $P = 7 \times 7$. The noise level is $\sigma = 0.05, 0.2, 0.8$ from left to right for each column. The neural networks closely match the analytical solution on both training and test sets across all datasets and inverse problems, confirming our theoretical findings. Yet, some discrepancies can be observed, especially at low noise levels on the test set, which we attribute to generalization issues discussed in Section 4.2.

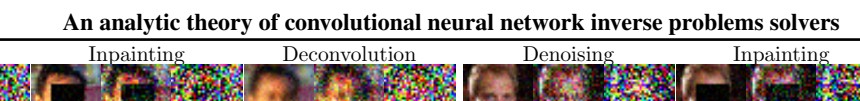

*Figure 17.* Qualitative comparison between the analytical formula LE-MMSE, UNet2D, ResNet, and PatchMLP, with $B = I$ on FFHQ, CIFAR10 and FashionMNIST. The patch size is $P = 9$. The noise level is $\sigma = 0.05, 0.2, 0.8$ from left to right for each column. The neural networks closely match the analytical solution on both training and test sets across all datasets and inverse problems, confirming our theoretical findings. Yet, some discrepancies can be observed, especially at low noise levels on the test set, which we attribute to generalization issues discussed in Section 4.2.

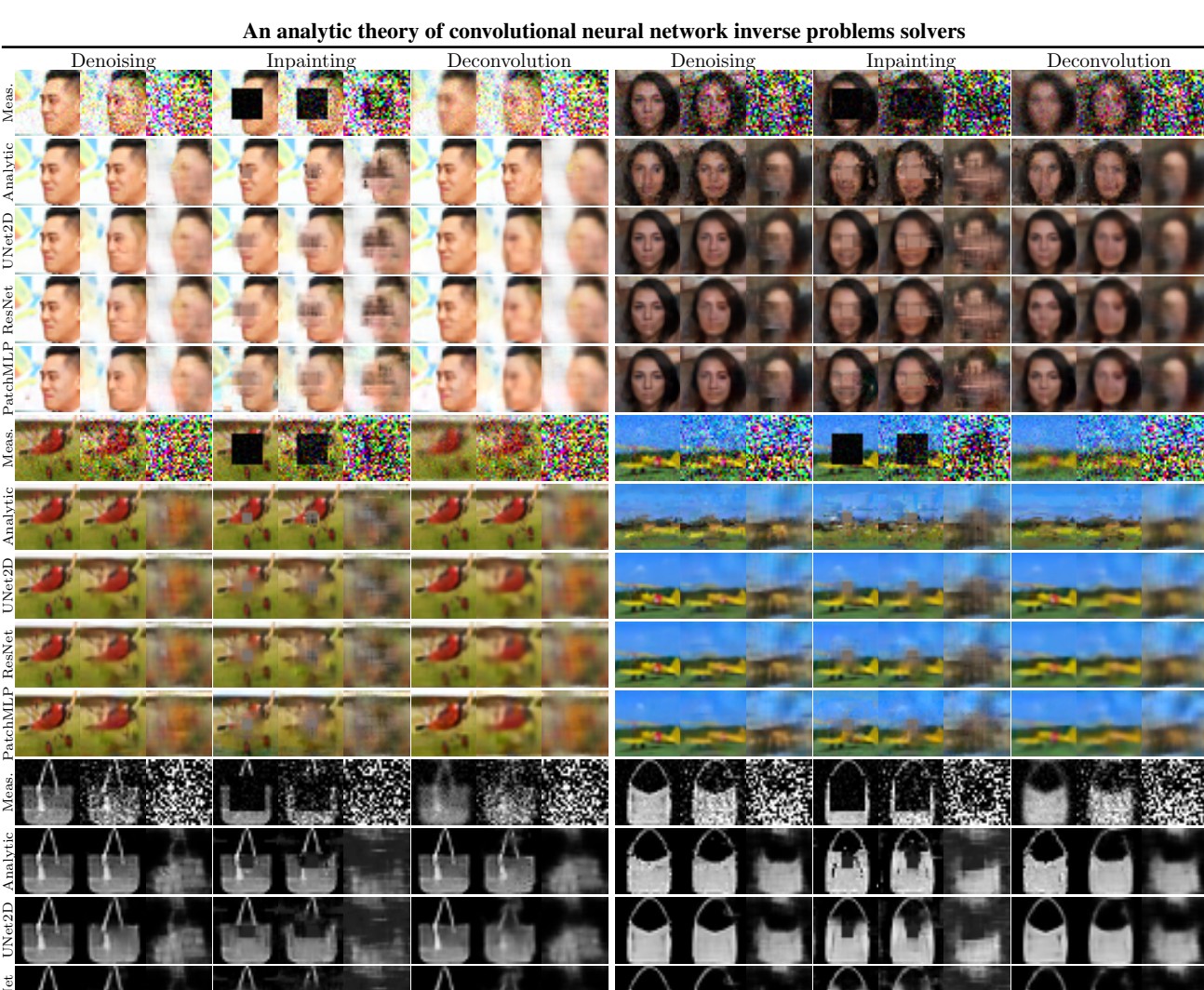

*Figure 18.* Qualitative comparison between the analytical formula LE-MMSE, UNet2D, ResNet, and PatchMLP, with $B=I$ on FFHQ, CIFAR10 and FashionMNIST. The patch size is $P = 11$. The noise level is $\sigma = 0.05, 0.2, 0.8$ from left to right for each column. The neural networks closely match the analytical solution on both training and test sets across all datasets and inverse problems, confirming our theoretical findings. Yet, some discrepancies can be observed, especially at low noise levels on the test set, which we attribute to generalization issues discussed in Section 4.2.

### D.3. Influence of the patch size (receptive field)

**Density of the patch distribution**    We analyze here the influence of the patch size on the density of the patch distribution in FFHQ-32 dataset. We compute the negative-log-density of patches as a function of the patch size, for points on either the training set or the test set. The patch density is exactly the denominator term in the analytical LE-MMSE formula Equation (7). The results are shown in Figure 19.

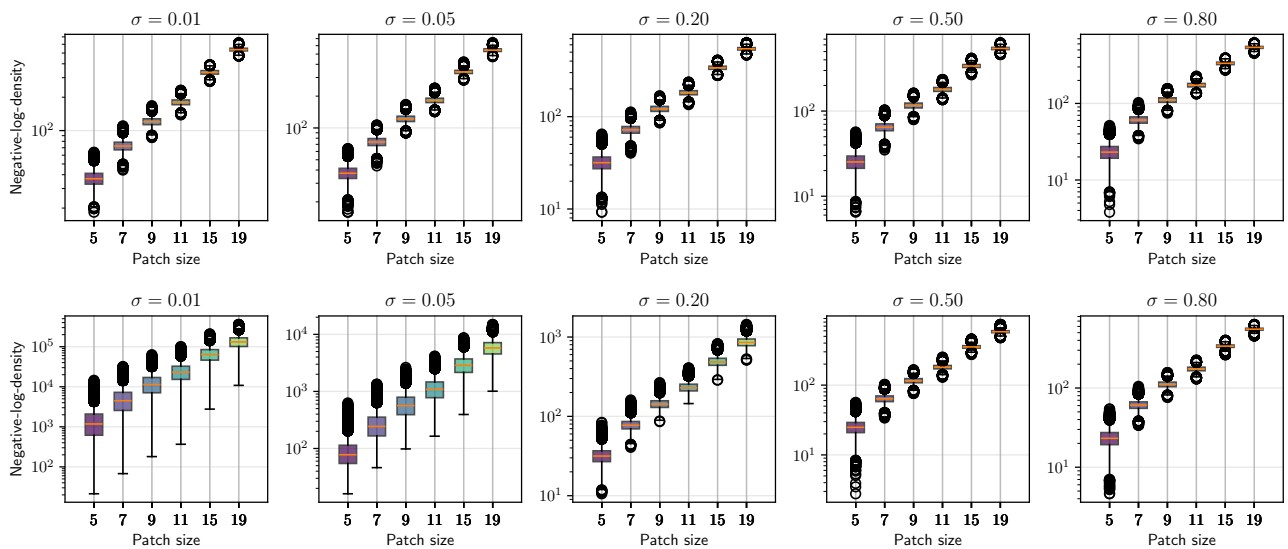

*Figure 19.* The negative-log-density of patches in FFHQ-32 training set (top) and test set (bottom) as a function of the patch size. As the patch size increases, the density of patches decreases significantly, indicating a sparser coverage of the patch space by the dataset. In the training set, the negative-log-density is relatively lower (around $10^2$) than in the test set (around $10^4$ for small noise levels), indicating a better coverage of the patch space by the training set. This observation helps explain the drop in PSNR between trained neural networks and the analytical LE-MMSE estimator in the test set for low noise levels.

**Patch size influence** In Figure 20, we show the PSNR between the trained UNet2D and the analytical LE-MMSE estimator for different patch sizes on both training and test sets of FFHQ-32 dataset. For low noise levels and in the test set, the PSNR decreases as the patch size increases, which we attribute to the sparser coverage of the patch space by the dataset for larger patches, as shown in Figure 19.

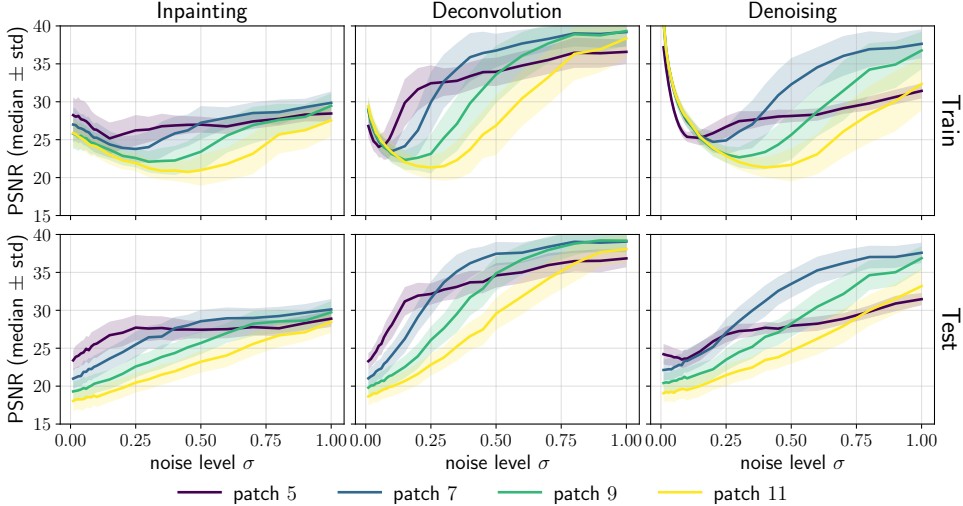

*Figure 20.* PSNR between trained UNet2D and the analytical formula of the LE-MMSE versus the patch size, on the FFHQ-32 dataset. Top: on the training set. Bottom: on the test set.

In Figure 21, we show the PSNR between the analytical LE-MMSE estimator and the ground truth as a function of the patch size on both training and test sets of FFHQ-32 dataset. We observe that – on the test set – smaller patch size are preferable for low noise levels, while larger patch sizes yield better performance for high noise levels. The behavior on the training set is different: for low noise levels, the analytical formula copy-pastes exactly the right patches, explaining the blow up of PSNR at the origin.

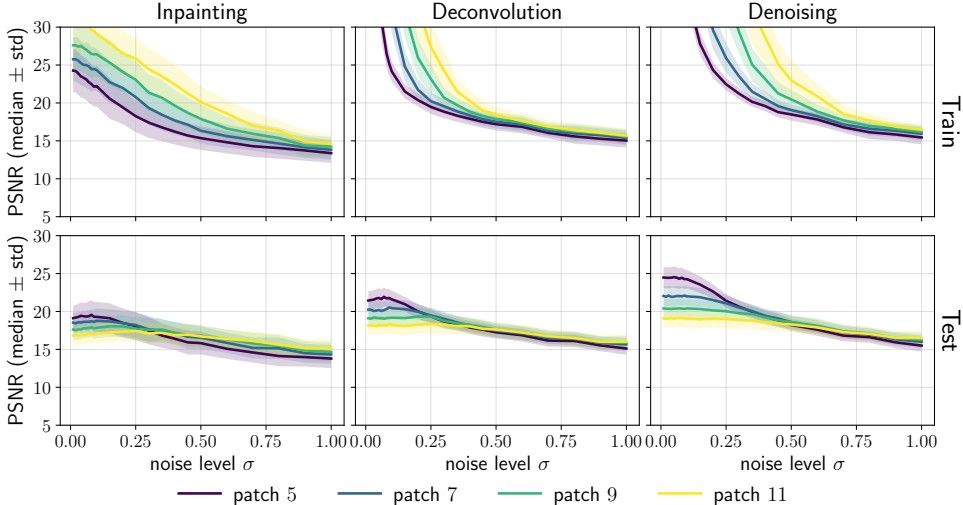

*Figure 21.* PSNR between the analytical formula of the LE-MMSE and the ground truth versus the patch size, on the FFHQ-32 dataset. Top: on the training set. Bottom: on the test set.

### D.4. Mass concentration

The LE-MMSE estimator Theorem 3.8 is a weighted average of the central pixel values of patches in the dataset. To understand the behavior of the estimator, we analyze how many patches contribute significantly to the estimate at each pixel location. In Figure 22, we show the number of patches contributing to 99% of the mass of the LE-MMSE estimator as a function of the noise level $\sigma$. We observe that for low noise levels, the estimator concentrates its mass on fewer patches (even nearest neighbors). As the noise level increases, the number of contributing patches increases significantly, with a critical value of $\sigma$ where the number of patches starts to increase rapidly. This behavior shows that although the MMSE estimator is typically associated with averaging, it can behave like a nearest-neighbor estimator when the noise is small, due to the strong concentration of mass on a very limited subset of patches. We also observe a clear difference between physics-agnostic and physics-informed settings for deconvolution, where the latter has a significantly higher number of contributing patches due to the amplification of noise by the pseudo-inverse of the blurring operator. We also visualize

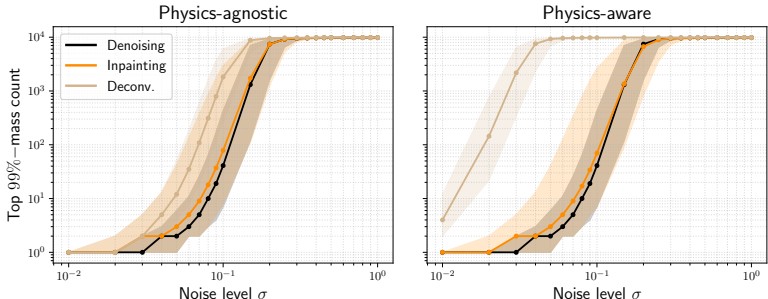

*Figure 22.* For low noise levels, the LE-MMSE estimator concentrates its mass on fewer patches (even nearest neighbors). Here we show the median and IQR (over all pixels of 50 samples on the test set, for each $\sigma$) of the number of patches contributing to 99% of the mass of the LE-MMSE estimator. There is a significant increase in number of contributed patches as the noise level $\sigma$ increases, and a critical value of $\sigma$ where the number of patches starts to increase rapidly. The patch size is $P = 5 \times 5$, on FFHQ-32. Note that due to computational constraints, we only keep the top $10^4$ nearest patches (over $32 \times 32 \times 10^4 \approx 10^7$ patches) when computing mass concentration, but the estimator is still exact as we accumulate all the mass in an online manner.

in Figures 23 and 24 the $\log_{10}$ of number of patches contributing to 99% of the mass of the LE-MMSE estimator at each pixel location for different inverse problems on FFHQ-32 dataset. We observe that for low noise levels, the nearest patch (or a very small number of patches) contributes 99% of the mass at each pixel location.

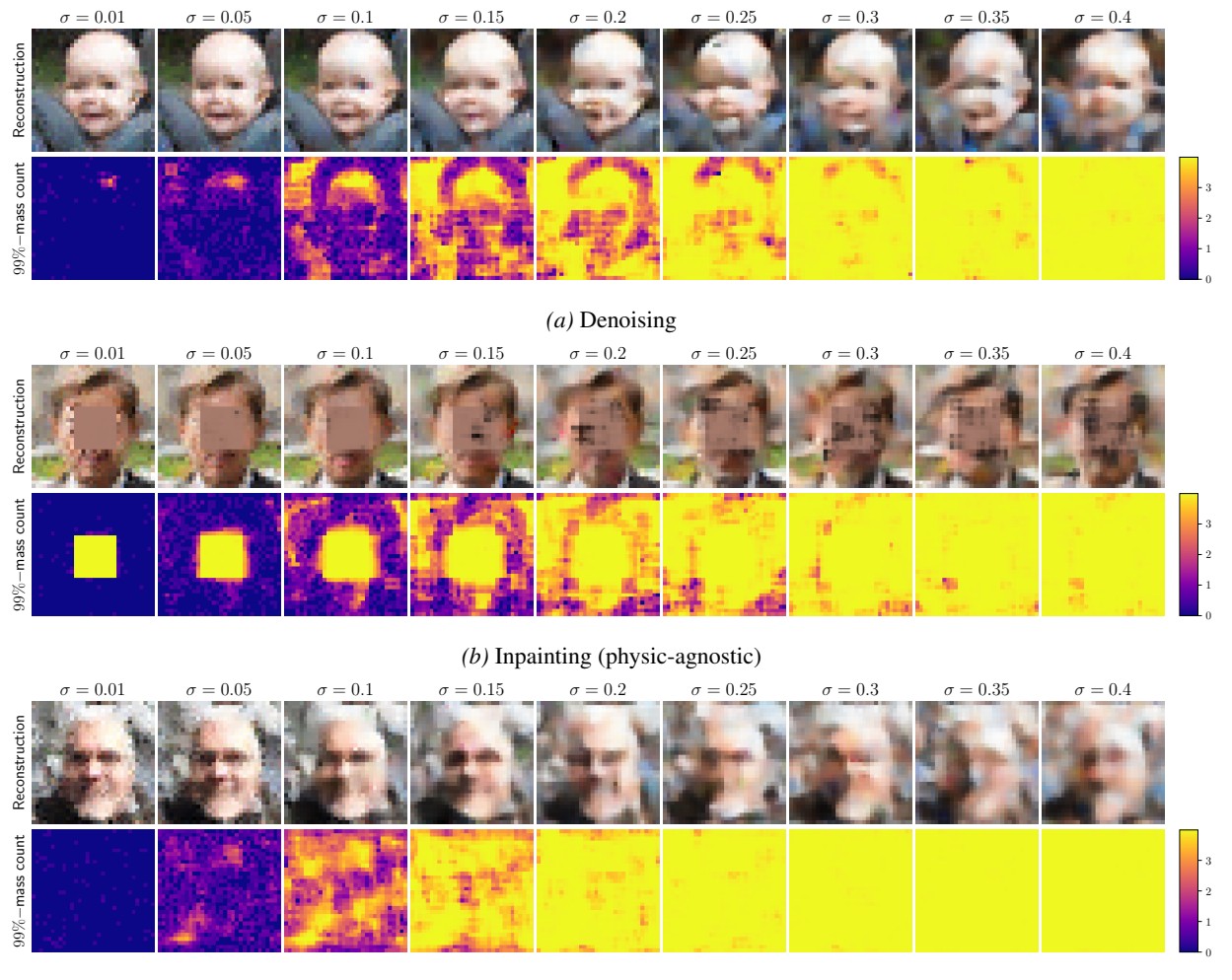

*(a)* Denoising

*(b)* Inpainting (physic-agnostic)

*(c)* Deconvolution (physic-agnostic)

*Figure 23.* Visualization of the number of patches (in $\log_{10}$ scale) contributing to 99% of the mass of the physic-agnostic LE-MMSE estimator at each pixel location for different inverse problems on FFHQ-32 dataset.

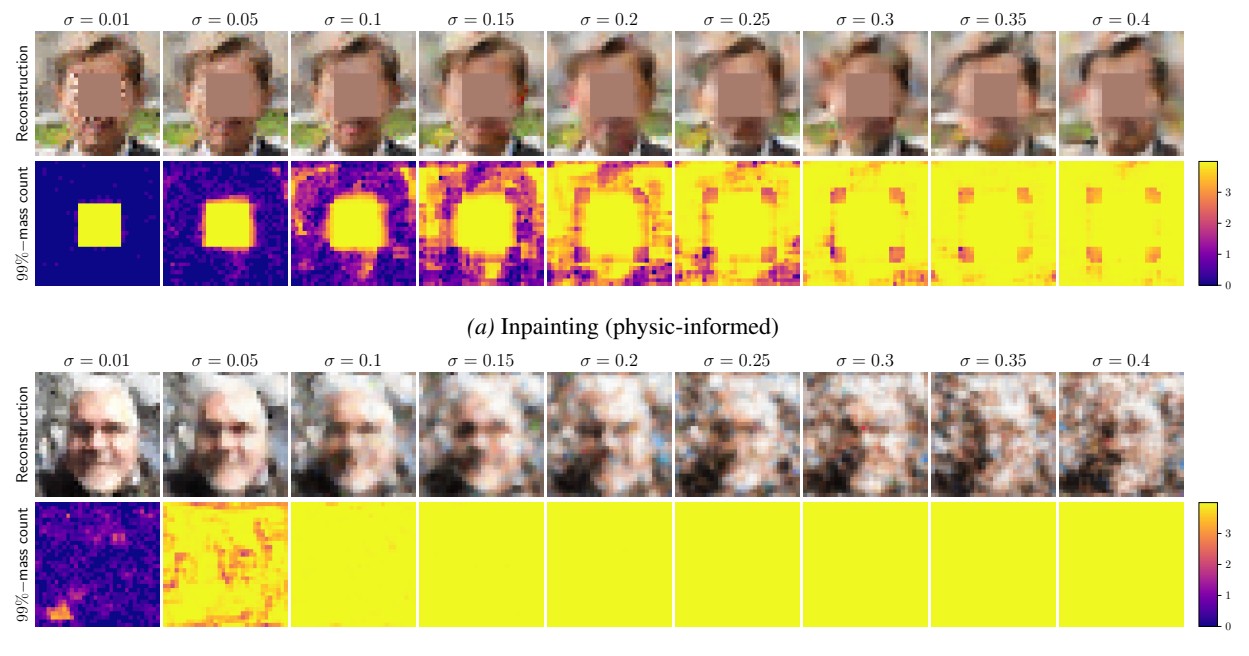

*(a)* Inpainting (physic-informed)

*(b)* Deconvolution (physic-informed)

*Figure 24.* Visualization of the number of patches (in $\log_{10}$ scale) contributing to $99\%$ of the mass of the physic-inform LE-MMSE estimator at each pixel location for different inverse problems on FFHQ-32 dataset. For deconvolution, the number of contributing patches is significantly higher than for the physic-inform case, due to the amplification of noise by the pseudo-inverse of the blurring operator.

### D.5. LE-MMSE is a patchwork

From the LE-MMSE formula Theorem 3.8, we see that the estimate at each pixel location is a weighted average of the central pixel values of patches in the dataset. In fact, the estimator often concentrates its mass on very few patches, as shown in Appendix D.4. Moreover, contiguous pixels may come from the central pixels of patches of the same image in the dataset, leading to a patchwork behavior.

We visualize in Figures 25 and 26 this behavior for different inverse problems on FFHQ-32 and FashionMNIST datasets. More precisely, at each pixel location, the source index is the index of the image in the dataset that contains the patch whose central pixel contributes at least $50\%$ of the mass of the LE-MMSE estimator at that pixel location. We observe that, large contiguous regions of the estimate come from the same source image in the dataset, which we can interpret as the estimator to be a patchwork of training patches. As the noise level increases, more pixels become white, meaning that the corresponding pixel value is not mostly due to a single patch: the "patchwork effect" diminishes, as more patches contribute to the estimate at each pixel location.

Note that we use a patch size of $P = 11 \times 11$ to better visualize the patchwork behavior. This choice results in slightly weaker reconstruction performance, as discussed in Appendix D.3.

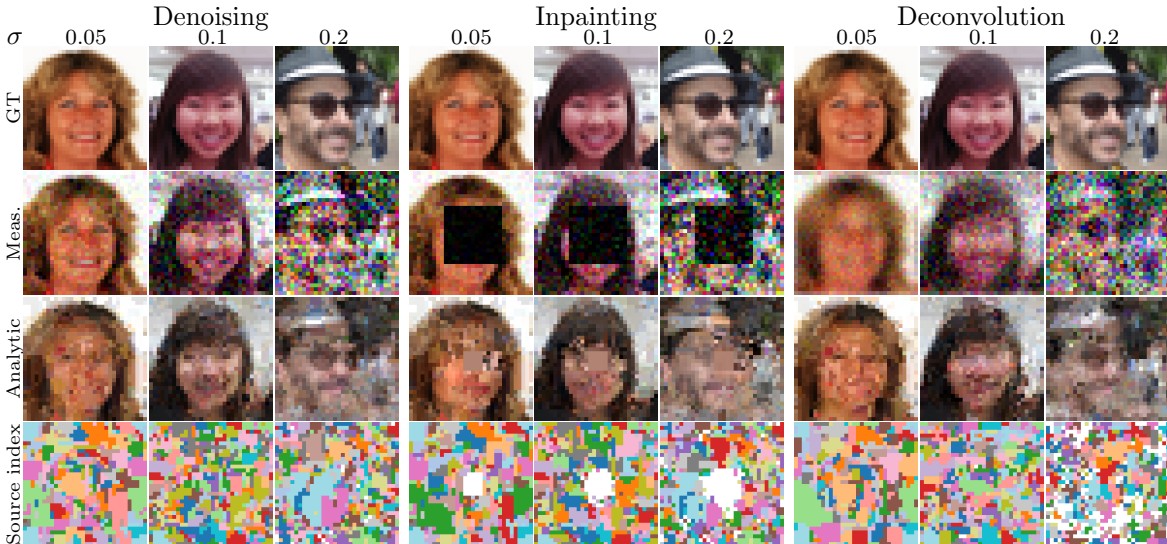

*Figure 25.* Illustration of the patchwork behavior of the LE-MMSE estimator (with $B = \mathrm{I}$) on FFHQ-32 dataset for different inverse problems and noise levels. We use a patch size of $P = 11 \times 11$. We observe contiguous regions coming from the same source image in the dataset for low noise levels. White pixels correspond to locations where no patch contributes more than $50\%$ of the mass.

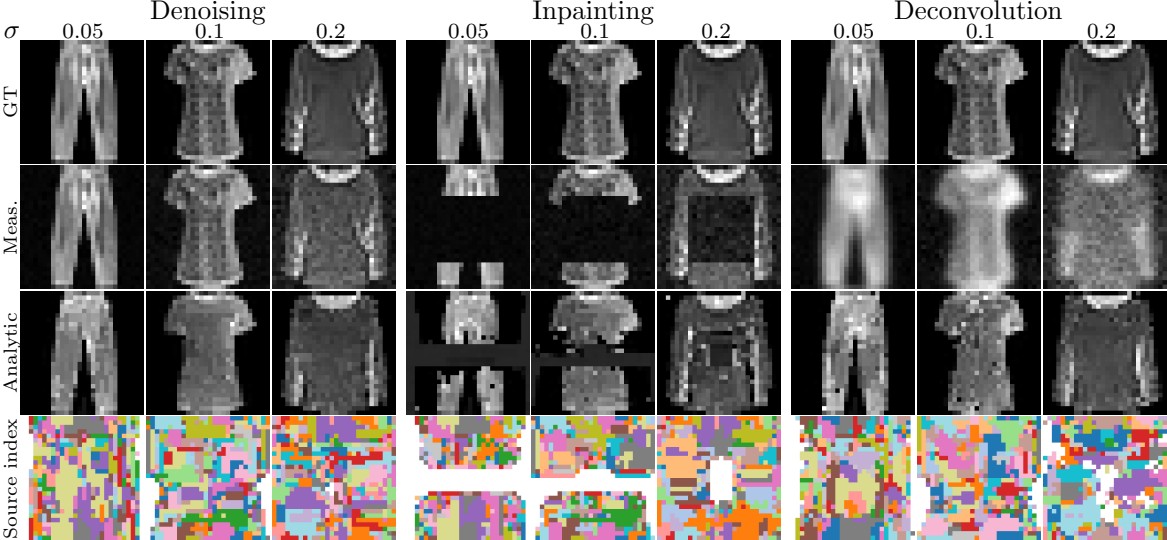

*Figure 26.* Similar illustration as in Figure 25 but on FashionMNIST dataset.

### D.6. Out-of-distribution data

We provide additional results on out-of-distribution (OOD) data using UNet2D architecture with receptive field of size $5 \times 5$. The models and the formula are trained / evaluated on $\mathcal{D} = $ FFHQ-32. They are then evaluated on OOD dataset $\mathcal{D}' = $ CIFAR10. We show in Figure 27 the PSNR between trained UNet2D and the analytical LE-MMSE estimator. For large noise levels, as the Gaussians overlap significantly even on OOD data, the value of $-\log p(y)$ is low and the trained neural network closely matches the analytical LE-MMSE estimator with PSNR values above 25 dB. For low noise levels, the PSNR is about 3 dB lower than in the in-distribution case shown in Figure 5, which we attribute to the low-density of the measurement density $p(y)$, as discussed in Section 4.2 and can be seen in Figure 27.

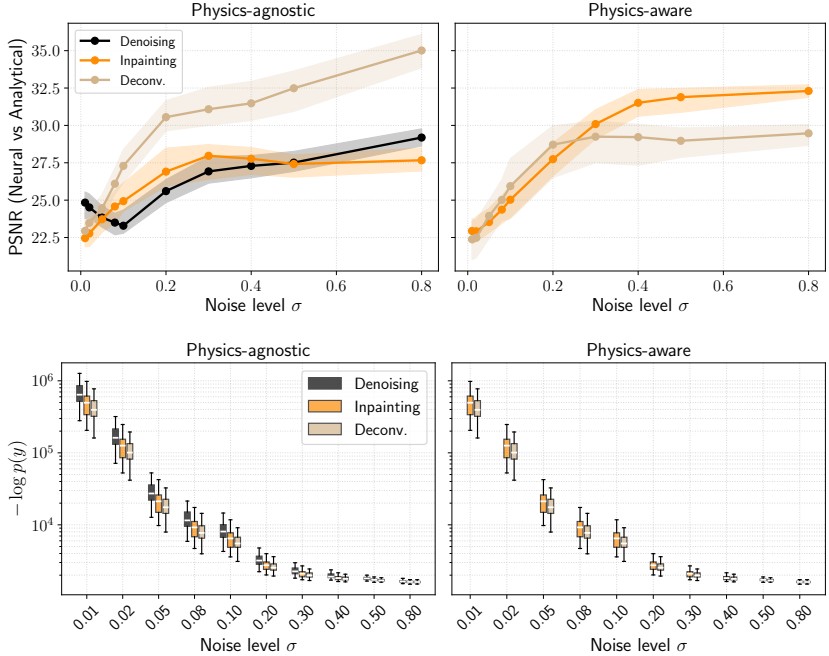

*Figure 27.* Comparison of UNet2D and the analytical LE-MMSE estimator on OOD dataset CIFAR10 when both are trained on FFHQ-32. Median and IQR using 50 images per $\sigma$, $P = 5 \times 5$ and $B = I$. On top: PSNR between UNet2D and the analytical LE-MMSE estimator. On bottom: negative-log-density of measurements $y$ in CIFAR10 under the measurement distribution induced by FFHQ-32.

## D.7. Dataset size influence

We analyze here the influence of the dataset size on the alignment between trained neural networks and the analytical LE-MMSE estimator. We train UNet2D models with receptive field of size $P = 5 \times 5$ and $B = I$ on various dataset sizes from $10^3$ to $5 \times 10^4$ images from FFHQ-32. The PSNR between trained UNet2D and the analytical LE-MMSE estimator is reported in Figure 28. We observe that the dataset size has limited influence on the alignment between trained neural networks and the analytical LE-MMSE formula, with a slight improvement when increasing the dataset size.

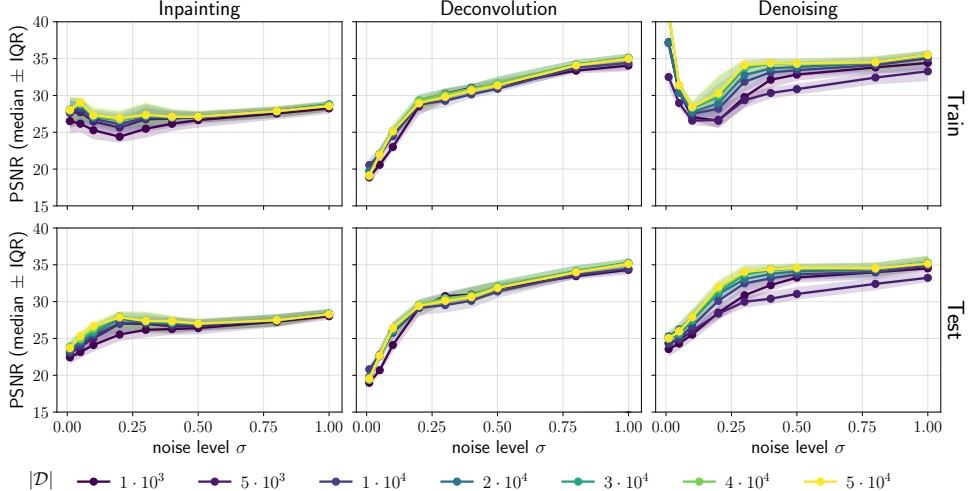

*Figure 28.* Dataset size has limited influence on the alignment between neural networks and the LE-MMSE formula. Median and IQR using 50 images per $\sigma$, $P = 5 \times 5$ and $B = I$.

## D.8. Results on $3 \times 64 \times 64$ images

We provide additional results on images at $3 \times 64 \times 64$ resolution using UNet2D architecture with receptive field of size $11 \times 11$. The models are trained on $10^4$ images from FFHQ downscaled to $64 \times 64$, with the same training procedure as in Appendix C.3.

*Table 5.* Details of neural network architectures with various receptive fields used in our experiments for images at $3 \times 64 \times 64$ resolution.

| | RECEPTIVE FIELD (PATCH SIZE) | ARCHI. HYPER-PARAMETERS | | NUM. PARAMETERS |
|---|---|---|---|---|
| UNET2D | | `block_out_channels` | `kernel_size` | |
| | 11 | $(64, 128, 256, 512)$ | $(3, 1, 1, 1, 3){-}1$ | 13.3M |

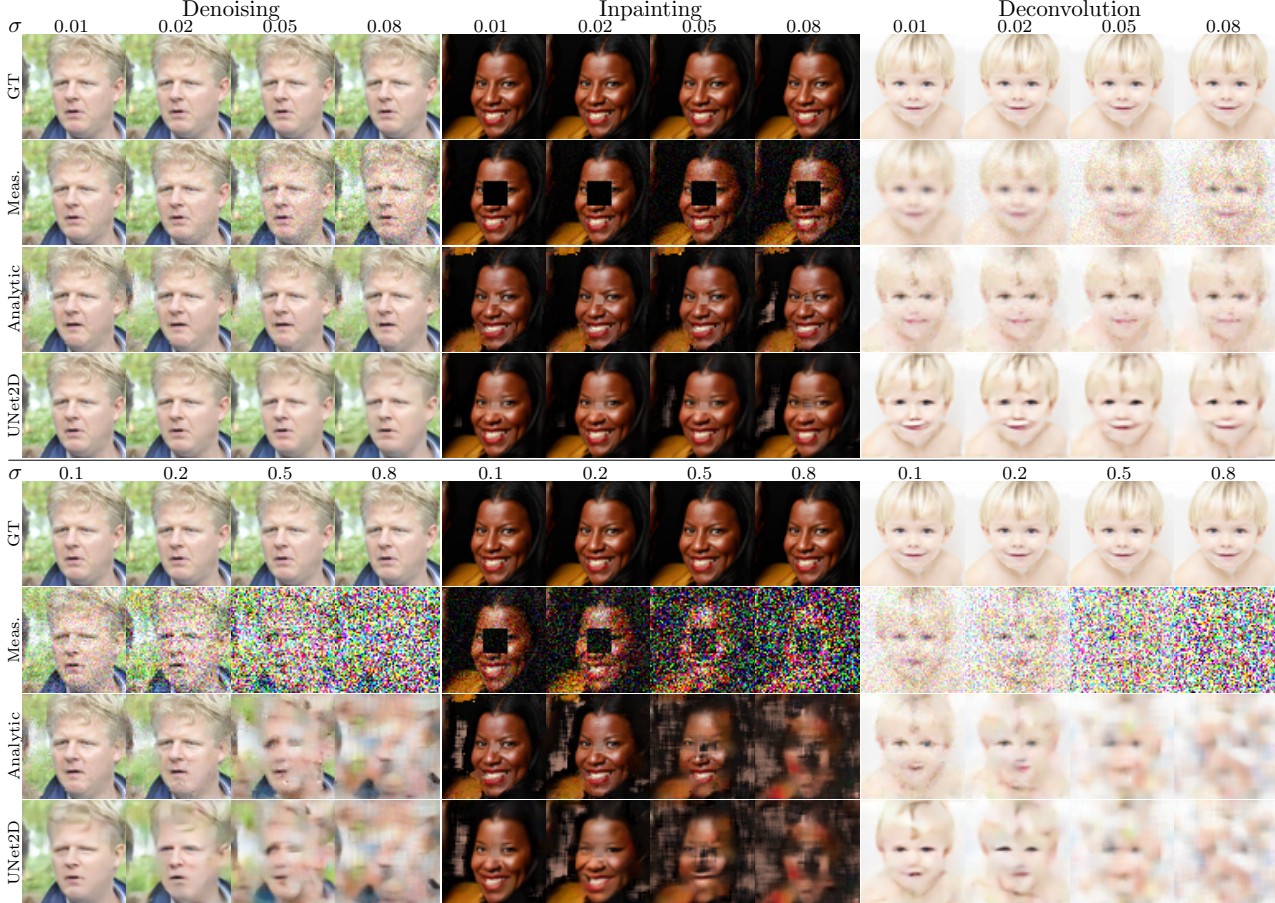

*Figure 29.* Additional qualitative comparison between UNet2D and the analytical LE-MMSE estimator on FFHQ-64 across tasks.

## D.9. Results on $3 \times 128 \times 128$ images

We provide additional results on 3000 color images of size $128 \times 128$ from the FFHQ dataset ($\sim 50$ million patches) using ResNet architecture with receptive field of size $P = 11 \times 11$, with the same parameters as in Table 5 and training procedure as in Appendix C.3. We observe that on high-density regions (train, large noise), the LE-MMSE and ResNet have a good adequation with a slight drop in low-density region (test, small noise).

*Table 6.* The alignment between trained ResNet and the analytical LE-MMSE estimator on FFHQ-128.

|  | **Train** | | **Test** | |
| --- | --- | --- | --- | --- |
|  | Denoising | Deconvolution | Denoising | Deconvolution |
| $\sigma$ | 0.05/0.5 | 0.05/0.5 | 0.05/0.5 | 0.05/0.5 |
| PSNR ↑ | 30.13/27.02 | 26.99/30.04 | 25.03/29.33 | 24.72/30.83 |
| SSIM ↑ | 0.92/0.91 | 0.86/0.94 | 0.76/0.93 | 0.75/0.95 |
| LPIPS ↓ | 0.027/0.084 | 0.089/0.060 | 0.118/0.057 | 0.175/0.047 |

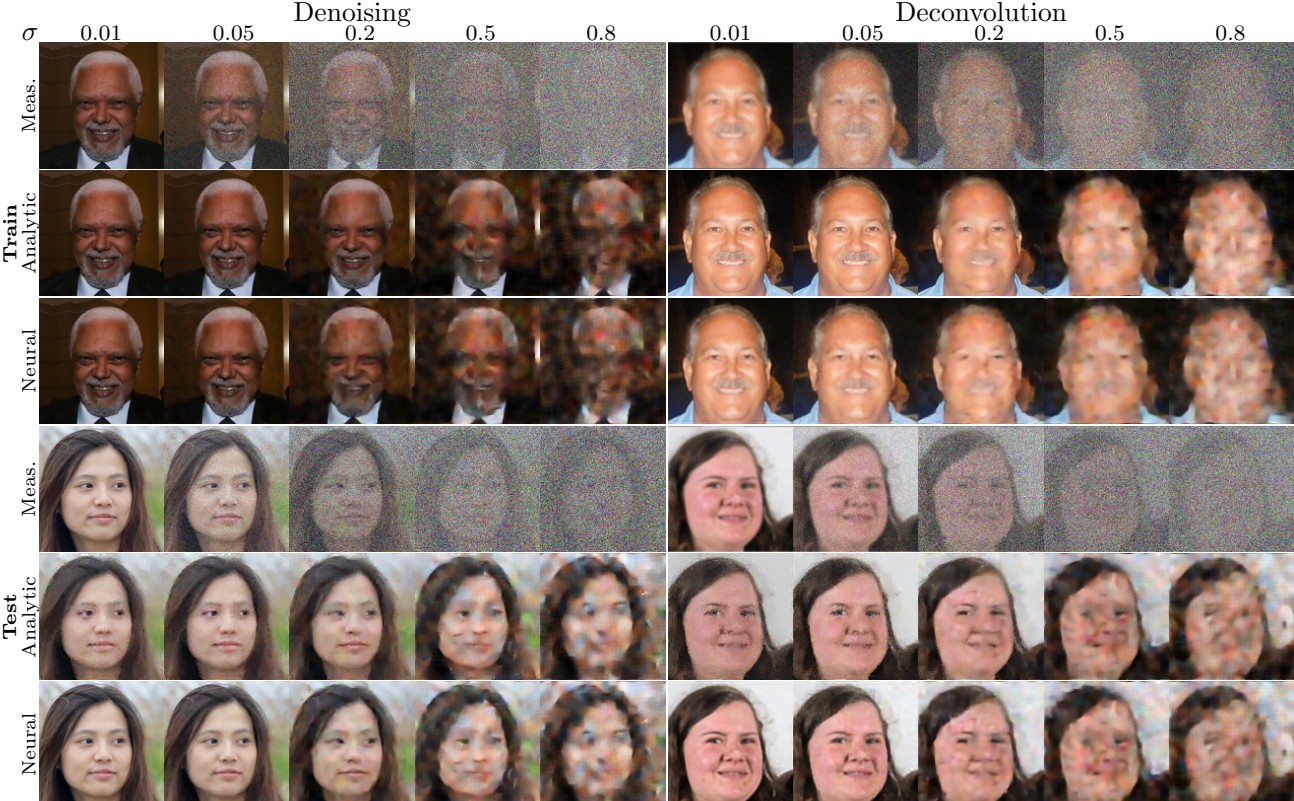

*Figure 30.* Additional qualitative comparison between ResNet and the analytical LE-MMSE estimator on FFHQ-128.

