# OpenReview forum: "An analytic theory of convolutional neural network inverse problems solvers"
_ICML.cc/2026/Conference — ICML 2026 regular_

### Official Review · Reviewer_U87F · 2026-03-05

**Soundness:** 2
**Presentation:** 3
**Significance:** 2
**Originality:** 2
**Overall Recommendation:** 5
**Confidence:** 3

**Summary:**

The paper studies supervised CNNs on imaging inverse problems and analyses estimators that generalize MMSE estimators.  The results are tested for multiple inverse problems and studies the  differences between physics-aware and  physics-agnostic estimation (that in this paper refer to a pre-conditioner operator that is either the identity or a pseudo-inverse of the direct operator).

**Compliance With Llm Reviewing Policy:**

Affirmed.

**Final Justification:**

The authors have answered well to my questions. My main concern was the possible use of Gaussian prior model, and this issue has now been answered. I also studied the experimental results the authors have included in the answers to the other reviewers and those are impressive. Due to this, I have raised my score.

**Key Questions For Authors:**

1.  The operator $\phi^\star_M$ in formula (3) contains map B, but above one considers $\hat x_M=\phi_M^\star B$  also contains $B$. Why $B$ appears twice here, but only once in formula (2)?

2. In contributions, item 2, one comments arbitrary inverse problems even though the solution algorithms considered in the paper are equivariant. Does the equivarance of the solution algorithm require that the measurement model is also equivariant?

3. In Propostion 2.1 you seem to use an implicit way a Gaussian prior. If this is so, could you please add more details on this?

**Limitations:**

Yes

**Strengths And Weaknesses:**

Soundness: The analysis is not very clear

Presentation: The submission well written

Significance: Main theoretical contributions are not very strong.

Originality: The work seem to study quite classical ideas

---

> ### Author Rebuttal · Authors · 2026-03-28
>
> We thank the reviewer for their time and effort to evaluate our paper, and we are sorry to hear about the negative assessment. We address the specific points raised by the reviewer below.
>
> > *The analysis is not very clear. Main theoretical contributions are not very strong. The work seem to study quite classical ideas.*
>
> We appreciate the reviewer's time, but we respectfully disagree with the assessment that our theoretical contributions rely solely on classical ideas.
> While MMSE estimation is indeed a well-established field, our specific contribution is highly novel: we derive a closed-form LE-MMSE estimator that explicitly incorporates modern CNN inductive biases (translation equivariance and locality) specifically for linear inverse problems. Surprisingly, the analytical formula derived have tight links with state-of-the-art handcrafted techniques for denoising such as nonlocal means, but extend to far more general  inverse problems.
> This represents a substantial, mathematically nontrivial extension of the recent framework introduced by Kamb & Ganguli (2025), providing a previously unavailable, principled explanation for why supervised CNNs achieve state-of-the-art results in imaging inverse problems.
> We genuinely want to ensure our analysis is as clear as possible. Because we firmly believe these results significantly advance the field, we would greatly appreciate it if you could specify which particular parts of the analysis require clarification, or which specific derivations appear classical. We are fully prepared to provide detailed clarifications during this rebuttal period.
>
>
> ### Key Questions
>
> > *In formula (3), the operator includes map B while B also appears in the preceding definition; why does B appear twice in formula (3) but only once in formula (2)?*
>
> We are sorry to hear about the confusion. To clarify, equation 2 defines the structure of the neural network estimator: $\hat{x}(y) = N_w(By)$. Equation 3 defines the constrained optimization problem to find the optimal function (neural network) $\phi \in \mathcal{M}$. The estimator then reads $\hat{x}(y) = \phi(By)$. Moreover, the operator $B$ appears only once in equation (3).
>
> > *In contribution item 2, arbitrary inverse problems are discussed while solution algorithms are equivariant; does equivariance of the solution algorithm require the measurement model to also be equivariant?*
>
> To clarify, the formula derived in the paper do not require the measurement model to be equivariant, it can be arbitrary. However, the reconstruction equivariance (equation 5) holds true if $A$ and $B$ are circular convolution (see Corollary 3.7).
>
> > *In Proposition 2.1, is a Gaussian prior being used implicitly? If so, could you provide more details?*
>
> Proposition 2.1 does not assume a Gaussian prior on the image $x$. The prior on $x$ is the empirical data distribution $p_\mathcal{D}$ (a normalized sum of Dirac masses over the training set). The Gaussian term $\mathcal{N}$ arises entirely from the likelihood function $p(y|x)$, which is Gaussian because our forward model (Equation 1) assumes additive white Gaussian noise.

---

> > ### Author Rebuttal · Reviewer_U87F · 2026-04-03
> >
> > The authors have answered well to my questions and presented impressive new experiments in other responses for reviewers.

---

> > > ### Author Response · Authors · 2026-04-03
> > >
> > > We would like to thank the reviewer for your positive feedback and for confirming that your concerns have been fully resolved. We are glad that our responses and the additional experiments provided the clarity you were looking for.
> > >
> > > We sincerely appreciate your support and your score adjustment.

---

### Official Review · Reviewer_6bxG · 2026-03-13

**Soundness:** 3
**Presentation:** 3
**Significance:** 3
**Originality:** 4
**Overall Recommendation:** 5
**Confidence:** 3

**Summary:**

This paper offers a theoretical view of CNNs for imaging inverse problems. It interprets them as approximating the MMSE estimator under an empirical prior derived from the training data. By incorporating two core CNN inductive biases, translation equivariance and local receptive fields, the authors derive a closed form estimator called the Local Equivariant MMSE, which reconstructs each pixel as a likelihood weighted average of local patches from the training set. The performance of the proposed estimator matches that of various CNN-based architectures on denoising, deconvolution, and inpainting tasks.

**Compliance With Llm Reviewing Policy:**

Affirmed.

**Final Justification:**

I have increased my recommendation to accept, as the authors were able to conduct supplementary experiments on higher-resolution images and on a different task (MR reconstruction) within the short rebuttal period. They also clarified details regarding their experimental setup and provided insightful discussion on different inductive biases. This strengthens the paper and suggests promising potential for future extensions, particularly in analyzing the inductive biases of attention-based architectures for inverse problem solvers.

**Key Questions For Authors:**

1. Could you evaluate the method on images with larger spatial resolutions, for example \$256 \times 256\$ color images?
2. Could you also report structural similarity (SSIM) results for selected experiments to examine whether the analytical estimator and neural network methods achieve comparable SSIM performance?

**Limitations:**

yes

**Strengths And Weaknesses:**

**Strengths**:
1. The proposed estimators are an interesting contribution, especially for analyzing CNN-based inverse problem solvers. This is relevant not only to machine learning but also to medical imaging, where the method could potentially be used to identify sources of bias in medical image quality enhancement methods.
2. The paper is generally well written, and the ideas are easy to follow.

**Weaknesses**:
1. While the estimators match the performance of various CNNs, many recent state-of-the-art methods for image restoration are based on transformers, which typically lack strong inductive biases.
2. Since the approach has only been tested on images of size $32 \times 32$ or $64 \times 64$, it would be important to evaluate it on datasets with larger spatial resolutions.
3. During evaluation, relying solely on PSNR may not be sufficient. Including additional metrics such as SSIM and LPIPS is recommended.

**Some Other Points of Consideration**:
1. At the end of Section 1 ($y = A\bar{x} + e$), I assume that $y, \bar{x}$ and $e$ are vectors, but they are not written in bold. In addition, matrix variables are typically written in bold capital letters, which would improve notational consistency.

---

> ### Author Rebuttal · Authors · 2026-03-28
>
> We thank the reviewer for their time and effort to evaluate our paper, we also thank the reviewer for the positive evaluation and for highlighting the relevance of our theoretical framework to broader fields like medical imaging and appreciate the constructive suggestions regarding notation, evaluation metrics, and spatial resolution, all of which will certainly strengthen the final manuscript. We address your main concerns below.
>
> > *While the estimators match the performance of various CNNs, many recent state-of-the-art methods for image restoration are based on transformers, which typically lack strong inductive biases.*
>
> We thank the reviewer for raising this important context regarding current state-of-the-art architectures. We completely agree that many recent leading methods for image restoration are based on Vision Transformers, which typically discard strict local receptive fields in favor of global attention mechanisms, which is *permutation-equivariant* (assuming no positional embedding).
>
> However, we believe our work remains highly relevant to the community for several reasons:
>   1. Theoretical explanation: our goal is fundamental understanding of how and why CNNs succeed in linear inverse problems. The LE-MMSE provides a rigorous mathematical explanation for the performance of CNNs, which is valuable in its own right.
>   2. Highly successful modern models frequently reintroduce the very biases we study. For example, Swin Transformers enforce strict locality via windowed attention, and ConvNeXt demonstrates that purely convolutional architectures remain highly competitive.
>
> Please also refer to the response to reviewer **jeRx** for a detailed discussion on how the LE-MMSE framework could potentially be extended to architectures with non-local interactions, and the technical challenges involved in doing so.
>
> We will add a brief discussion to the revised manuscript explicitly contextualizing our theoretical contributions within the broader landscape of Transformer architectures.
>
>
> > *Evaluation on larger spatial resolutions*
>
> We thank the reviewer for this suggestion.  We have trained a ResNet with a receptive field of 11x11 on 3000 color images of size 128x128 from the FFHQ dataset (~50 million patches), for denoising and deconvolution as representative examples. We report the metrics between the ResNet and LE-MMSE output below. We observe that on high-density regions (train, large noise), the LE-MMSE and ResNet have a good adequation with a slight drop in low-density region (test, small noise).
> $$
> \\begin{array}{l|c c|c c}
> \\hline
> & \\textbf{Train} &  & \\textbf{Test} \\\\
> \\hline
> & \\text{Denoising} & \\text{Deconv.} & \\text{Denoising} & \\text{Deconv.} \\\\
> \\sigma & 0.05 / 0.5 & 0.05 / 0.5 & 0.05 / 0.5  & 0.05 / 0.5  \\\\
> \\hline
> \\text{PSNR}  & 30.13 / 27.02 & 26.99 / 30.04 & 25.03 / 29.33 & 24.72 / 30.83 \\\\
> \\text{SSIM}  & 0.92 / 0.91 & 0.86 / 0.94 & 0.76 / 0.93 & 0.75 / 0.95 \\\\
> \\text{LPIPS} & 0.027 / 0.084 & 0.089 / 0.060 & 0.118 / 0.057 & 0.175 / 0.047 \\\\
> \\hline
> \\end{array}
> $$
> (*Note: comprehensive metrics will be included in the final version*)
>
> In addition, we have conducted an experiment on $4\\times$ accelerated MRI reconstruction with masking and also found a good alignment between the ResNet and LE-MMSE estimator, showcasing the relevance of our framework for scientific inverse problems. Please refer to the response to reviewer **jeRx** for details.
> > *Could you also report structural similarity (SSIM)*?
>
> We computed both SSIM and LPIPS between the ResNet output and the LE-MMSE output, for the same setting as Table 2 in the main paper. The metrics are shown below. (Format: **SSIM / LPIPS**, where higher SSIM and lower LPIPS are better).
> $$
>     \\begin{array}{ll c c c}
>     \\hline
>     \\text{Model} & \\text{Split} & \\text{Denoising} &\\text{Inpainting} & \\text{Deconvolution} \\\\
>     \\hline
>     \\text{FFHQ-32} & \\text{Train} & 0.94/0.003  & 0.86/0.011 & 0.82/0.035 \\\\
>     & \\text{Test} & 0.87/0.009 & 0.82/0.013 & 0.86/0.018 \\\\
>     \\hline
>     \\text{CIFAR10} & \\text{Train} & 0.93/0.004 & 0.85/0.021 & 0.84/0.041 \\\\
>     & \\text{Test} & 0.88/0.007 & 0.81/0.019 & 0.86/0.026 \\\\
>     \\hline
>     \\end{array}
> $$
> *(Note: for brevity, we only show for $\\sigma=0.05$, more comprehensive evaluation will be included in the appendix).*
>
> We observe that the conclusions remain unchanged when using these metrics. This further strengthens the evidence of good agreement between the neural and LE-MMSE beyond just PSNR.
>
> > *Notation consistency concern: vectors and matrices should be consistently bolded.*
>
> We apologize for the misleading notation and thank the reviewer for the feedback. Indeed, in the paper, bold lowercase letters denote random vectors, while normal lowercase letters denote their realizations (also vectors). This is detailed in the appendix; we will move it in the main text in the final version of the paper.

---

> > ### Author Rebuttal · Reviewer_6bxG · 2026-04-05
> >
> > The authors have adequately addressed my concerns and proposed additional supplementary experiments to support their findings. Great work.

---

> > > ### Author Response · Authors · 2026-04-05
> > >
> > > We sincerely thank the reviewer for their positive feedback and for acknowledging the efforts made during the rebuttal process. We are glad that the additional experiments and clarifications addressed your concerns. We also truly appreciate you adjusting your score to reflect this; your support is very meaningful to the final quality of our work.

---

### Official Review · Reviewer_jeRx · 2026-03-13

**Soundness:** 4
**Presentation:** 4
**Significance:** 3
**Originality:** 3
**Overall Recommendation:** 5
**Confidence:** 4

**Summary:**

This paper analyzes trained neural networks using Minimum Mean Square Error (MMSE) estimators.  Specifically, the authors start from reviewing the theoretical analysis of unconstrained MLP, for which MMSE estimator is a proxy.  Then, the authors extends the analysis to MMSE variants including E-MMSE and LE-MMSE, which captures the translation equivariance (E) and locality (L) of CNNs.  Finally, the theoretical analysis is validated using various numerical tests.

**Compliance With Llm Reviewing Policy:**

Affirmed.

**Final Justification:**

This is a strong theoretical paper that gives a principled analytic account of how CNN inverse-problem solvers behave, with a convincing formula-to-network agreement, though its empirical scope and architectural coverage remain relatively narrow. On balance, I end at an accept.

**Key Questions For Authors:**

1.  The numerical examples considered in the paper are standard natural image restoration tasks like denoising, inpainting, and deconvolution.  I'm wondering how well the LE-MMSE interpretation transfers to scientific inverse problems where the forward map and data distribution are more complex.  I'm not sure if the current set of numerical results are sufficient to illustrate the theoretical analysis.
2. I think that UNet2D is not exactly translation equivariant.  If the input translated by one entry, then the downsampled coarse layers take average on different set of entries.  I'm wondering if the wording in the first paragraph of 4.1. Experimental setup should be changed to approximately tranlation equivariant?
3. How well does the proposed LE-MMSE theory extend beyond strictly local translation-equivariant CNNs to more modern architectures with nonlocal interactions?  For example, if one adds an attention block at the coarsest level of the UNet2D, then should one expect the performance to change a lot?

**Limitations:**

Yes

**Strengths And Weaknesses:**

Strength:
1. This paper provides a clear and thorough theoretical interpretation of CNNs for linear inverse problems by connecting CNN's locality and translation equivariance to a closed-form LE-MMSE estimator.
2. This paper has a very clear structure and is written with polished language.
3. This paper offers useful conceptual insight into various topics including the benefit of using physics-aware estimator, the data augmentation interpretation, hyperparameter selection and extensions beyond current assumptions.

Weakness:
1. Limited algorithmic novelty; the main contribution is interpretive rather than a new method.
2. Empirical results are somewhat limited in scale and realism as it only touches natural image restoration.

---

> ### Author Rebuttal · Authors · 2026-03-30
>
> We sincerely thank reviewer **jeRx** for the strong support of our work and for recognizing the value of the conceptual insights provided by our framework. We greatly appreciate your constructive feedback and address your points below.
>
> > *Limited algorithmic novelty; main contribution is interpretive rather than proposing a new method.*
>
> We agree that our primary contribution is interpretive rather than algorithmic.  As noted in your assessment of the paper's strengths, our core objective was to provide a thorough theoretical interpretation of how and why CNNs succeed in linear inverse problems.
> We believe that explainability is a critical challenge in deep learning today. By offering a principled, closed-form explanation for CNNs, our framework not only demystifies the current state-of-the-art but also provides a concrete foundation that can guide the design of future, physics-aware architectures.
>
> Interestingly, the resulting estimator reveals strong connections to classical methods such as **non-local means**, showing that modern CNNs can be interpreted as structured, data-adaptive generalizations of these approaches. We will emphasize this connection more clearly in the final version.
>
>
> > *Empirical results are somewhat limited in scale and realism, focusing on standard natural image restoration tasks. How well does the LE-MMSE interpretation transfer to scientific inverse problems with more complex forward maps and data distributions?*
>
> We agree that extending beyond standard image restoration tasks is important. To address this, we conducted additional experiments on a scientific inverse problem: $4\\times$ accelerated MRI reconstruction, with forward model $A = SF$ (subsampled Fourier) and pre-inverse $B = F^{-1}$.
>
> We trained a ResNet (receptive field $11 \\times 11$) on the *fastMRI knee_singlecoil_train* dataset (4865 slices, $128\\times128$). The alignment between the neural network and the LE-MMSE estimator is summarized below:
>
> $$
> \\begin{array}{l|cccccc}
> \\hline
>     \\text{Train (Neural vs Analytical)} & \\sigma=0.01 & \\sigma=0.02 & \\sigma=0.05 & \\sigma=0.1 & \\sigma=0.2 & \\sigma=0.3 \\\\
> \\hline
>     \\text{PSNR}\\uparrow & 32.88 & 32.85 & 32.74 & 32.52 & 32.62 & 32.43 \\\\
>     \\text{SSIM}\\uparrow & 0.891 & 0.890 & 0.889 & 0.892 & 0.908 & 0.894 \\\\
>     \\text{LPIPS}\\downarrow & 0.081 & 0.081 & 0.077 & 0.067 & 0.062 & 0.128 \\\\
> \\hline
>     \\text{Test (Neural vs Analytical)} & \\sigma=0.01 & \\sigma=0.02 & \\sigma=0.05 & \\sigma=0.1 & \\sigma=0.2 & \\sigma=0.3 \\\\
> \\hline
>     \\text{PSNR}\\uparrow & 28.66 & 28.67 & 28.83 & 29.39 & 30.24 & 30.40 \\\\
>     \\text{SSIM}\\uparrow & 0.827 & 0.828 & 0.839 & 0.863 & 0.895 & 0.880 \\\\
>     \\text{LPIPS}\\downarrow & 0.237 & 0.232 & 0.212 & 0.152 & 0.105 & 0.156 \\\\
> \\hline
> \\end{array}
> $$
> We observe a good alignment between the neural and LE-MMSE estimator both on the train ($\\geq 32$dB) and the test set ($\\geq 28.6$dB), showcasing the relevance of our framework for scientific inverse problems. We will include these results in the final version of the paper.
>
> > *UNet2D may not be exactly translation equivariant due to downsampling behavior; should wording in Sec. 4.1 be changed to approximately translation equivariant?*
>
> We agree that the original UNet2D architecture is not exactly translation equivariant due to the downsampling and upsampling operations. In our experiments, however, we use a modified version of UNet2D that incorporates circular padding and avoids any operations that break translation equivariance. The details of this modified architecture are described in supplementary material. We will clarify this point in Sec. 4.1 to avoid any ambiguity.
>
> > *How does LE-MMSE extend beyond strictly local translation-equivariant CNNs to architectures with nonlocal interactions (e.g., adding attention blocks in UNet2D)?*
>
> Pure CNNs remain a relevant class of neural networks from a practical perspective (see response to **6bxG**), though understanding how to extend the LE-MMSE framework to architectures with nonlocal interactions is an appealing direction for future work.
> Regarding your question:
> - The LE-MMSE framework does not extend directly to non-local, non translation-equivariant models such as transformers.
> Breaking the local, translation-equivariant inductive bias will likely break the LE-MMSE/network alignment. Some attention mechanisms are known to be token-permutation-equivariant, which is much weaker.
> - Different notions of locality or equivariance could be considered. A structural description of the functional class reachable by a family of network models is needed.  The key technical difficulty is to obtain and implement an expression for the resulting constrained MMSE.
> - The conjunction of attention with additional devices (MLP, CNNs, positional embedding, etc), requires describing the functional class of the resulting architecture, which is a non-trivial task.

---

> > ### Author Rebuttal · Reviewer_jeRx · 2026-04-04
> >
> > The rebuttal fully resolves my concerns. The authors provide additional evidence that the LE-MMSE interpretation remains relevant beyond standard natural image restoration by including a scientific inverse-problem experiment, clarify the architectural point regarding translation equivariance, and give a clear and appropriately scoped discussion of how the theory relates to architectures with nonlocal interactions.

---

> > > ### Author Response · Authors · 2026-04-04
> > >
> > > We thank the reviewer for their thoughtful engagement with our rebuttal and for the positive feedback. We are pleased that the additional scientific inverse-problem experiments and the clarifications regarding architectural equivariance and nonlocal interactions successfully addressed your concerns. We appreciate your recognition of the scope and theoretical grounding of our work

---

### Official Review · Reviewer_KWz4 · 2026-03-20

**Soundness:** 3
**Presentation:** 3
**Significance:** 3
**Originality:** 3
**Overall Recommendation:** 4
**Confidence:** 3

**Summary:**

This paper proposes a theoretical framework for understanding supervised CNNs for linear inverse problems through constrained empirical MMSE estimators. Specifically, the work investigates whether the architectural biases of translation equivariance and locality are sufficient to derive a closed-form surrogate that closely predicts trained network outputs. The paper defines MMSE, E-MMSE, and LE-MMSE, derives a pixelwise closed form for LE-MMSE, interprets it as a patchwork recombination of training patches. These theoretical insights are then empirically validated on denoising, inpainting, and deconvolution tasks across FFHQ, CIFAR-10, FashionMNIST, and several CNN architectures, such as ResNet and UNet.

**Compliance With Llm Reviewing Policy:**

Affirmed.

**Key Questions For Authors:**

1, Since this paper is closely connected to the Kamb & Ganguli, 2025, what is the clearest conceptual advance beyond extending the framework from denoising/diffusion-style settings to linear inverse problems?

2, Based on the empirical evidence in Sec. 4.4, is it possible to derive this smoothing behavior directly from the LE-MMSE framework itself? Furthermore, would analyzing other inductive biases of CNNs help improve the mathematical alignment in low-density regimes?

3, It is unclear how well the paper’s conclusions extend beyond the small-resolution regime studied here. The main validation is at 32×32, and the limited 64×64 results already appear to show degraded analytical-vs- CNN agreement in low-density regions. This raises concern about whether LE-MMSE remains a faithful surrogate for CNN behavior as image resolution increases.

**Limitations:**

Yes, the authors discuss the limitations of their theory.

**Strengths And Weaknesses:**

Strengths:

1, The paper’s main technical strength is the derivation of closed-form constrained MMSE estimators, especially LE-MMSE, for inverse problems with locality and equivariance constraints. The resulting formula is concrete, interpretable, and empirically shows strong pixel-level agreement with trained CNN outputs across multiple tasks and architectures.

2, The paper provides useful conceptual insight by showing how locality shifts the estimator from global memorization to local patch recombination, and by analyzing the tradeoff between physics-aware and physics-agnostic choices of $B$.

3, The paper is generally well written and easy to follow, with clear and consistent mathematical notation. The experimental evaluation and ablations are comprehensive and provide solid support for most of the main claims.

Weakness:

1, The core conceptual framework, using translation equivariance and locality constraints to derive LE-MMSE, appears to rely substantially on prior work, especially Kamb & Ganguli (2025). The authors should clarify more precisely about their conceptual contributions. The paper is best understood as a mathematically nontrivial but still follow-up extension of that framework from diffusion/denoising settings to linear inverse problems.

2, The connection to real CNNs remains indirect. The theory is exact for an abstract surrogate class, not for trained CNNs themselves, and the paper gives no theorem bounding the gap between LE-MMSE and actual networks.

3, The density-gap discussion is empirical rather than theortical driven. The paper shows that LE-MMSE and CNNs align better in high-density regions of $p(y)$, but does not explain why CNNs themselves should exhibit that same density-dependent behavior from inductive bias perspective.

---

> ### Author Rebuttal · Authors · 2026-03-28
>
> We thank the reviewer for the positive evaluation of our work. We greatly appreciate the constructive feedback.
>
> > *The framework appears to rely on prior work (Kamb and Ganguli, 2025). What is the clearest conceptual advance beyond extending the framework from denoising/diffusion settings to linear inverse problems?*
>
> We agree that Kamb & Ganguli (2025) [A] is a key inspiration. While their analysis focuses on *denoising $A = B = I$ in diffusion models*, our work introduces several conceptual and technical advances:
>
> - **Extension to general linear inverse problems**: the formula involve *operator-dependent metrics $Q_n$* with nontrivial null spaces, which has no analogue in the denoising setting.
> - **Physics-aware vs agnostic:** our work reveals a *bias–variance trade-off*: $B$ improves signal discrimination but may amplify noise. This is the *first theoretical comparison* between physics-aware and physics-agnostic networks.
> - **Direct prediction of trained CNN:** [A] evaluates the score and integrate over time. In contrast, we  compare the analytic estimator and the trained CNN for fixed $\sigma$, providing a more direct validation of the theory as a surrogate for trained networks.
> - **Patch size, density, and alignment:** we explicitly analyze how the *measurement density* $p(y)$ affects alignment (Figs. 6, 17-19), showing that agreement depends on *patch-space density*, explaining both strong matches and failure modes. In contrast, [A] progressively reduce the patch size during diffusion for a better alignment.
> - **Patchwork structure:** we show that reconstructions are *patchwise recombinations of training data* in general inverse problems (Figs. 23–24), whereas [A] establishes this in the diffusion limit.
>
> > *Theory is exact for a surrogate class, but no theorem bounds the gap between LE-MMSE and trained CNNs*
>
> Our theory is exact for a class $\mathcal{M}$, which does not exactly coincide with the range $\mathcal{M}'$ of a network. We use $\mathcal{M}$ as a *tractable surrogate capturing key inductive biases* without accounting for the approximation error. Deriving an explicit error bound between $\mathcal{M}$ and $\mathcal{M}'$ requires strong assumptions. Yet, we show that:
> - different architectures (UNet, ResNet, PatchMLP) converge to *nearly identical estimators*,
> - and these estimators match the LE-MMSE prediction with high accuracy (PSNR $\geq 25$ dB).
>
> This suggests that, in practice, the dominant behavior of trained CNNs is governed by the functional constraints defining $\mathcal{M}$, rather than architectural details.
>
> Please also refer to the response to reviewers **jeRx** and **6bxG** for further discussion on the connection to other architectures.
>
> > *The density-gap analysis is mainly empirical and does not yet explain why CNN inductive biases should induce the same density-dependent behavior*
>
> The density-dependent behavior is not specific to LE-MMSE, but follows more generally from *empirical risk minimization (ERM) under finite data*:
>
> - The LE-MMSE estimator explicitly minimizes risk under the empirical distribution $p(y)$, which is a Gaussian mixture centered on training measurements. As a result, it naturally gives higher weight to *high-density regions*.
> - Trained CNNs are also obtained via ERM on samples from the same distribution. Therefore, their behavior is primarily data-constrained in high-density regions, and driven by inductive biases in *low-density regions*.
>
> The observed alignment can be explained as follows: both LE-MMSE and CNNs are driven by the same empirical distribution, and thus exhibit similar behavior in high-density regions, with increasing variability outside this regime. In the low-density regions, the inductive biases (neural network architecture/optimization dynamics) play a larger role in shaping the behavior of trained networks. Our analysis mostly applies to the "local generalization" regime, close to the training distribution, as illustrated in Fig. 6.
>
> > *From the evidence in Sec. 4.4, can the smoothing behavior be derived directly from the LE-MMSE framework? Would including other CNN inductive biases improve alignment in low-density regimes?*
>
> Locality and translation equivariance do not capture the CNN smoothing inductive bias (spectral bias). Deriving a closed-form MMSE estimator incorporating additional smoothing constraints is an interesting but challenging direction, as it requires both (i) formalizing these biases at the functional level and (ii) maintaining analytical tractability.
> The experiment in Sec. 4.4 suggests that this would further improve alignment, but theoretical aspect is beyond the scope of this work.
>
> > *How well do the conclusions extend beyond the small-resolution regime?*
>
> The conclusion extends beyond the small-resolution mentioned in the submission. We conducted several additional experiments on 128x128 images. Please refer to the rebuttal to the reviewers **6bxG** and **jeRx** for quantitative results.

---

> > ### Author Rebuttal · Reviewer_KWz4 · 2026-04-03
> >
> > The authors have fully address my concerns, and I do not have further questions.

---

> > > ### Author Response · Authors · 2026-04-04
> > >
> > > We thank the reviewer for their thoughtful engagement with our rebuttal and for the positive feedback. We are pleased that the additional experiments and the clarifications successfully addressed your concerns. Given that these points were central to your initial assessment, we are glad the manuscript is now significantly strengthened in these areas.

---

### Decision · Program_Chairs · 2026-04-30

**Decision:**

Accept (regular)

**Comment:**

This paper proposes a theoretical framework for understanding supervised CNNs for linear inverse problems through constrained empirical MMSE estimators. Specifically, the work investigates whether the architectural biases of translation equivariance and locality are sufficient to derive a closed-form surrogate that closely predicts trained network outputs.

Strengths.
+ The derivation of closed-form constrained MMSE estimators, especially LE-MMSE, for inverse problems with locality and equivariance constraints is a good technical contribution.

+ The paper provides useful conceptual insights by showing how locality shifts the estimator from global memorization to local patch recombination.

+ The paper is generally well written and easy to follow, with clear and consistent mathematical notation.

Weaknesses.
- Most of the experiments are limited to tiny images.
- It is mainly a theory paper and architectural coverage remain relatively narrow.

The paper received four reviews with ratings 4,5,5,5.

Authors submitted rebuttals with clarification and additional experiments.

Overall, a strong paper with consensus among reviewers toward accept.